# 'Palaeoshellomics' reveals the use of freshwater mother-of-pearl in prehistory

Jorune Sakalauskaite[1,2], Søren H Andersen[3], Paolo Biagi[4], Maria A Borrello[5], Théophile Cocquerez[2], André Carlo Colonese[6], Federica Dal Bello[7], Alberto Girod[8], Marion Heumüller[9], Hannah Koon[10], Giorgia Mandili[7,11], Claudio Medana[7], Kirsty EH Penkman[12], Laurent Plasseraud[13], Helmut Schlichtherle[14], Sheila Taylor[12], Caroline Tokarski[15†], Jérôme Thomas[2], Julie Wilson[16], Frédéric Marin[2], Beatrice Demarchi[1,6]*

[1]Department of Life Sciences and Systems Biology, University of Turin, Turin, Italy; [2]UMR CNRS 6282 Biogéosciences, University of Burgundy-Franche-Comté, Dijon, France; [3]Moesgaard Museum, Højbjerg, Denmark; [4]Department of Asian and North African Studies, University of Ca' Foscari, Venice, Italy; [5]Independent researcher, Genève, Switzerland; [6]Department of Archaeology, University of York, Heslington, United Kingdom; [7]Department of Molecular Biotechnology and Health Sciences, University of Turin, Turin, Italy; [8]Italian Malacological Society, Sorengo, Switzerland; [9]Niedersächsisches Landesamt für Denkmalpflege, Hannover, Germany; [10]School of Archaeological and Forensic Sciences, University of Bradford, Bradford, United Kingdom; [11]Centre for Experimental and Clinical Studies, University of Turin, Turin, Italy; [12]Department of Chemistry, University of York, Heslington, United Kingdom; [13]Institute of Molecular Chemistry, ICMUB UMR CNRS 6302, University of Burgundy-Franche-Comté, Dijon, France; [14]Landesamt für Denkmalpflege im Regierungspräsidium Stuttgart, Gaienhofen, Germany; [15]Miniaturization for Synthesis, Analysis & Proteomics, USR CNRS 3290, University of Lille, Lille, France; [16]Department of Mathematics, University of York, Heslington, United Kingdom

*For correspondence:
beatrice@palaeo.eu

Present address: †Institute of Chemistry & Biology of Membranes & Nano-objects, UMR CNRS 5248, Proteome Platform, University of Bordeaux, Bordeaux, France

Competing interests: The authors declare that no competing interests exist.

**Abstract** The extensive use of mollusc shell as a versatile raw material is testament to its importance in prehistoric times. The consistent choice of certain species for different purposes, including the making of ornaments, is a direct representation of how humans viewed and exploited their environment. The necessary taxonomic information, however, is often impossible to obtain from objects that are small, heavily worked or degraded. Here we propose a novel biogeochemical approach to track the biological origin of prehistoric mollusc shell. We conducted an in-depth study of archaeological ornaments using microstructural, geochemical and biomolecular analyses, including 'palaeoshellomics', the first application of palaeoproteomics to mollusc shells (and indeed to any invertebrate calcified tissue). We reveal the consistent use of locally-sourced freshwater mother-of-pearl for the standardized manufacture of 'double-buttons'. This craft is found throughout Europe between 4200–3800 BCE, highlighting the ornament-makers' profound knowledge of the biogeosphere and the existence of cross-cultural traditions.
DOI: https://doi.org/10.7554/eLife.45644.001

## Introduction

The selection of shell as a raw material by prehistoric populations implies that it possesses an inherent attractiveness that makes it suitable for displaying social connections, wealth and prestige (*Bar-*

**eLife digest** Just like people do today, prehistoric humans liked to adorn themselves with beautiful objects. Shells, from creatures like clams and snails, were used to decorate clothing or worn as jewelry at least as far back as 100,000 years ago. Later people used shells as the raw materials to make beads or bracelets. Learning where the shells came from may help scientists understand why prehistoric people chose certain shells and not others. It may also offer clues about how they used natural resources and the cultural significance of these objects. But identifying the shells is difficult because they lose many of their original distinctive features when worked into ornaments.

New tools that use DNA or proteins to identify the raw materials used to craft ancient artifacts have emerged that may help. So far, scientists have mostly used these genomic and proteomic tools to identify the source of materials made from animal hide, ivory or bone – where collagen is the most abundant protein molecule. Yet it is more challenging to extract and characterize proteins or genetic material from mollusc shells. This is partly because the amount of proteins in shells is at least 300 times lower than in bone, and also because the makeup of proteins in shells is not as well-known as in collagen.

Sakalauskaite et al. have now overcome these issues by combining the analytical tools used to study the proteins and mineral content of modern shells with those of ancient protein research. They then used this approach, which they named palaeoshellomics, to extract proteins from seven "double-buttons" – pearl-like ornaments crafted by prehistoric people in Europe. The double-buttons were made between 4200 and 3800 BC and found at archeological sites in Denmark, Germany and Romania. Comparing the extracted proteins to those from various mollusc shells showed that the double-buttons were made from freshwater mussels belonging to a group known as the Unionoida.

The discovery helps settle a decade-long debate in archeology about the origin of the shells used to make double-buttons in prehistoric Europe. Ancient people often crafted ornaments from marine shells, because they were exotic and considered more prestigious. But the results on the double-buttons suggest instead that mother-of-pearl from fresh water shells was valued and used by groups throughout Europe, even those living in coastal areas. The palaeoshellomics technique used by Sakalauskaite et al. may now help identify the origins of shells from archeological and palaeontological sites.

DOI: https://doi.org/10.7554/eLife.45644.002

*Yosef Mayer et al., 2009*; *d'Errico et al., 2005*; *Giacobini, 2007*; *Kuhn et al., 2001*; *Saunders, 1999*; *Zilhão et al., 2010*). Unravelling these nuances is especially important in times of life-style transformation (e.g. from mobile hunting and gathering to sedentism), when the way people perceived their 'homeland' may also have shifted: personal adornments thus help track cultural continuity or discontinuity (*Rigaud et al., 2018*; *Rigaud et al., 2015*; *Stiner, 2014*; *Stiner et al., 2013*; *Taborin, 1974*; *Vanhaeren and d'Errico, 2006*). *Exotic* marine shells, transported to inland sites, are thought to have acquired special value and are interpreted as a marker of status and a proxy for long-distance exchange and trade (*Alarashi et al., 2018*; *Bajnóczi et al., 2013*; *Borrello and Micheli, 2011*; *Taborin, 1974*; *Trubitt, 2003*). The exploitation of *local* shells is instead usually viewed in utilitarian terms (e.g. *Colonese et al., 2011*), as an easily accessible, convenient, but inherently less prestigious resource. However, the collection of certain local marine shells may have a deeper meaning; for example, creating a feeling of familiarity with a new environment for seafarers and colonisers (*Bar-Yosef Mayer, 2018*). The use of *freshwater* shells as raw material, although often documented in archaeological sites, has been somewhat overlooked (*Borrello and Girod, 2008*), and the relative importance of freshwater over marine shells has never been systematically addressed, introducing a bias in archaeological interpretations.

A typical example of such bias is the interpretation as *exotica* of the findings of prehistoric mother-of-pearl (shell) miniature double-buttons (*doppelknöpfe*), worked in a way to look like 'true' pearls (*Figure 1*). Experimental archaeological work has shown that they are excellent as ornaments pressed into thin leather, for example armbands or belts (*Kannegaard, 2013*). The raw materials

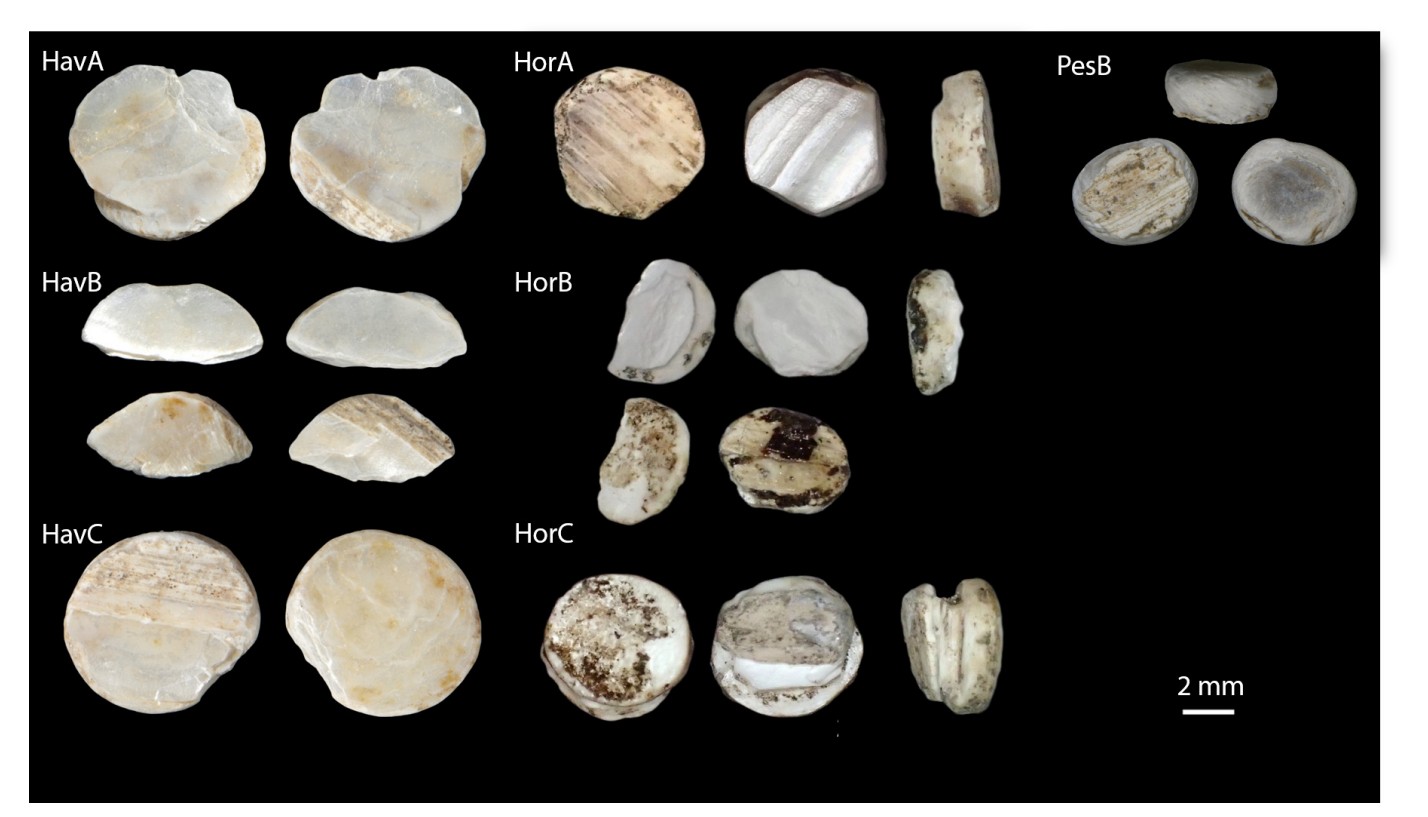

**Figure 1.** The double-buttons analysed in this study.
DOI: https://doi.org/10.7554/eLife.45644.003

actually vary from shell to (possibly) copies made of stone, teeth, bone and ceramic, but they are all consistently white, and their appearance standardized (*Heumüller, 2009*). Double-buttons have been reported primarily from a range of Neolithic Central European sites on the Danube Valley, but occasional findings occur from the Ligurian coast to Northern and Southeastern Europe (*Girod, 2010a*; *Heumüller, 2012*). Among these sites, the shell double-buttons from the Danish Ertebølle/Early Funnel Beaker shell midden at Havnø (*Andersen, 2008*) and the hundreds of examples from the Neolithic submerged pile dwelling settlement of Hornstaad-Hörnle IA on Lake Constance (*Heumüller, 2009*) are especially significant, as they could represent an instance of exchange of materials or ideas between hunter-gatherers (Havnø) and farmers (Hornstaad-Hörnle IA; *Figures 1* and *2*, *Table 1*). Furthermore, if the double-buttons were made of marine shell (in particular of oyster shells, abundant at Havnø [*Andersen, 2008*]), then a case could be made for exotic materials from the coastal Ertebølle site of Havnø being imported (because highly desirable) to the inland Neolithic settlement of Hornstaad-Hörnle IA (*Rowley-Conwy, 2014*). However, the edible flat oyster (*Ostrea edulis*), so abundant at Havnø, can hardly be the raw material of the double-buttons: visual inspection (later confirmed by our analytical data) revealed that these are made of mother-of-pearl (nacre) and *O. edulis* does not form this microstructure. Furthermore, the 'true' pearl oysters (marine genus *Pinctada*) are not found in North Atlantic cold waters.

Despite being the basis for archaeological inference, the knowledge of the biological origin of molluscan mother-of-pearl in prehistoric Europe is only vague (*Taborin, 1974*). This is usually based on macroscopic and microscopic observations of heavily worked and degraded objects, which have thus lost any diagnostic feature that might have been useful for identification (*Demarchi et al., 2014*). Using analytical methodologies that are the basis of shell biomineralization studies (*Marin et al., 2016*; *Marin et al., 2013*), including the first application of proteomics to archaeological shells ('palaeoshellomics'), we investigated seven archaeological double-buttons, dated

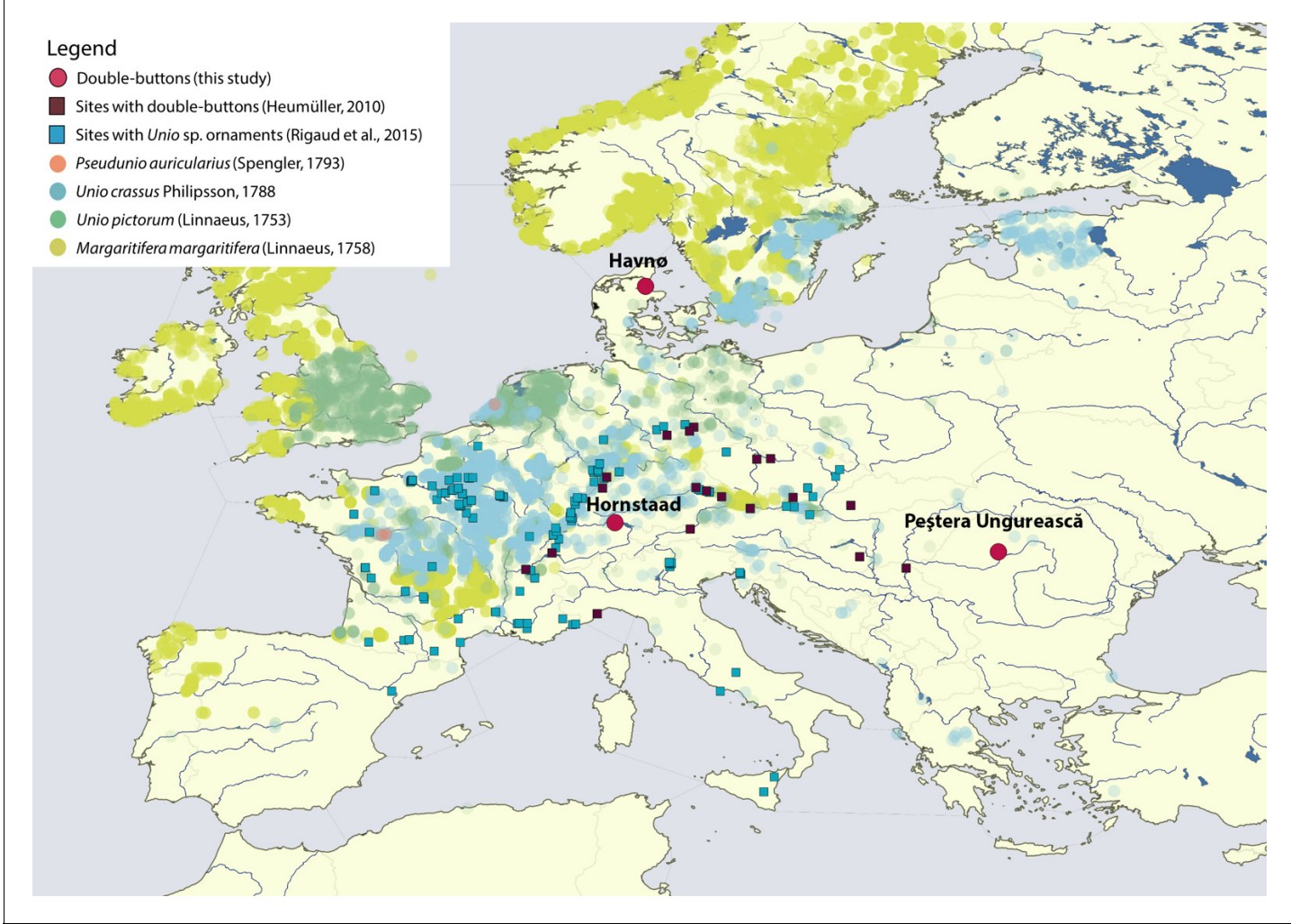

**Figure 2.** Map displaying the location of Havnø, Hornstaad-Hörnle IA and Peştera Ungurească, together with other archaeological sites from which double-buttons (*Heumüller, 2012*) and a variety of ornaments made with *Unio* sp. shells (*Rigaud et al., 2015*) have been reported.
DOI: https://doi.org/10.7554/eLife.45644.004

between ~4200 and ~3800 BCE and recovered from a wide geographic area (*Figure 2*): three (HorA, HorB, HorC) from Hornstaad-Hörnle IA (*Borrello and Girod, 2008*; *Heumüller, 2012*), three (HavA, HavB, HavC) from Havnø (*Andersen, 2008*), and one (PesB) from Peştera Ungurească, a cave site in Transylvania (*Biagi and Voytek, 2006*; *Girod, 2010a*). The richest record comes from Hornstaad-Hörnle IA, where more than 564 double-buttons were recovered, although many were partially fragmented and affected by fire. The origin of the raw material had been debated (*Table 1*): either edible oyster (marine) or freshwater pearl mussel (*Borrello and Girod, 2008*; *Heumüller, 2012*). Around 40 double-buttons were excavated from the shell midden in Havnø - the raw material used to make these was undetermined. A handful of double-buttons were found at the cave site of Peştera Ungurească, together with *Unio crassus* valves, and therefore this freshwater mollusc was assumed to be the raw material used to make the double-buttons at Peştera Ungurească (*Girod, 2010a*).

Our work had two main aims:

1. to develop 'palaeoshellomics', a new molecular approach to characterize ancient proteomes preserved in mollusc shells. The recovery and identification of these proteomes is challenging, because shell contains only a small fraction of proteins embedded in the mineral skeleton (approximately 0.1–1% vs 30% in bone) and there is a lack of molecular sequence data, which

**Table 1.** Summary of the materials analysed in this study (archaeological ornaments and reference shells) and information on their context, chronology and taxonomic determination.

| Sample type | Site | Cultural group | Time span | Taxonomic determination before this study | Other molluscan fauna present at the site |
|---|---|---|---|---|---|
| Double-buttons | Havnø | Ertebølle | 4200–4000 cal BCE (radiocarbon dating of the Late Mesolithic horizon; *Andersen, 2000*; *Andersen, 2008*) | Unknown, presumed marine shell | Abundant edible marine shells: *Ostrea edulis*, *Littorina* sp., *Mytilus edulis*, *Cerastoderma edule* (*Andersen, 2000*; *Andersen, 2008*) |
| | Hornstaad-Hörnle IA | Hornstaad group (early phase of the regional Late Neolithic) | 3917–3902 BCE (dendrochronology; *Billamboz, 2006*) | Debated: marine (*Ostrea edulis*) vs freshwater (*Margaritifera margaritifera*; *Heumüller, 2010*; *Borrello and Girod, 2008*) | Mediterranean marine shells (exotic, non edible): *Columbella rustica* (*Borrello and Girod, 2008*), *Callista chione*, *Astarte borealis*, *Dentalium vulgare* (*Heumüller, 2010*) |
| | Peştera Ungurească | Toarte Pastilate and transition to Coţofeni | 4260–3820 cal BCE (range of radiocarbon dates, at 1σ, of layers 2B, 2A3 and 2A, Toarte Pastilate) (*Biagi and Voytek, 2006*) | *Unio* cf. *crassus* (*Girod, 2010a*, *Figure 6B*) | Abundant terrestrial taxa (naturally occurring). Occasional presence of freshwater species (*Anisus spirorbis*, *Pisidium milium*, *Lithoglyphus naticoides*, *Lymnaea truncatula*, *Planorbis* cf. *carinatus*; *Girod, 2010a*) |
| Reference shell | Limfjord (Northern Jutland) | | Modern | *Ostrea edulis* | |
| | Limfjord (Northern Jutland) | | Modern | *Modiolus modiolus* | |
| | Limfjord (Northern Jutland) | | Modern | *Margaritifera margaritifera* (determined by F.M.) | |
| | France (Izeure) | | Modern | *Unio pictorum* | |
| | Peştera Ungurească | Toarte Pastilate and transition to Coţofeni | 4260–3820 cal BCE (range of radiocarbon dates, at 1σ, of layers 2B, 2A3 and 2A, Toarte Pastilate) (*Biagi and Voytek, 2006*) | *Unio* cf. *crassus* (*Girod, 2010a*) | Abundant terrestrial taxa (environmental signal). Occasional occurrence of freshwater species (*Anisus spirorbis*, *Pisidium milium*, *Lithoglyphus naticoides*, *Lymnaea truncatula*, *Planorbis* cf. *carinatus*). Not suitable as raw material for the double-buttons (*Girod, 2010a*) |
| | Isorella | Vhò | 5226–5023 cal BCE at 2σ (*Starnini et al., 2018*) | *Pseudunio auricularius* (*Biddittu and Girod, 2003*; *Girod, 2010b*) | Marine taxa (typically used as ornaments): *C. rustica*, *Spondylus* (fragment of a bracelet; *Girod, 2010b*) |

DOI: https://doi.org/10.7554/eLife.45644.005

are needed for comparing taxa within the same clade. When studying ancient materials, these issues are further compounded by the effects of diagenesis and by the small sample sizes available. Therefore, developing palaeoshellomics has an impact on understanding the fundamental mechanisms of biomineralization (protein-mineral interactions) in molluscan shells and other invertebrates, the extent of proteome variability within the same molluscan clade, as well as the diagenetic pathways of degradation and preservation.

2. to integrate 'palaeoshellomics' within a set of well-established analytical techniques, in order to determine the origin of the shell ornaments from Hornstaad-Hörnle IA and Havnø, and to confirm that of the double-buttons from Peştera Ungurească. This is archaeologically significant, as similar ornaments were recovered from three geographically distant sites belonging to three different and broadly contemporary cultural groups (*Table 1*): Late Mesolithic (Ertebølle), early Late Neolithic (Hornstaad Group), and Copper Age (Toarte Pastilate/Coţofeni).

To achieve both of these goals, the archaeological double-buttons were studied alongside a selected set of reference shells (*Table 1*). These include two marine species (*Ostrea edulis*, *Modiolus modiolus*), which commonly occur in northwestern European seas, and four freshwater mussels,

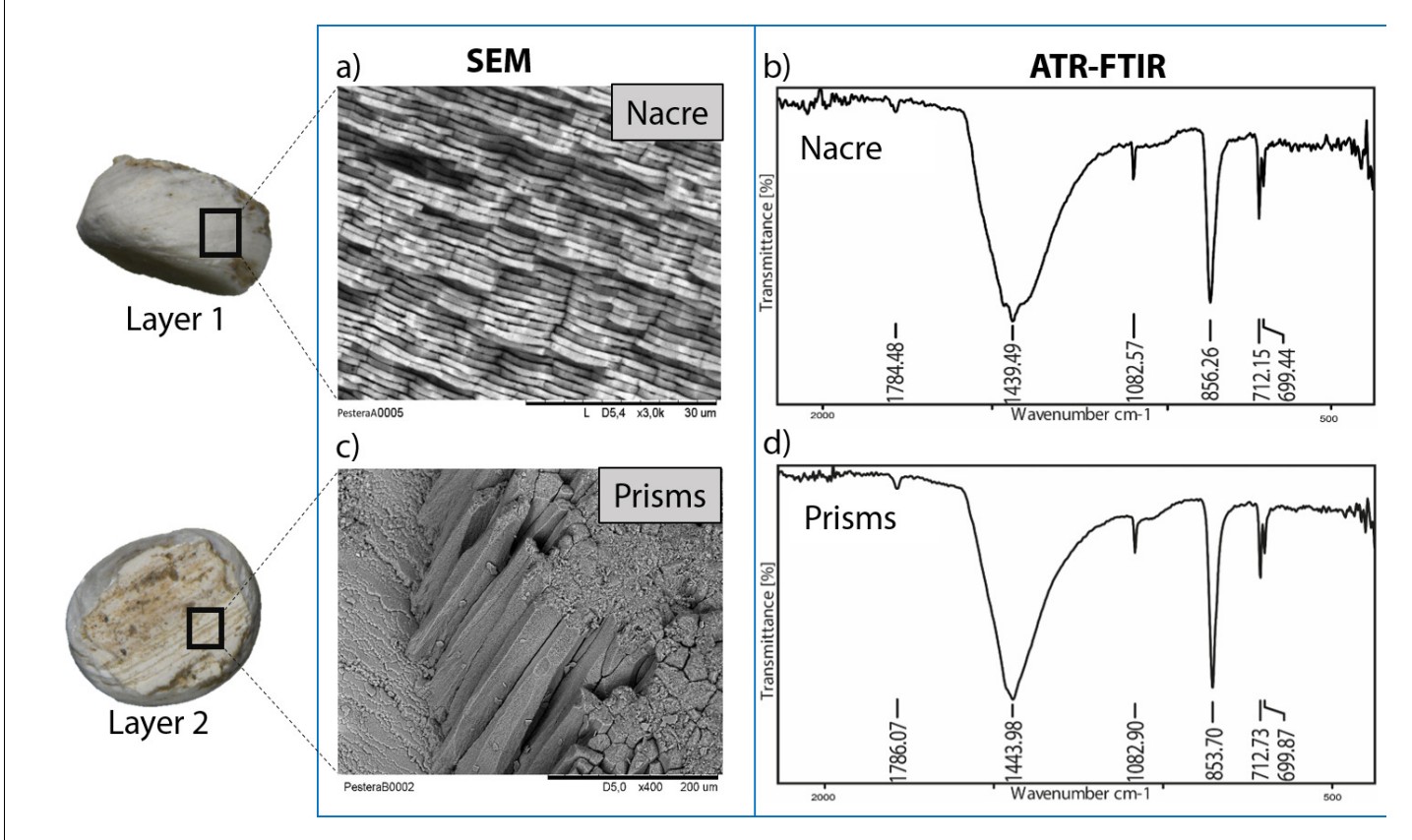

**Figure 3.** Microstructure (SEM) and mineralogy (FTIR-ATR) of double-buttons, showing shiny nacreous (a) and matte prismatic (c) layers, both aragonitic (b, d).

DOI: https://doi.org/10.7554/eLife.45644.006

which are characterized by a thick layer of mother-of-pearl, that is *Unio pictorum*, *Unio crassus*, *Margaritifera margaritifera*, *Pseudunio auricularius*.

## Sample selection

Two modern marine mollusc shells, *O. edulis* and *M. modiolus*, were collected in northern Jutland (Denmark) by Søren H. Andersen and were selected for the following reasons: *O. edulis* shells had been suggested as the potential raw material for the Hornstaad-Hörnle IA assemblage (*Heumüller, 2010*) and are very abundant at the shell midden site of Havnø; *M. modiolus* is a thick-shelled mussel with a nacreous layer, therefore a suitable raw material for the Havnø ornaments (Appendix 1, section 2). Furthermore, close relatives of both species are present in public sequence databases, which is important for palaeoproteomics: *O. edulis* belongs to family Ostreidae (genomes available for *Crassostrea gigas* and *C. virginica*) and *M. modiolus* to family Mytilidae (genome available for *Mytilus galloprovincialis*).

With regard to the freshwater species (order Unionoida), *U. pictorum* and *U. crassus* belong to family Unionidae, *P. auricularius* and *M. margaritifera* to family Margaritiferidae. Modern *U. pictorum* shells were collected in a stream close to Izeure (Burgundy) by Frédéric Marin and modern *M. margaritifera* was collected in northern Jutland by Søren H. Andersen. The morphological determination of both taxa was carried out by Frédéric Marin. *U. crassus* and *P. auricularius* are archaeological shell specimens from the sites of Peştera Ungurească and Isorella (Neolithic, Po Plain, Italy [*Starnini et al., 2018*]). The determination of *U.* cf. *crassus* had been carried out by Alberto Girod on the basis of morphological observations of the whole shell valves and comparison to extant specimens from the area (*Girod, 2010a*). Archaeological *P. auricularius* was used as this species is critically endangered (*Altaba, 1990*) and extant populations rare (*Appendix 1—figure 1*). The morphological

determination of the species had been carried out by Alberto Girod using comparative specimens from museum collections (*Biddittu and Girod, 2003*). An advantage of including archaeological shells as reference materials is that we were able to assess the extent of molecular preservation in shells that are contemporary to the double-buttons. Furthermore, whole and fragmented valves of *U.* cf. *crassus* had been recovered from all the archaeological layers that yielded the double-buttons at Peştera Ungurească (*Girod, 2010a*). Therefore, in this case, potential raw material and finished product have experienced the same post-depositional conditions.

None of the Unionoida species is well-represented in public sequence databases, especially with regard to proteins related to shell biomineralization (see Methods section).

## Results and discussion

### Morphological analysis

The double-buttons described here are circular, with a groove in the middle of the body and no perforation. The main body is shiny but all of them have a thin matte layer on one of the two surfaces (*Figures 1* and *3*). These two layers, both aragonitic (as shown by infrared spectroscopy, see Appendix 1, section 3.2), are the nacre and prisms of a mollusc shell. Scanning electron microscopy showed the presence of the 'brickwall' microstructure of nacre (sheet nacre) juxtaposed with the thin layer of prisms, the latter having elongation axes perpendicular to the nacre plane (*Figure 3*). No secondary calcite was observed and there was no sign of the occurrence of diagenetic recrystallization. The overall nacre appearance is typical of bivalves and not of gastropods. The combination of nacroprismatic microstructure and aragonitic mineralogy is observed in freshwater unionoid mussels but also in marine trigonioids and anomalodesmatans (Appendix 1, section 2).

### Stable isotopes of carbon and oxygen

Stable isotope analyses for all of the samples yielded average $\delta^{18}O$ and $\delta^{13}C$ values of $-5.3 \pm 0.4$ and $-11.1 \pm 0.6$ ‰ for Havnø, $-6.1 \pm 1.0$ and $-11.9 \pm 1.7$ ‰ for Peştera Ungurească and $-9.3 \pm 0.5$ and $-10.6 \pm 1.6$ ‰ for Hornstaad, respectively (Appendix 1, section 3.3). The consistently low $\delta^{18}O$ and $\delta^{13}C$ values of shells from Peştera Ungurească and Hornstaad indicate a local freshwater origin for the shells (*Keith et al., 1964*; *Leng and Lewis, 2016*), whereas the $\delta^{18}O$ values at Havnø suggest some mixing of marine water or changes in the atmospheric circulation, with precipitations slightly enriched in $^{18}O$ compared to present day over the region. Our interpretations are broadly supported by the average annual $\delta^{18}O$ values of modern local precipitations for the sites (*WaterIsotopes.org, 2018*). The isotope data therefore suggest that the shells were locally sourced (*Keith et al., 1964*).

### Amino acid analysis

The absence of recrystallization observed by SEM was consistent with the concentration, composition and relatively low D/L values for all the amino acids analysed (e.g. alanine D/L ~0.1 for Peştera Ungueraşca, ~0.2 for Havnø and Hornstaad), except for sample HorA. This supports a non-fossil origin for the shells used to make the double-buttons, that is the makers used 'fresh' or recently dead mollusc shells. However, the extent of degradation for HorA was significantly higher and both D/L and concentration values showed a clear 'burning' signal (*Crisp, 2013*; *Demarchi et al., 2011*). The amino acid composition was similar to that of freshwater bivalves (*Unio*, *Margaritifera*) present in the reference database of *Demarchi et al. (2014)*, although HorA and PesB appeared to be rather different from the other double-buttons (Appendix 1, section 3.4).

### Palaeoproteomics

We characterized the proteomes preserved within the seven double-buttons and performed bioinformatic searches (PEAKS 8.5, Bioinformatics Solutions Inc [*Ma et al., 2003*]) of the product ion spectra against both a Protein sequences and an Expressed Sequence Tags (ESTs) database, restricting the taxonomy to Mollusca (see Methods section). This resulted in the identification of 1973 and 3233 peptide sequences, respectively, which represent the 3.5% and 5.1% of the total number of sequences generated by the de novo algorithm of the software (excluding contaminant sequences). For comparison purposes, we also performed shotgun proteomics on the shell matrices extracted

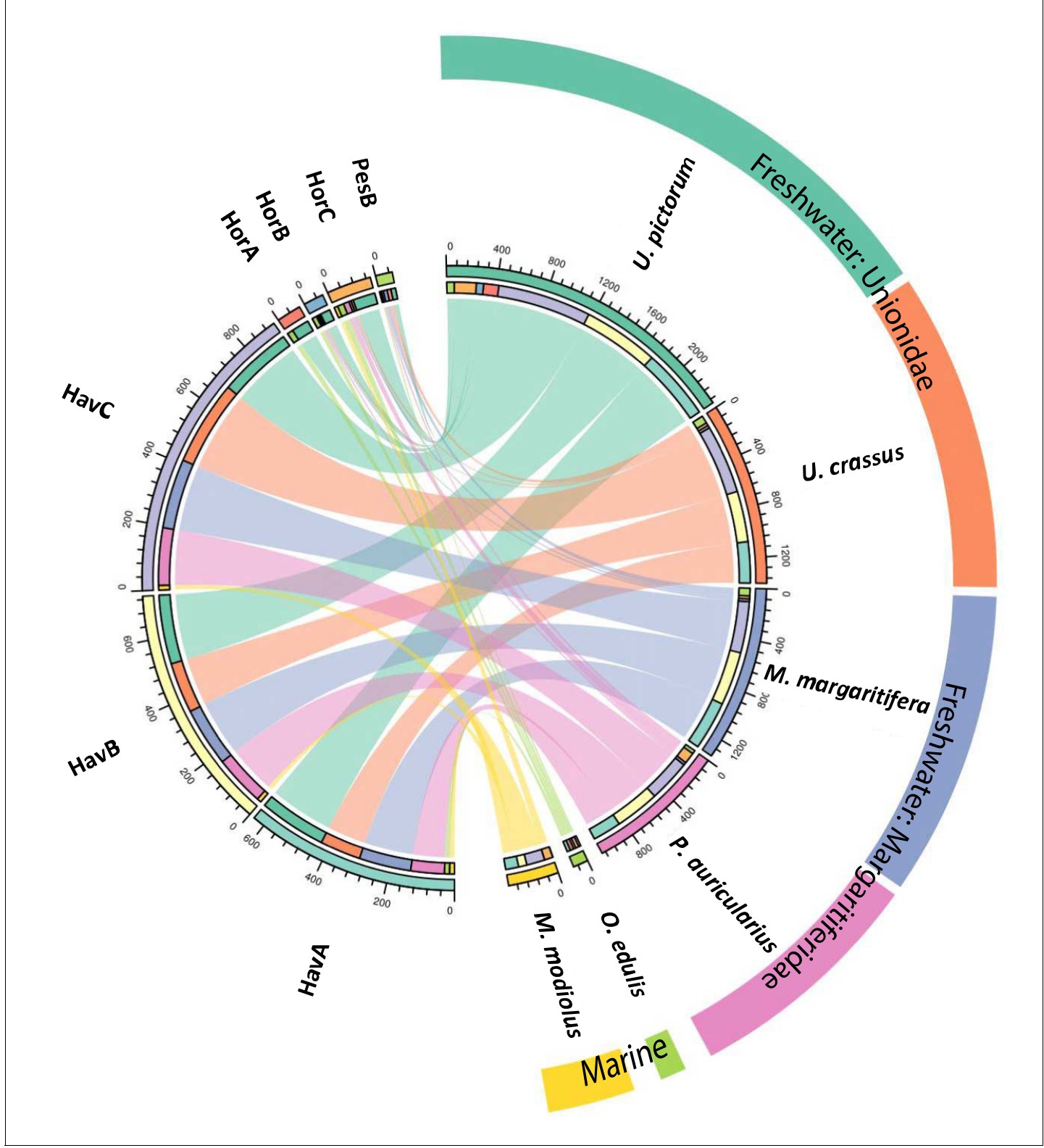

**Figure 4.** Circular diagram representing the similarity between the proteomes of seven double-buttons (left) and six mollusc shell taxa (right).

DOI: https://doi.org/10.7554/eLife.45644.009

The following source data is available for figure 4:

**Source data 1.** R code and data files for *Figure 4*.

DOI: https://doi.org/10.7554/eLife.45644.010

**Table 2.** Main protein sequences identified in the double-buttons from Havnø, Hornstaad-Hörnle IA and Peştera Ungurească and their presence/absence in the analysed set of reference freshwater and marine shells (black dots).

Numbers indicate total number of peptide sequences identified and the cell colour is proportional to the coverage of the sequence itself. Threshold values for peptide and protein identification: false discovery rate (protein FDR) = 0.5%, protein score $-10lgP \geq 40$, unique peptides $\geq 2$, de novo sequences scores (ALC%) $\geq 50$. Asterisks (*) indicate proteins identified only when using less stringent parametres: protein score $-10lgP \geq 20$; unique peptides $\geq 1$. Note that molecular sequence databases for molluscan species are incomplete and biased towards well-studied model organisms. The peptide sequences recovered in our study were identified using sequence homologies with proteins originally described from *Hyriopsis cumingii*, *Crassostrea* sp., *Pinctada* sp., *Mytilus* sp. and several others. As a result of database insufficiency, the bioinformatic search of these 'shellomes' could not identify the exact taxon of our samples, but provided a strong indication of the fact that the closest taxon to that of the ornaments (and of the freshwater reference shells) is the pearl-producing triangle sail mussel *Hyriopsis cumingii* (Unionoida).

| Proteins present in database from | | | Double-buttons | | | | | | | Freshwater | | | | Marine | |
|---|---|---|---|---|---|---|---|---|---|---|---|---|---|---|---|
| Order | Genus | Identified proteins | HavA | HavB | HavC | HorA | HorB | HorC | PesB | U.p | U.c | M.m | P.a | Mo. M | O. e |
| Unionoida | *Hyriopsis* | Hic74 [*Hyriopsis cumingii*] | 132 | 158 | 260 | | 6 | 11 | 21 | • | • | • | • | | |
| | | Hic52 nacreous layer matrix protein [*Hyriopsis cumingii*] | 1* | 1* | 2* | | | | | • | • | • | • | | |
| | | Silkmapin (isoforms: nasilin 1 and nasilin 2) [*Hyriopsis cumingii*] | 1* | 3 | 3 | | 5 | | | • | • | •* | •* | | |
| Ostreida | *Pinctada* | MSI60-related protein [*Pinctada fucata*] | 6 | | 27 | | | 12 | | • | • | | • | • | |
| | | Insoluble matrix protein [*Pinctada fucata*] | 4 | | 33 | | | | | • | • | | • | | |
| | *Crassostrea* | Glycine-rich cell wall structural protein-like [*Crassostrea virginica/gigas*] | 17 | 11 | 12 | | | | 14 | | • | | | | |
| | | Glycine-rich protein 23-like [*Crassostrea virginica*] | 8 | 11 | 6 | | | | | • | | • | | | |
| | | Antifreeze protein Maxi-like [*Crassostrea virginica*] | | 4 | 4 | | | | | | | | • | | |
| Mytilida | *Bathymodiolus* | MSI60-related protein partial [*Bathymodiolus platifrons*] | | 6 | 11 | | | | | • | | | • | | • |
| | *Mytilus* | Precollagen D [*Mytilus edulis*] | 16 | 26 | 23 | | | | 9 | • | • | • | | | |
| | | Nongradient byssal precursor [*Mytilus edulis*] | 10 | 10 | | | | | 25 | | • | | | | |
| Other | *Other* | Predicted: transcription factor hamlet-like partial [*Octopus bimaculoides*] | 5 | 6 | 11 | | | | | • | | • | | | |
| | | Hypothetical protein OCBIM_22008720 mg partial [*Octopus bimaculoides*] | 6 | 11 | | | | | | | | | | | |
| | | Coverage | | | | | | | | ≥55% | ≥35% | ≥15% | ≥10% | ≥1% | |

Presence •

DOI: https://doi.org/10.7554/eLife.45644.007

The following source data is available for Table 2:
Source data 1. Palaeoshellomics.
The complete proteomics dataset obtained on reference shells and archaeological ornaments
DOI: https://doi.org/10.7554/eLife.45644.008

from the six reference mollusc species (Appendix 1, section 3.5) and analysed the data using the same databases and parametres (see *Table 2—source data 1* for full results of the palaeoproteomic analyses).

*Table 2* shows the top-scoring proteins from the seven double-buttons: the numbers indicate the peptides supporting each protein identification, while protein coverage (i.e. the percentage of

sequence for which we could detect peptides) is represented by different colours. Additionally, on the right hand side of the table we indicate if each of the double-button proteins also occurred in the reference shell proteomes (the list of all shell proteins identified in both the double-buttons and the reference shells can be found in *Table 2—source data 1*).

## Shellomes: significant protein hits

The main protein sequences from the double-buttons were identified as belonging to the pearl-producing triangle sail mussel *Hyriopsis cumingii* (*Bai et al., 2013*).

Protein Hic74 (GenBank: ARG42316.1) was found in all of the archaeological samples, except HorA. The percentage coverage for the Hic74 sequence was highest for the Havnø beads (35–55%), where it was supported by 132, 153, 255 unique peptides in HavA, HavB and HavC, respectively (Appendix 1, section 3.5). This protein was also securely identified in all of the freshwater unionoid reference shells (coverage varying from 34% in *M. margaritifera*, supported by 67 peptides, to 50% in *U. crassus*, supported by 203 peptides). Hic74 is an acidic, Ala- and Gly-rich shell matrix protein (*Liu et al., 2017a*). Consisting of 19 poly-A blocks, GA repeats, short acidic motifs (that probably bind to the mineral) and a GS-rich domain at the C-terminus (which resembles that of lustrin-A), this silk fibroin-like protein is likely to play a structural role in nacre formation and in enhancing its mechanical properties (*Liu et al., 2017a*).

Protein Hic52 (GenBank: ARH52598.1) was identified in all the reference unionoid shells and in the Havnø samples, but only when less stringent parameters were used for the identification (i.e. number of unique peptides $\geq 1$ (instead of 2) and protein score $-10\lg P \geq 20$ (instead of 40)). Hic52 is a very basic (theoretical pI > 10), Gly- and Gln-rich protein, with few poly-Q and poly-G blocks and several degenerate G-rich repeats of different lengths along the sequence. It possesses a collagen-like structure which suggests a structural role in nacre formation (*Liu et al., 2017b*). Silkmapin (Gen-Bank: AIZ03589.1, and its isoforms nasilin 1 and 2) are Gly-rich non-acidic proteins with a structural function, probably related to the formation of both nacreous and prismatic layers (*Liu et al., 2015*; *Marie et al., 2017*). Present in the shell matrix of all the unionoids, these proteins were also detected in the Havnø samples and in one of the Hornstaad beads (HorB). Finally, we also identified protein sequences from marine mollusc genera (mainly *Pinctada*, *Crassostrea* and *Mytilus*), but all these 'marine' sequences only displayed repeated low-complexity (RLC) domains (typically consisting of Ala and Gly-rich repeats and/or poly-Ala blocks). RLC-containing peptides are not sufficient for distinguishing between freshwater and marine shells. On the contrary, in double-buttons and in unionoid reference shells, the top-scoring protein Hic74 was supported by remarkably high (for shell proteins) coverages, and, together with Hic52 and silkmapin/nasilin, showed a number of specific peptides that do not exhibit RLC domains. These proteins showed no homologues with any other shell proteins of marine origin currently present in the NCBI database (BLASTp search), being unique to *H. cumingii* and suggesting their specificity to freshwater unionoid shells. We argue that their presence (where identified as the major shell matrix proteins, supported not only by RLC domains) is specific to Unionoida, freshwater mother-of-pearl shells, which in combination with the isotopic data, and supported by the microstructural and amino acid results, excludes a marine origin for the raw material used to make the double-buttons.

## Comparison with reference shells

We performed a search of the product ion spectra from the double-buttons and the six reference shells against the redundant EST database, so that we could recover complementary information from non-annotated sequences. For example, a search of *Hyriopsis cumingii* on NCBI will retrieve 246 protein sequences but 10156 EST sequences. The dataset was used to explore the similarities between double-buttons and molluscan shell proteomes, presented as a circular plot in *Figure 4*. This output was derived from an adjacency matrix, showing which proteins (EST sequence identifiers) occurred in two or more samples (R code and data files can be found in *Figure 4—source data 1*). From this, the subset of unique identifiers, present both in the double-buttons and in any of the reference shells, was associated with its sequence coverage (%) in the archaeological samples. This information is represented in the right-hand side of *Figure 4*: the length of the circular segment for each molluscan species is proportional to the number of sequences that the shell shares with the archaeological samples, scored on the basis of the coverage. On the left-hand side of the graph,

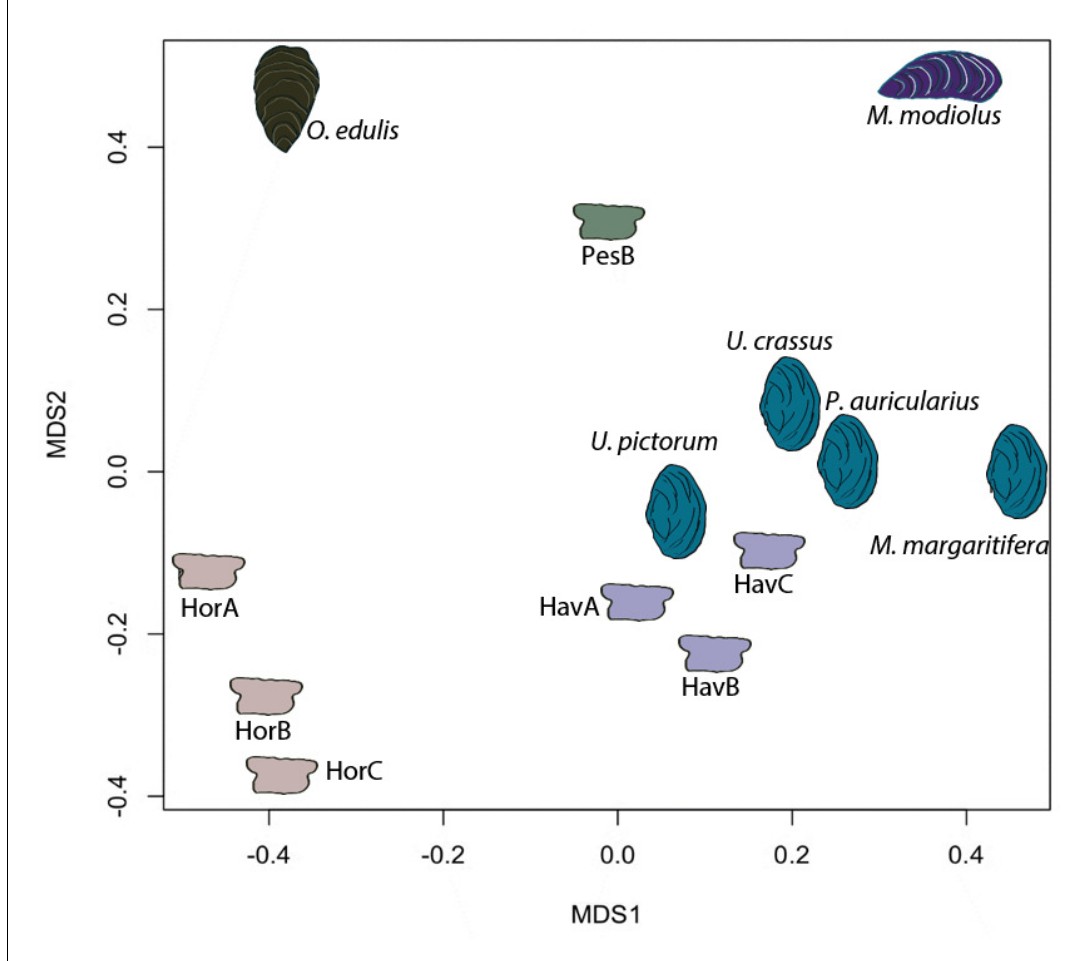

**Figure 5.** Proteome comparison based on peptide sequence similarity, represented by multi-dimensional scaling (MDS).
DOI: https://doi.org/10.7554/eLife.45644.011
The following source code is available for figure 5:

**Source code 1.** Pepmatch code Code (developed in C language) for *Figure 5*.
DOI: https://doi.org/10.7554/eLife.45644.012

each double-button is represented by a circular segment, which is proportional to the number of unique peptides that supports each shared protein sequence. The similarity between double-buttons and mollusc taxa can be visualised through the thickness of the connecting bands. Overall, the data showed that the EST sequences shared between the ornaments and the shells were mainly from the unionoids, consistent with the results obtained by searching the annotated protein database (*Table 2*). From all of the archaeological samples, the Havnø set showed the best match to the fresh-water unionoids, owing to better-preserved proteins, that is with high coverage and number of sup-porting peptides. The LC-MS/MS analysis of PesB produced a number of tandem mass spectra comparable to the other samples, but a lower number of sequences were identified. Sample HorA (burnt), from which no proteins had been identified using the annotated protein database, yielded some matches to EST sequences, most of which shared with *U. pictorum*.

## Database-independent comparison

In order to provide further, independent, evidence for the origin of the raw material, we developed an in-house tool (in C language, available in *Figure 5—source code 1*) for 'proteome comparison', using all the peptide sequences generated by the de novo algorithm of the software PEAKS, that is before performing any database search (Appendix 1, section 3.6). The tool was able to provide a

**Table 3.** Potential amino acid substitutions detected in the samples analysed in this study, compared to the reference Hic74 sequence [*Hyriopsis cumingii*].

Positions are derived from the sequence alignment shown in *Figure 6*. Dashes indicate that the position was not covered for that sample; question mark symbols indicate ambiguous substitutions. Hic74 coverages for each sample and supporting product ion spectra are presented in *Figure 6—figure supplement 1*, *Figure 6—figure supplement 2*, *Figure 6—figure supplement 3*, *Figure 6—figure supplement 4*, *Figure 6—figure supplement 5*, *Figure 6—figure supplement 6*, *Figure 6—figure supplement 7*, *Figure 6—figure supplement 8*, *Figure 6—figure supplement 9*, *Figure 6—figure supplement 10*, as well as in *Figure 6—source data 1*.

| Hic74 [*Hyriopsis cumingii*] | A | A | A | A | A | A | A | G | D | G | S | E | G | A | A | L | V | G | L | I | A | G | A | Q | R | E |
|---|---|---|---|---|---|---|---|---|---|---|---|---|---|---|---|---|---|---|---|---|---|---|---|---|---|---|
| AA position | 83 | 85 | 91 | 93 | 106 | 108 | 110 | 111 | 151 | 152 | 163 | 172 | 175 | 282 | 283 | 284 | 289 | 292 | 306 | 310 | 403 | 801 | 804 | 821 | 822 | 827 |
| *U. crassus* | - | - | A | A | A | A | A | G | D | G | - | Q | G | A | ? | L | ? | G | L | F | A | - | - | Q | R | E |
| *U. pictorum* | - | - | ? | ? | ? | - | - | - | - | - | - | Q | G | - | - | - | - | - | L | F | ? | S | S | Q | R | E |
| *M. margaritifera* | - | - | - | A | - | - | - | - | - | - | - | Q | G | - | A | L | L | G | L | F | A | - | - | Q | R | E |
| *P. auricularius* | - | A | A | A | A | A | A | - | ? | ? | D | E | L | - | A | F | V | E | F | I | - | - | - | H | H | D |
| HavA | A | S | A | A | ? | ? | A | - | - | - | - | Q | G | A | A | L | L | G | L | F | - | - | - | Q | G | E |
| HavB | ? | S | A | A | V | - | - | - | - | - | - | - | - | - | A | L | L | G | L | F | - | - | - | Q | G | E |
| HavC | ? | S | A | A | V | ? | A | D | G | G | - | G | ? | - | A | L | L | G | L | F | - | G | S | Q | G | E |

DOI: https://doi.org/10.7554/eLife.45644.025

score for the sequence similarity between two lists of peptides and to generate a similarity matrix from all pairwise comparisons, which was then converted to a distance matrix. Multidimensional scaling (MDS; *Gower, 1966*) was used to visualise the similarity of observations (*Figure 5*) and confirmed that the Havnø set and the freshwater reference unionoids display the higher degree of proteome similarity, while the samples from Hornstaad and PesB fall in a different area of the plot from each other and from the marine reference shells. The results were also in accordance with those obtained from another database-independent approach *Appendix 1—figure 25*, based on direct product ion spectra comparison (*Rieder et al., 2017*), which was adapted for this study (Appendix 1, section 3.6). Overall, our study, which represents one of the few that attempts to compare molluscan proteomes within the same clade (genus, family or order), shows that unionoid shells exhibit very similar proteome profiles, sharing many sequences between species *Appendix 1—figure 23*. This may suggest that this group has a rather conserved, homogeneous and recognisable proteomic signature, a conclusion that is completely congruent with earlier findings (*Marie et al., 2017*). Furthermore, all analyses showed that the three sets of archaeological ornaments have similar proteome profiles (*Table 2*, *Figures 4* and *5*, *Appendix 1—figure 24*, but do not exhibit a simple correspondence to a molluscan species, at least among the Unionoida considered here, further highlighting the complexity of molluscan shell proteomes.

## Analysis of the Hic74 sequence

We examined the sequence of the top-scoring protein, Hic74, recovered from the reference shells and ornaments, with the aim of assessing the presence and frequency of any amino acid substitutions, which could potentially yield taxonomic resolution within Unionoida. *Figure 6* shows the alignment (performed using the software Geneious Prime 2019.1.1) of these incomplete sequences to the reference (Hic74 from *Hyriopsis cumingii*). The sequence coverage of each sample was obtained from the '*Spider*' output of PEAKS 8.5. The *Spider* algorithm takes into account potential amino acid substitutions, as well as a large number of in vivo, laboratory-induced (e.g. carbamidomethylation), and diagenetically-relevant (e.g. deamidation) modifications, therefore it is especially useful in highlighting possible mutation sites. In our sequence reconstruction we only considered peptides displaying typical sample preparation-induced or diagenesis-induced modifications (*Figure 6—figure supplement 1*, *Figure 6—figure supplement 2*, *Figure 6—figure supplement 3*, *Figure 6—figure supplement 4*, *Figure 6—figure supplement 5*, *Figure 6—figure supplement 6*, *Figure 6—figure supplement 7*, *Figure 6—figure supplement 8*, *Figure 6—figure supplement 9*, *Figure 6—figure supplement 10*). The potential amino acid substitutions and their positions are summarised in *Table 3* (supporting product ion spectra can be found in *Figure 6—source data 1*).

In the best-case scenario, only around half of the Hic74 (*Hyriopsis cummingii*) reference sequence was covered (*Table 2*, *Figure 6*). This may be due to: genuine sequence differences between *Hyriopsis* and all the other Unionoida examined here; low susceptibility of low-complexity domains to enzymatic cleavage; selective *post mortem* (or laboratory-induced) degradation of half of the sequence; errors in the transcriptome assembly of the protein. It is likely that a combination of all of these factors is responsible for this, particularly as the Hic74 regions not covered in the samples are mainly low-complexity domains, sometimes highly polar and thus prone to hydrolysis (Ser and Asp-rich). Within the limits due to the incomplete coverage of the mutation sites in reference shells, it is interesting to note that the Hic74 sequence of *P. auricularius* diverges significantly from those of *U. pictorum*, *U. crassus* and *M. margaritifera*, which appear to have a higher degree of sequence similarity (*Table 3*). While this is in contrast with a recent taxonomy reassessment of Unionida based on mitochondrial DNA (*Lopes-Lima et al., 2018*), it has been noted before that shell proteomes do not follow a simple phylogenetic signal (*Jackson et al., 2010*). The sequence coverage was insufficient to attempt any further consideration for the Hornstaad and PesB samples, but the better-preserved Havnø double-buttons shared the same amino acid substitutions, supporting the hypothesis that the same species was used to make these three ornaments (*Table 3*). Furthermore, *Table 3* shows that this taxon was unlikely to be *P. auricularius*, and more likely to be *Unio* or *Margaritifera* sp.

## Raw material identification

The raw material used to manufacture the seven double-buttons can be firmly and consistently identified as Unionoida, freshwater shells with a thick mother-of-pearl layer, on the basis of morphological, microstructural, mineralogical, geochemical and biomolecular data. From a microstructural viewpoint alone, the combination of aragonitic prisms and sheet nacre ('brickwall type') structures is restricted to three bivalvian orders: the Unionoida, the Trigonioida and the Anomalodesmata, the first two belonging to sub-class Palaeoheterodonta (*Taylor et al., 1973*). The Trigonioida relic order could be ruled out, since it is represented nowadays by a single genus, *Neotrigonia*, with very small shells and living exclusively on the Australian and Tasmanian coasts. Geochemical data, that is stable isotope values of carbon and oxygen, overall indicated that all biominerals studied here were formed in freshwater environments, and $\delta^{18}O$ values in double-buttons tracked the average annual $\delta^{18}O$ values of local precipitations. This excluded Anomalodesmata as potential candidates, since this order of enigmatic, rare and specialised bivalves are strictly marine (*Taylor et al., 1973*). This finally left only one possibility, Unionoida, the representatives of which are all freshwater bivalves.

Biomolecular analyses showed that the proteome similarity is highest between the double-buttons and the unionoid reference shells. The identification of proteins Hic74, Hic52 and silkmapin in almost all of the archaeological samples confirms the freshwater nacre (Unionoida) origins of the double-buttons. With our current knowledge on shell proteins, these sequences probably represent taxon-specific adaptations for the biomineralization of nacroprismatic structures in unionoid shells: they do not bear any homologues with other shell proteins and are not found in the proteomes of other non-nacreous shell structures characterized here (*O. edulis*). Furthermore, the analysis of the amino acid substitutions on the Hic74 sequence, recovered from both the Unionoida shells and the ornaments, indicates that *Unio* or *M. margaritifera* (not *Pseudunio*) had been used for making the Havnø ornaments.

## Sources of bias in 'palaeoshellomics'

Except technical bias, inherent to standard proteomics per se and discussed elsewhere (*Marin et al., 2016*), we identified three potential sources of bias that may hamper, limit or confound the current use of 'palaeoshellomics': 1) the intrinsic peculiarities of several shell matrix proteins; 2) the completeness of the dataset used for identification searches; 3) the diagenetic degradation of shell proteins.

1. Sequence analysis of shell matrix proteins has revealed that a large proportion exhibits unconventional primary structures, with abundant long stretches of repeated residues, the RLCDs/LCDs (repetitive low-complexity domains; *Marie et al., 2013*). The sequences of such domains are neither taxon-specific nor do they carry phylogenetic information. Because they play a structural role, they are ubiquitous and may be detected in phylogenetically-distant lineages. For example, our proteomic analysis identified many sequences known in marine bivalves such as glycine-rich, insoluble matrix and MSI60-related proteins, but these findings are only

supported by RLC domains that can be detected in very different molluscs. More generally, silk fibroin-like domains (poly-Ala), acidic (D-rich or poly-D) or collagen-like repeats (G-X-A triplets) are widespread, ubiquitous and cannot be simply assigned to one given shell protein and/or mollusc genus.

2. As stated before, protein identification is database-dependent and the quality of the interpretation is proportional to the size of the data set. The list of known shell proteins has dramatically expanded in the past few years, with the use of high-throughput screening (*Marin et al., 2016*). However, the 'shellomes' of only about thirty mollusc genera have been identified, in a phylum that comprises between 80000 and 100000 species. We do not know yet whether the set of sequences at our disposal is a representative sample of the sequence diversity for the whole phylum. In other words, the taxonomic origin of a given archaeological shell sample (such as the double-buttons analysed here) may be indeterminable because extant representatives of the corresponding genus (or of closely related genera) are not yet registered in molecular databases via their genomes, mantle transcriptomes or shellomes. As a consequence, palaeoshellomics, for the time being, will provide information mainly at intermediate taxonomic levels (order, family, sub-family). Nevertheless, the continuous expansion of the dataset of shell protein sequences will increase the power of palaeoshellomics: archaeological shell samples, which are currently unidentified, can be revisited and re-interpreted in the future via in silico investigations alone (i.e. without any further proteomic analysis). Furthermore, the detection of possible amino acid mutations in the sequence of different reference shells, such as Hic74 in this study (*Figure 6*, *Table 3*), can be helpful in narrowing the range of candidates and in obtaining more precise taxonomic information.

3. Diagenesis, that is the slow transformation of sediments and their contents with time, also affects shell biominerals, whether from continental, lacustrine or marine environments. As shell biominerals are organo-mineral composites, diagenesis alters shell proteins too. The diagenetic stability of one given shell protein is a complex phenomenon that depends on several parameters, including its primary structure, its conformation, its localization within the biomineral, the availability of water molecules at the vicinity of the protein, the presence of saccharide moieties. Some shell proteins that could be important for taxonomic determination may be diagenetically unstable, that is easily degraded. As a result, the information they carry would be lost. In the present case, we were fortunate to obtain a consistent set of peptides that do not correspond to LC/RLC domains, i.e. peptides that carry relevant information for protein/clade identification. These unambiguously target Hic74, Hic52 and silkmapin proteins that correspond to freshwater unionoid bivalves. Beside taxonomic determination per se, our data suggest that these proteins (especially Hic74) are diagenetically stable and that they may constitute accurate markers for further studies.

While diagenesis may represent a true source of bias, we were able to accurately evaluate its effects, and we found that the extent of protein degradation (racemization, deamidation) was consistent with the age and burial history of the samples. More specifically, we observed that samples from Havnø were the best preserved - the coverage of the main proteins was high, especially for Hic74 (up to 55%). Surprisingly, we found that the coverage of this protein in Havnø samples is similar to that of modern *U. pictorum* and Neolithic *U. crassus* and *P. auricularius* - indeed, the number of Hic74 peptides in Havnø even surpasses that of the reference unionoid shells, despite the fact that the sample size for the archaeological double-buttons was at least 100 times smaller (Appendix 1, section 3). We assume that this effect is due to early diagenetic changes (such as protein unfolding, loss of linked sugars) that render the protein backbone more accessible to proteolytic enzymes, thus increasing the chance of releasing and identifying peptides. Interestingly, we observed a similar phenomenon in other mineralised systems, for example ostrich eggshell (*Demarchi et al., 2016*).

Samples from Hornstaad yielded significantly lower coverages (7% and 12% respectively for HorB and HorC) with fewer supporting peptides (6 and 11; *Table 2*). In one instance (sample HorA), no proteins were identified. This is consistent with the results of the chiral amino acid analysis, which had flagged this sample as compromised and probably burnt, as well as with archaeological evidence for widespread fire destruction of the settlement (*Heumüller, 2012*). However, the same sample yielded a high number of unidentified peptide sequences (~6000), some of which appeared to be highly acidic and reminiscent of biogenic carbonate-associated proteins (*Marin and Luquet, 2008*). Furthermore, neither the microstructure nor the mineralogy of the double-buttons from Hornstaad showed any apparent sign of recrystallization to secondary calcite. We therefore hypothesize that the exposure to high temperatures had been relatively moderate, sufficient for inducing protein

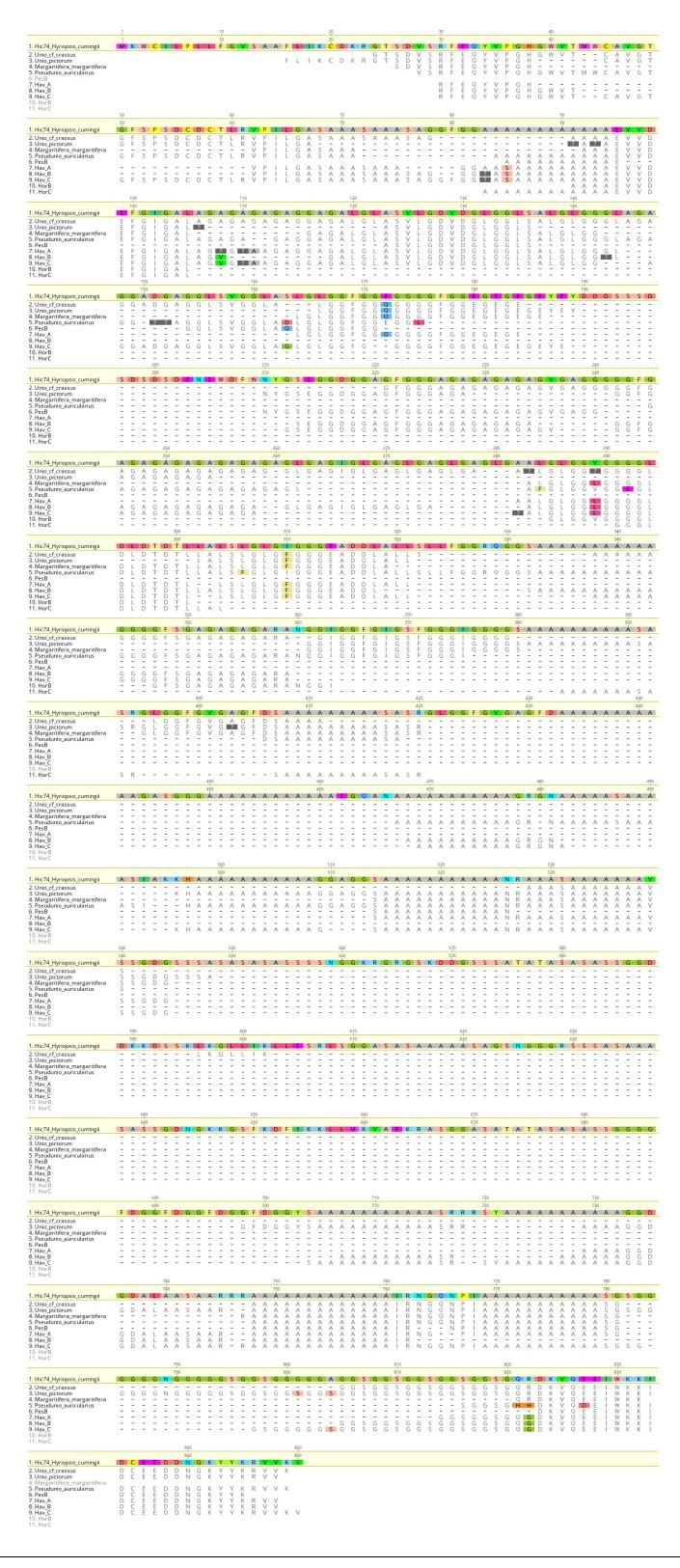

**Figure 6.** Alignment of Hic74 sequences recovered from the Unionoida reference shells and the ornaments. The reference Hic74 [*Hyriopsis cumingii*] is shown at the top of the alignment and is highlighted in yellow. Dashes indicate where the sequence was not covered in the samples analysed in this study; amino acid residues highlighted in colour show all disagreements with the reference Hic74 [*Hyriopsis cumingii*].

DOI: https://doi.org/10.7554/eLife.45644.013

*Figure 6 continued on next page*

*Figure 6 continued*

The following source data and figure supplements are available for figure 6:

**Source data 1.** Product ion spectra supporting the amino acid mutations shown in *Figure 6*.
DOI: https://doi.org/10.7554/eLife.45644.024
**Figure supplement 1.** Unio crassus - Hic74 coverage.
DOI: https://doi.org/10.7554/eLife.45644.014
**Figure supplement 2.** Unio pictorum - Hic74 coverage.
DOI: https://doi.org/10.7554/eLife.45644.015
**Figure supplement 3.** Margaritifera margaritifera - Hic74 coverage.
DOI: https://doi.org/10.7554/eLife.45644.016
**Figure supplement 4.** Pseudunio auricularius - Hic74 coverage.
DOI: https://doi.org/10.7554/eLife.45644.017
**Figure supplement 5.** PesB - Hic74 coverage.
DOI: https://doi.org/10.7554/eLife.45644.018
**Figure supplement 6.** HavA - Hic74 coverage.
DOI: https://doi.org/10.7554/eLife.45644.019
**Figure supplement 7.** HavB - Hic74 coverage.
DOI: https://doi.org/10.7554/eLife.45644.020
**Figure supplement 8.** HavC - Hic74 coverage.
DOI: https://doi.org/10.7554/eLife.45644.021
**Figure supplement 9.** HorB - Hic74 coverage.
DOI: https://doi.org/10.7554/eLife.45644.022
**Figure supplement 10.** HorC - Hic74 coverage.
DOI: https://doi.org/10.7554/eLife.45644.023

degradation and/or modification (Asp and Ser decomposition, amino acid racemization, *Appendix 1—figure 9*) but not high enough to induce mineral conversion, which starts to occur around 300 °C (*Yoshioka and Kitano, 1985*). Double-button PesB yielded low D/L values but only a modest number of peptides were identified (~100, much less compared to the Havnø samples, where at least 200–400 peptides were matched to known shell protein sequences); this suggests that the sample had not been diagenetically compromised (also supported by the amino acid data, *Appendix 1—figure 7*, *Appendix 1—figure 9*).

## Freshwater mother-of-pearl: archaeological significance

We found that mother-of-pearl of freshwater origin (Unionoida) was used in three European sites over a wide geographical range but relatively short time span (~4200–3800 BCE). Crucially, the crafters manufacturing such highly-standardized ornaments belong to different cultural groups: Late Mesolithic, Neolithic and Copper Age. Our results settle the 'marine vs freshwater' debate (*Heumüller, 2012*) for the double-buttons from Hornstaad (*Borrello and Girod, 2008*), and confirm previous identifications for the Peştera Ungurească examples (*Girod, 2010a*). The use of freshwater nacre (*Unio* or *Margaritifera*) comes as a surprise for the Havnø material, a coastal shell midden with a dominance of marine resource exploitation and rich in marine shells perfectly suitable for the purpose of making beads, including the horsemussel *M. modiolus*. Therefore, this finding suggests that the importance of freshwater mother-of-pearl be re-evaluated.

*Unio* sp. shells were probably selected to make 'disc beads' in the Epipaleolithic of the Levant, at Eynan (Natufian, 10,000–8,000 BCE; *Bar-Yosef Mayer, 2013*), and in Europe the presence of *Unio* sp. beads has been recorded 259 times according to the dataset gathered from the literature by *Rigaud et al. (2015)*, mainly from Neolithic sites. Despite this relative frequency of (presumed) freshwater mollusc ornaments in prehistoric Europe, a systematic study of their exploitation as raw materials is almost completely lacking. This is especially surprising since it is known that unionoid shells were exploited for mother-of-pearl until the Middle Ages (*Bertin, 2015*). Indeed, North American freshwater mussels were the basis for the 'pearl rush' during the 19th century, and their overexploitation for pearl harvesting, for making nuclei to be inserted in *Pinctada* pearl oysters as well as button-making on an industrial scale, almost drove a high number of species to extinction (*Haag, 2012*).

The lack of comprehensive archaeological studies on freshwater molluscs can be explained by two main factors. The first is methodological: the typical *chaîne opératoire* of bead-making involves several steps that obliterate most of the anatomical features (e.g. hinge apophysis) that are usable for taxonomic identification. These include: cutting and abrading small pieces until they take a circular shape; perforating the disc (*Gurova and Bonsall, 2017*) or, in the case of the double-buttons, working the side (with an abrasive wire?) to shape the central groove (*Bertin, 2015*; *Borrello and Girod, 2008*). Our work provides a series of analytical tools for overcoming this issue and for determining the biological origin of the raw material.

A second, perhaps more relevant, factor is the long-standing perception that freshwater molluscs are inherently less 'prestigious' than marine species, because of their presumed local origin. However, marine and freshwater molluscs are used side-by-side in a number of instances, for example the high-status burials at Mulhouse-Est (*Bonnardin, 2009*), or complex *parures* from the Swiss Early Bronze Age (*Borrello and Girod, 2008*). This clearly demonstrates that both were held in the same 'esteem' by the craftsman and that her/his choice was dictated by reasons other than the 'exoticism' of the material. The use of freshwater mother-of-pearl at Hornstaad and Peştera Ungurească, two sites with large procurement networks of exotic raw materials and at which there are clear signs of specialised production of ornaments (including gold at Peştera Ungurească; *Biagi and Voytek, 2006*; *Heumüller, 2012*), also confirms that freshwater pearl mussels were seen as prized materials, locally available. Furthermore, the use of freshwater molluscs for the manufacture of the *doppelknöpfe* recovered from Havnø (together with unworked fragments of the shells, *Appendix 1—figure 2*) shows that the manufacture of these ornaments was consistently associated with the use of freshwater mother-of-pearl, even in marine settings. Therefore, the Late Mesolithic people of Jutland and the Neolithic people of central Europe were either exchanging the finished products/raw materials, or the knowledge that the manufacture of the double-buttons required the use of unionoid shells.

## Why freshwater nacre?

It is clear that mother-of-pearl (nacre) from freshwater molluscs was a prime material of choice for the manufacture of shell double-buttons. Further investigation of other types of shell ornaments may reveal that this raw material was more frequently selected than previously thought, but in the meanwhile it is necessary to consider the reasons behind this choice.

Unionoids inhabit clean flowing waters (they are occasionally also found in lakes) and are dependent upon the presence of sufficient salmonid fish to carry the larval glochidial stage of the pearl mussel life cycle (*IUCN: International Union for Conservation of Nature, 2019* ). It is highly likely that freshwater mussels were collected near the site (as supported by the 'local' isotope signatures in this study), and that the procurement of the mussels was not especially difficult (for example, *M. margaritifera* lives at depths of up to two meters) nor too time-consuming. Therefore, the choice of this material must have been linked to reasons other than its long-distance provenance, the skills involved in procurement, or its rarity; rather, it is more likely a result of the characteristics of the raw material per se (mechanical properties and aesthetic qualities) and its connection to other *things*, be these in the sensory world (the river and its water, the landscape) or in the symbolic.

Mother-of-pearl is exceptionally hard - a thousand times more resistant to fractures than its mineral alone (*Currey, 1977*; *Jackson et al., 1990*) - and unionoid shells (*Unio* and *Margaritifera* sp., but not *Anodonta*) typically have a rather thick layer of nacre, while in some of the marine molluscs (particularly those occurring in European waters, such as *Modiolus* sp.), the ratio between nacre and prisms shifts, favouring the latter, where the nacre only partly covers the inner surface of the shell. The preservation of the prismatic matte layer may indicate that the coloured periostracum, which can give an appealing effect of chromatic contrast, was deliberately kept, for aesthetic reasons. Alternatively, if the periostracum was removed by mild abrasion, this would have resulted in fully white ornaments, showing both the brilliance of nacre and the dullness of prisms. The white colour of the ornaments may have been associated to wellbeing, peace and fertility (*Trubitt, 2003*). White was certainly a sought-after effect, so much so that red-purple *Spondylus* shells were often worked in order to remove the striking hue and reveal the white underneath (*Borrello and Micheli, 2011*). At the same time, the gloss of mother-of-pearl has been linked, in historical periods, with spirituality, life, royalty, and pearl fishing is a tradition that dates back to the same period considered here, around or a few centuries before 5500 BCE, in the Arabian Peninsula (*Charpentier et al., 2012*).

The choice of the raw material could also be a reflection of the role of freshwater environments: the Neolithic is the period in which water, together with plants and animals, is 'domesticated' (*Garfinkel et al., 2006*; *Mithen, 2010*). Rivers provided fast access routes to Central, Western and Northern Europe for hunter-gatherers during the Palaeolithic and Mesolithic and, later on, for agriculturalists coming from the East (*Rowley-Conwy, 2011*). Despite their 'fluidity', rivers and lakes were meaningful and persistent places in the prehistoric landscape.

In summary, the streams, rivers and lakes near occupation sites were inhabited by organisms that provided the crafters with exceptional-quality raw material, easy to procure and which could be worked following a well-established *chaîne opératoire* in order to obtain a standardized result. The small white double-buttons could then be threaded using the central groove or pressed into the fabric or leather (*Kannegaard, 2013*). Our work thus highlights an interpretative bias whereby exoticism is considered the primary reason for choice of raw materials, and suggest that local environments held an equally important place in the mind of prehistoric people.

## Conclusions

The first application of 'palaeoshellomics' has demonstrated that it is possible to recover and identify ancient proteins sequences from mollusc shell, despite significant analytical challenges due to the combined effects of several factors, including low protein concentrations, small samples sizes, diagenesis and database insufficiency (*Table 2*).

Our molecular data showed that molluscan proteins are similar across the four freshwater taxa we examined (*U. pictorum*, *U. crassus*, *M. margaritifera*, *P. auricularius*) and differ significantly from the two marine species (*O. edulis*, *M. modiolus*; *Table 2*, *Figures 4* and *5*). We confirmed that freshwater molluscan matrix proteins are characterized by highly repetitive low complexity domains (RLCs). This is consistent with results obtained on other shell taxa, and improves our understanding of the biomineralization mechanisms within these invertebrate systems.

The archaeological double-buttons examined here were all confidently identified as Unionoida, freshwater shells with a thick layer of mother-of-pearl, using a combination of mineralogical, geochemical and biochemical techniques (SEM, FTIR-ATR, oxygen and carbon isotopes, chiral amino acid analyses, palaeoshellomics). The analysis of the sequence of the shell matrix protein Hic74 supports the use of *Unio* or *Margaritifera* as the raw material for the three Havnø ornaments (excluding *Pseudunio*), but lack of coverage of most amino acid modification sites in the reference samples hampered identification to a lower taxonomic level (*Figure 6*, *Table 3*).

The high degree of standardization of the ornaments (*Figures 1* and *2*), as well as the consistent choice of freshwater mother-of-pearl as raw material indicate that, in Europe, between ~ 4200 and~3800 BCE, there was a common notion of the manufacture of the *doppelknöpfe,* which was shared by different cultural groups: Late Mesolithic (Ertebølle), early Late Neolithic (Hornstaad Group), and Copper Age (Toarte Pastilate/Coţofeni). Our in-depth study therefore puts into question the most commonly accepted interpretations, which privilege the preponderant use of exotic marine shells as prestigious raw materials for the manufacture of prehistoric shell ornaments.

## Methods

### Non-destructive characterization

Whole beads and fragments of the reference shells were observed using an environmental Scanning Electron Microscope (Hitachi TM1000 Tabletop Microscope). The mineralogy of the beads was identified by infrared spectroscopy in attenuated total reflectance (ATR) mode (FTIR-ATR) (Appendix 1, sections 3.1 and 3.2).

### Biogenic carbonate isotopic analyses

Isotopic analysis was carried out on biogenic carbonate to obtain bulk $\delta^{13}C$ and $\delta^{18}O$ values for the double-buttons. Small amounts of cleaned samples (bleached using concentrated NaOCl (12% w/v) for 48 hr) were analysed using a Delta V Plus mass spectrometer coupled with a Kiel IV carbonate device (ThermoFisher). All steps are detailed in Appendix 1, section 3.3.

**Protein analysis**

All reference shells and beads were powdered using a clean mortar and pestle and accurately weighed.

## Amino acid racemization

~2 mg of powder were selected for each double-button and were bleached for 48 hr using concentrated NaOCl (12% w/v) in order to isolate the intra-crystalline amino acids. The analysis of total hydrolysable amino acids (THAA) was carried out as detailed in *Demarchi et al. (2014)* and Appendix 1 (section 3.4).

## Proteomics (Appendix 1, sections 3.5, 3.6)

Powdered reference shells and double-buttons were bleached using diluted NaOCl (2.6%) for 48 and 3 hr respectively. Demineralization was carried out using cold diluted acetic acid, the resulting solutions were thoroughly desalted, concentrated and lyophilized. All samples were digested using two proteolytic enzymes (trypsin and elastase) in order to maximise sequence coverage. LC-MS/MS analyses were carried out using a nanoflow HPLC instrument (U3000 RSLC Thermo Fisher Scientific) coupled to a Q Exactive Plus mass spectrometer (Thermo Fisher Scientific) for *M. modiolus*, *O. edulis*, *U. pictorum*, *M. margaritifera* and the Havnø and Hornstaad-Hörnle IA ornaments (MSAP CNRS laboratory, University of Lille); an Ultimate 3000 Dionex nanoHPLC instrument coupled with an Orbitrap Fusion (Thermo Fisher Scientific) mass analyzer was used for the analysis of *U. crassus, P. auricularius* and for sample PesB (Mass Spectrometry Biomolecules core facility, University of Turin). No systematic difference was detected between the data obtained at the two facilities, which could have potentially affected the identification of the samples.

Bioinformatic analysis was carried out using PEAKS Studio 8.5 (Bioinformatics Solutions Inc, *Ma et al., 2003*). The thresholds for peptide and protein identification were set as follows: protein false discovery rate (FDR) = 0.5%, protein score $-10lgP \geq 40$, unique peptides $\geq 2$, de novo sequences scores (ALC%) $\geq$ 50. The FDR is calculated by the software PEAKS using an approach called decoy fusion (*Zhang et al., 2012*), whereby target and decoy are concatenated for each protein, rather than searching a target database and a decoy database separately (which can result in FDR underestimation); the effect is that of improving accuracy without impacting on sensitivity. The de novo algorithm derives the peptide sequences from the tandem mass spectra without using a database, and it is therefore suitable for the study of organisms where molecular reference sequences are scarce, such as in the present case. Each amino acid residue in the sequence is given a score (0–99%), indicating how confident the software is for that *local* identification. The overall confidence of the de novo sequence is calculated as the Average of Local Confidence (ALC) score.

The Molluscan Protein Database used in this study comprised 633061 protein sequences, that is all sequences available on the National Centre for Biotechnology Information (NCBI) repository restricting the taxonomy to Mollusca (fasta database downloaded on 15/02/2018), excluding all common contaminants (cRAP; common Repository of Adventitious Proteins: http://www.thegpm.org/crap/). The ESTs database included 1149,723 expressed sequence tags, also restricting the taxonomy to Mollusca (fasta database downloaded on 15/02/2018) and including cRAP sequences. Unionoida molluscs are poorly represented in these databases: a search for Unionoidae on the NCBI Identical Protein Database retrieves 4562 entries, almost exclusively belonging to soft tissue proteins, for example cytochrome oxidase, NADH dehydrogenase and ATP synthase. Shell matrix-related sequences are few (<35), almost all from the transcriptome of the triangle sail mussel, *Hyriopsis cumingii* (first released in 2013 [*Bai et al., 2013*]), including proteins related to shell biomineralization (i.e. upsalin, Pif, calmodulin, hicsilin, hichin, Hic74, Hic52, nasilin, Ca-binding P-glycoprotein, perlucin). The same search for Margaritiferidae yields 207 entries, none of which related to shell matrix proteins.

**Data deposition**

All the mass spectrometry proteomics data have been deposited in the ProteomeXchange Consortium (http://proteomecentral.proteomexchange.org) via the PRIDE partner repository (*Vizcaíno et al., 2013*) with the data set identifier PXD011985.

## Acknowledgements

The authors are sincerely grateful to Peter Rowley-Conwy and two anonymous reviewers, as well as the Reviewing and Senior Editors, for exceptionally clear, valuable and insightful comments on this manuscript.

BD and JS would like to thank Rosa Boano, Cristina Giacoma, Massimo Maffei, Nadia Barbero and their teams at the University of Turin. JS and FM would like to thank Justine Briard (uB-FC) for help with sample preparation and Emmanuel Fara (uB-FC) for valuable insights and comments. BD, FM and JS are grateful to Matthew Collins (Universities of Copenhagen and Cambridge) and Harry Robson (University of York) for useful discussion.

BD is supported by the Italian 'Ministero dell'Istruzione dell'Università e della Ricerca' (Programma per Giovani Ricercatori 'Rita Levi Montalcini') and wishes to acknowledge the Department of Archaeology at the University of York (UK) and the EU FP7 Re(In)tegration grant PERG07-GA-2010–268429 (project: mAARiTIME) for funding pilot research.

JS, BD and FM are also supported by the PHC Galilée programme, Italo-French University (UIF/UFI; project G18−464/39612 SB) and JS acknowledges the support of MIUR and of a BGF funded by the French embassy in Italy.

The work performed at UMR CNRS 6282 Biogeosciences (JS, BD, FM), Dijon was financed via the annual recurrent budget of this research unit.

KP acknowledges the support of the Leverhulme Trust, PLP-2012–116.

LP thanks the CNRS (France) and the University of Bourgogne Franche-Comté (France).

GM is supported by 'Ricerca Locale Università di Torino'.

CT would like to thank Stéphanie Devassine, Fabrice Bray and Christian Rolando (MSAP, University of Lille) for their technical contribution for MS operation. CT also thanks the Institut Universitaire de France, the European community (FEDER), the Région Hauts de France, the IBISA network, the CNRS, the University of Lille and the FR 3624 FT-ICR.

## Additional information

### Funding

| Funder | Grant reference number | Author |
| --- | --- | --- |
| Ministry of Education, Universities and Research | Young Researchers | Beatrice Demarchi |
| European Commission | PERG-GA-2010-268429 | Kirsty EH Penkman<br>Beatrice Demarchi |
| Leverhulme Trust | | Kirsty EH Penkman |
| Centre National de la Recherche Scientifique | | Théophile Cocquerez<br>Laurent Plasseraud<br>Caroline Tokarski<br>Jérôme Thomas<br>Frédéric Marin |
| Campus France, Università Italo-Francese | PHC Galilée programme | Jorune Sakalauskaite<br>Frédéric Marin<br>Beatrice Demarchi |

The funders had no role in study design, data collection and interpretation, or the decision to submit the work for publication.

### Author contributions

Jorune Sakalauskaite, Resources, Data curation, Software, Formal analysis, Validation, Visualization, Methodology, Writing—original draft; Søren H Andersen, Paolo Biagi, Maria A Borrello, Alberto Girod, Resources, Writing—review and editing; Théophile Cocquerez, Laurent Plasseraud, Formal analysis, Investigation; André Carlo Colonese, Formal analysis, Writing—original draft; Federica Dal Bello, Sheila Taylor, Investigation; Marion Heumüller, Helmut Schlichtherle, Resources, Writing—original draft; Hannah Koon, Investigation, Writing—review and editing; Giorgia Mandili, Claudio Medana, Jérôme Thomas, Resources; Kirsty EH Penkman, Resources, Investigation, Writing—review

and editing; Caroline Tokarski, Resources, Formal analysis, Methodology, Writing—review and editing; Julie Wilson, Software, Formal analysis, Validation, Visualization, Writing—original draft; Frédéric Marin, Conceptualization, Resources, Formal analysis, Supervision, Validation, Investigation, Methodology, Writing—original draft; Beatrice Demarchi, Conceptualization, Resources, Data curation, Formal analysis, Supervision, Funding acquisition, Validation, Investigation, Visualization, Methodology, Writing—original draft, Project administration

### Author ORCIDs
Jorune Sakalauskaite (iD) https://orcid.org/0000-0002-8029-8120
Federica Dal Bello (iD) http://orcid.org/0000-0003-0726-3025
Kirsty EH Penkman (iD) https://orcid.org/0000-0002-6226-9799
Jérôme Thomas (iD) http://orcid.org/0000-0002-1602-4416
Beatrice Demarchi (iD) https://orcid.org/0000-0002-8398-4409

### Decision letter and Author response
Decision letter https://doi.org/10.7554/eLife.45644.061
Author response https://doi.org/10.7554/eLife.45644.062

## Additional files

### Supplementary files
• Transparent reporting form
DOI: https://doi.org/10.7554/eLife.45644.026

### Data availability
All data generated or analysed during this study are included in the manuscript and supporting files. Source data files have been provided for Figures 4 and 5. All the mass spectrometry proteomics data have been deposited in the ProteomeXchange Consortium (http://proteomecentral.proteomexchange.org) via the PRIDE partner repository with the data set identifier PXD011985.

The following dataset was generated:

| Author(s) | Year | Dataset title | Dataset URL | Database and Identifier |
|---|---|---|---|---|
| Demarchi B | 2019 | Palaeoshellomics | https://www.ebi.ac.uk/pride/archive/projects/PXD011985 | ProteomeXchange, PXD011985 |

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

## Appendix 1

DOI: https://doi.org/10.7554/eLife.45644.027

### 1. Archaeological sites and double-buttons

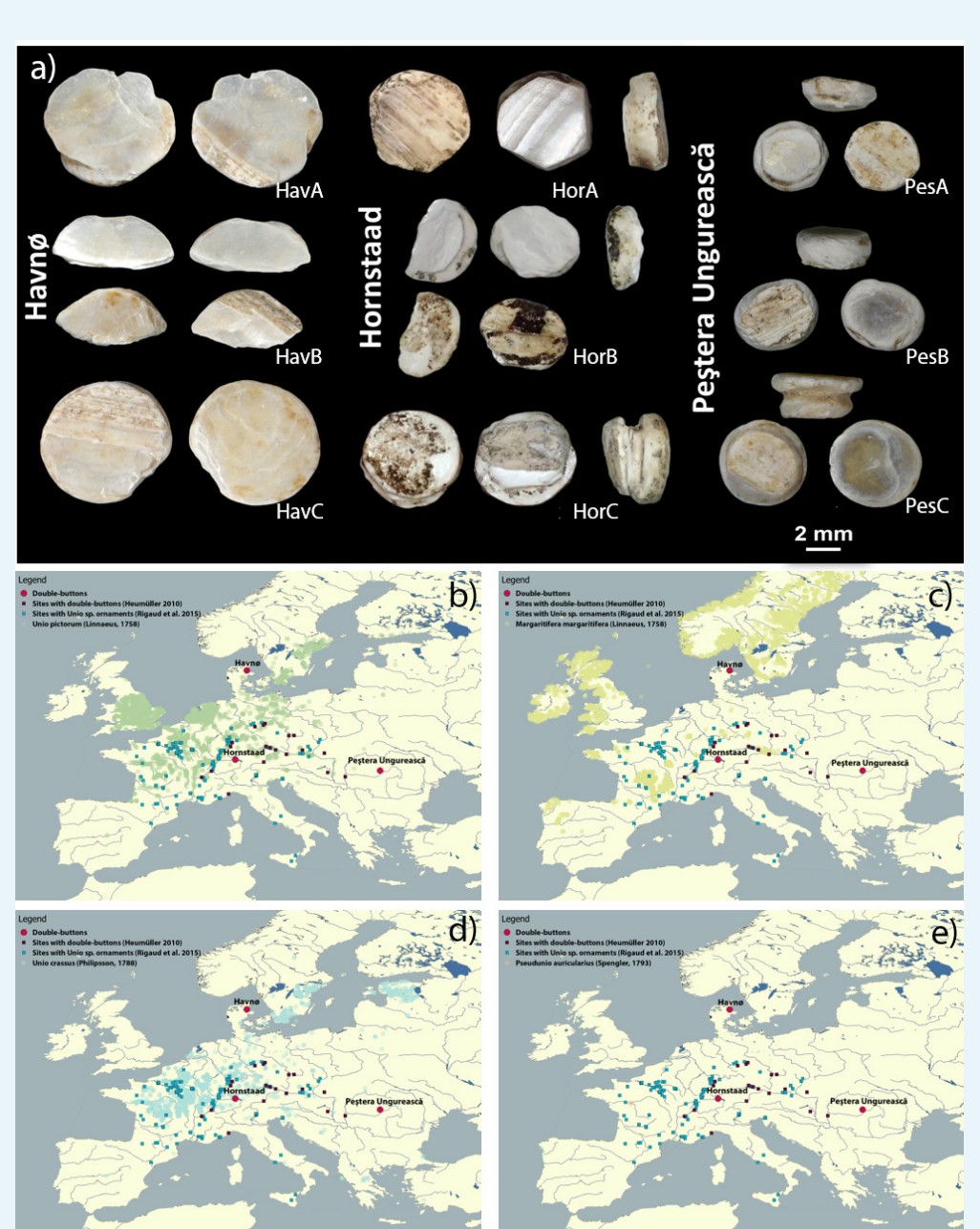

**Appendix 1—figure 1.** (a) Double-button samples from the archaeological sites of Havnø (Denmark), Hornstaad-Hörnle IA (Germany) and Peştera Ungurească (Romania). Findings of double-buttons and of *Unio* ornaments as reported in the literature, compared to the present occurrence of *Unio pictorum* (b), *Margaritifera margaritifera* (c), *Unio crassus* (d), *Pseudunio auricularius* (e) (data obtained from GBIF, the Global Biodiversity Information Facility, *GBIF.org, 2018*). The three sets of double-buttons (*Doppelknöpfe*) analysed here come from the archaeological sites of Havnø (Denmark), Hornstaad-Hörnle IA (Germany) and Peştera Ungurească (Romania) and approximately span the period between 4200 and 3800 BCE.

DOI: https://doi.org/10.7554/eLife.45644.028

## 1.1. Havnø

Havnø is a 'stratified' shell midden, spanning the Late Mesolithic/Early Neolithic transition, that is Ertebølle and Funnel Beaker cultures, dated to 3950 cal BC. The site is on the East coast of Jutland c. 80 km north of Århus. The Gudenå river, which originated at the end of the Ice Age, a prime spot for fishing salmons and the major river in Denmark, has its estuary on the Randers Fjord, some 20 km to the south of Havnø.

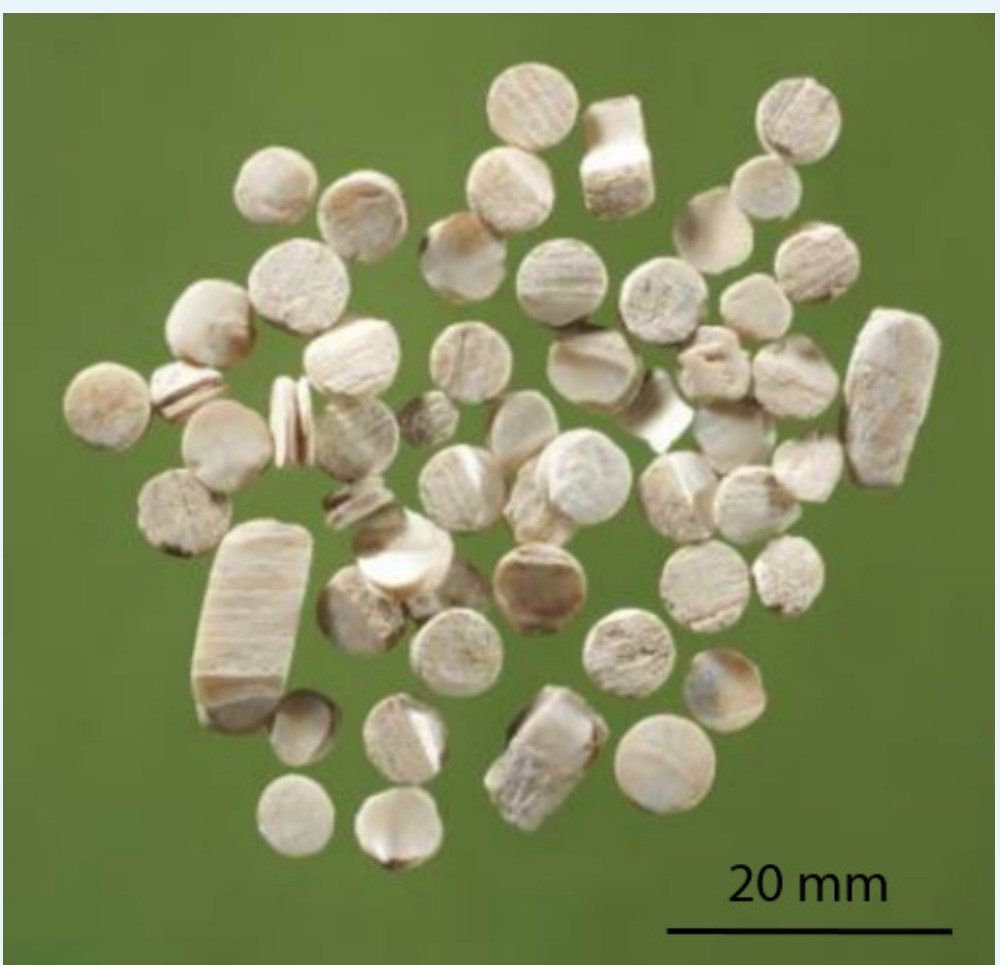

**Appendix 1—figure 2.** The double-buttons from Havnø.
DOI: https://doi.org/10.7554/eLife.45644.029

Now situated on the coast, Havnø was a small island during prehistoric times, covered in primeval forests of oak, elm, hazel and birch and an excellent base for fishing and marine hunting, as well as fowling and shell gathering (*Andersen, 2008*). The midden itself is 100-m-long and 25–27-m wide and was used for around 1300 years, demonstrating that this was a significant place during the Late Mesolithic and Early Neolithic. Indeed, among the abundant shells (oyster, cockle, mussel and periwinkle), one can find a variety of food remains as well as tools and ceramic vessels and traces of human settlements. Around 40 double-buttons were recovered from the Late Mesolithic horizon, dated to c. 4200–4000 cal BC (*Andersen, 2008*; *Andersen, 2000*). The buttons have a circular or oblong shape, one shiny surface and one matte, and a groove along the edge. Their diameters range between 0.69–0.79 cm. The working hypothesis has been that the raw material of the double-buttons was a marine shell:

*Modiolus modiolus*, *Arctica islandica* or (according to **Rowley-Conwy, 2014**), *Ostrea edulis*, given that these are the most abundant species in the midden and that the site reflects the great importance of marine environments for the Ertebølle people. Interestingly, eight double-buttons, similar (but smaller, only around 5 mm in diameter) to the ones from Havnø and from the same period, have been found at the Nederst site, on the Djursland peninsula, 50 km south of Havnø (**Kannegaard, 2013**).

## 1.2. Hornstaad-Hörnle IA

Hornstaad-Hörnle IA is a pile-dwelling Neolithic site on Lake Konstanz, Baden-Württemberg, present-day Southern Germany. It is part of the UNESCO World Heritage 'Prehistoric Pile Dwellings around the Alps'. Hornstaad is the typesite of the Hornstaad Group, an early cultural unit of the regional Late Neolithic (Jungneolithikum). The site contained several phases of wooden houses construction and large assemblages of artefacts and biofacts (**Dieckmann et al., 2016**; **Dieckmann et al., 2006**; **Schlichtherle, 1990**). The economic basis of the village, which comprised 40–80 houses, was a combination of agriculture and animal husbandry, but hunting, fishing and the collection of wild plants also played an important role. The site has yielded extraordinarily rich assemblages of ornaments, including the (approximately) 564 double-buttons (diameter 0.46–0.58 cm) studied here, which come from an occupation level dated by dendrochronology to 3917–3902 BC (**Heumüller, 2012**; **Heumüller, 2009**). Nearly all ornaments have been affected by the fire that destroyed the settlement in 3009 BC, and were found fragmented into pieces. The identification of the double-buttons from Hornstaad-Hörnle IA has been debated (**Heumüller, 2012**): macro- and micro-morphological observations resulted in the identification of the marine shell *Ostrea edulis* (edible oyster, also called European flat oyster) by one expert (H.-J. Niederhöfer) and of the freshwater pearl mussel *Margaritifera margaritifera* by another malacologist (A. Girod) (**Borrello and Girod, 2008**).

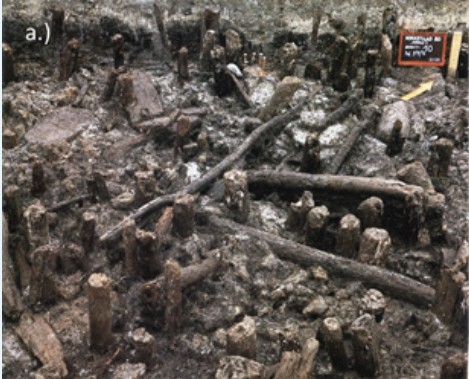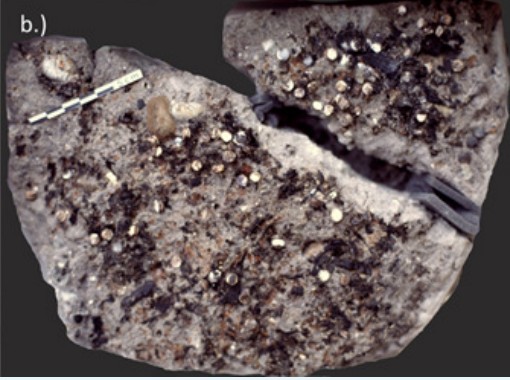

**Appendix 1—figure 3.** Excavations at Hornstaad D-Hörnle site (**a**) and the discovery of the double-buttons (**b**).
DOI: https://doi.org/10.7554/eLife.45644.030

## 1.3. Peştera Ungurească

Peştera Ungurească is one of the cave sites in the karstic Cheile Turzii Gorge, approximately nine kilometres west of Turda in central Transylvania, which was formed by the Hăşdate river. The river currently flows at the bottom of the gorge, some 100 m below the cave (**Biagi and Voytek, 2006**). The archaeological sequence spans the transition between the Middle Neolithic and the Bronze Age (Cheile Turzii-/Lumea Nouă-Iclod and Coţofeni).

The sediments from layers attributed to the Toarte Pastilate culture and the transition to Coţofeni were wet-sieved with a 1-mm mesh size and yielded a wealth of information on the environment and the subsistence strategies: hunting and rearing of large animals was supplemented by fishing and fowling, and both foraging and agriculture were practiced

(*Bartolomei, 2013*; *Biagi and Voytek, 2006*; *Boschian, 2010*; *Nisbet, 2010*). Of interest, the finding of large fish vertebrae and freshwater turtles suggested the exploitation of a large river, probably not the Hăşdate but the larger Arieşul, some 2.5 km east. The analysis of the knapped stone (chert and obsidian) assemblage revealed that the inhabitants of Peştera Ungurească sourced raw materials from western Hungary, north-western Ukraine and the Carpathians (*Biagi and Voytek, 2006*). Layer 2b (charcoal radiocarbon date GrN-29100: 5100 ± 40 BP, 3980–3790 cal BC (2σ) [*Biagi and Voytek, 2006*]) yielded a kiln that was used for gold smelting, testified by the recovery of gold beads and gold platelets. This is interesting because, in other sites in the same area, a number of jewellery workshops have also been unearthed, which are attributed to Coţofeni culture and which yielded over 30 beads, made of bone, marble, shells (including *Spondylus*), as well as copper tools and small fragments of malachite (*Lazarovici and Lazarovici, 2016*).

In Peştera Ungurească, the small (diameter 0.41–0.52 cm) shell double-buttons were recovered from the Late Neolithic and transitional (Toarte Pastilate/Coţofeni) layers (2b and 1a.2) (*Lazarovici and Lazarovici, 2013*). The semi-worked double-button (PesB) selected for this study comes from layer 1a.2. The raw material had been identified as *Unio* sp., also because many broken *Unio crassus* valves (possibly associated with alimentation or tool-making) had been recovered throughout the sequence (*Girod, 2010a*).

## 2. Ecology of the molluscan taxa investigated

Molluscs represent the second most diverse animal phylum in terms of numbers of described living species (*Lydeard et al., 2004*), however this global estimate may vary widely due to different synonym taxa names being used, the likelihood that there are many species not yet described from regions that are still unexplored, as well as issues in reconciling taxonomic classification based on morphology and molecular phylogenies. Molluscan 'skeletons' have fascinated natural scientists for decades; in particular, bivalve shells have been fairly well investigated with regard to their microstructure, mineralogy and physicochemical characteristics (*Boggild, 1930*; *Carter, 1990*; *Taylor et al., 1973*; *Taylor et al., 1969*). For example, mother-of-pearl (nacre) is one of the best-known and most widely studied shell microstructures. Nacre is present in four molluscan classes, including bivalves, gastropods, cephalopods and monoplacophorans. Among bivalves, nacre has a patchy distribution and is present in the following clades (orders): Nuculida, Mytilida (commonly known as true mussels, including *Modiolus*), Pterioida (which comprises, among others, the well-known and commercially-relevant pearl oyster *Pinctada* and the Mediterranean fan mussel *Pinna nobilis*), Trigoniida and Anomalodesmata. All these orders include exclusively marine animals. In addition, nacre is present in order Unionoida (freshwater bivalves). In bivalves, most of the nacreous microstructures belong to the 'brickwall' type (sheet nacre [*Taylor et al., 1969*]).

Six different mollusc taxa have been selected for morphological and biomolecular comparative analysis for this study, choosing shells with structural features alike to archaeological ornaments and that are native to the local environment of the sites:

- two marine bivalves - *Ostrea edulis* and *Modiolus modiolus,* common in northwestern European seas (also, oysters dominate the midden at Havnø).
- four freshwater bivalves belonging to order Unionida (*Bolotov et al., 2016*):
○ *Unio pictorum* (also known as painter's mussel) and *Unio crassus* (both belonging to genus *Unio*, family Unionidae).
○ *Margaritifera margaritifera* (also known as freshwater pearl mussel) and *Pseudunio auricularius* (both belonging to family Margaritiferidae);

Specimens of *U. crassus* and *P. auricularius* are archaeological (sub-fossil) shells from the sites of Peştera Ungurească and of Isorella (a Neolithic site in Italy [*Girod, 2010b*]).

### 2.1. Ecology of the molluscan taxa investigated

Unionoids inhabit clean flowing waters (*Bolotov et al., 2016*) and freshwater molluscs account for a great portion of the total number of mollusc species facing extinction: a significantly higher proportion of freshwater molluscs is present in the IUCN Red List of Threatened

Species than marine species. This is due to habitat degradation after the construction of dams, stream channelization, pollution and sediment toxicity, biological invasions and other human-driven impacts (**Bolotov et al., 2016**; **Lopes-Lima et al., 2017**). All the unionoid shells studied in this research are present in the Red List of threatened species (http://www.iucnredlist.org) with *Unio crassu*s being in the endangered category and *Margaritifera* species at the point of critically endangered.

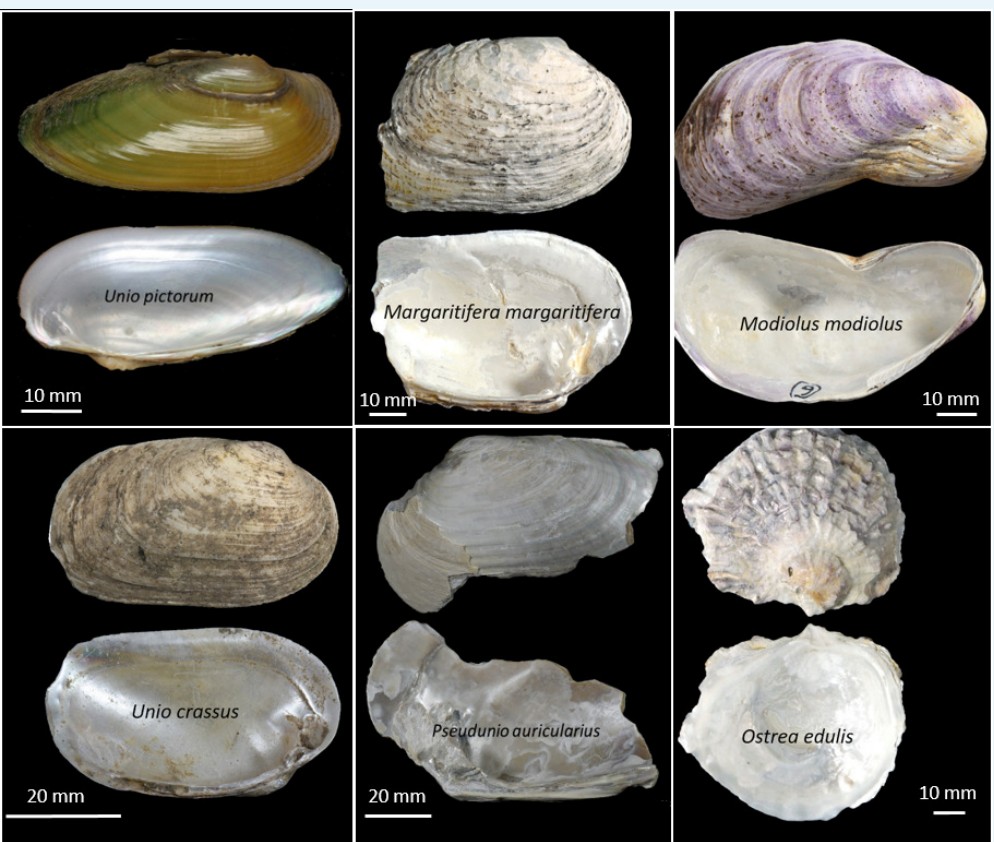

**Appendix 1—figure 4.** Marine and freshwater shells included in this study for comparative analysis: possible sources of raw material used for the manufacture of the double-buttons.
DOI: https://doi.org/10.7554/eLife.45644.031

*Ostrea edulis* (the European flat oyster) has a natural geographical distribution ranging along the European Atlantic coast, from Norway to Morocco, and all along the Mediterranean as well as the Black Sea (**Launey et al., 2002**). This species is also becoming a concern to ecologists since oyster reefs and beds are, globally, one of the most endangered types of habitat. *Modiolus modiolus* (commonly named as horse mussel) is an Arctic-boreal species that is limited in distribution by warmer temperatures to the south, but occasionally specimens have been reported as far South as North West Africa.

## 3. Analytical methods and detailed results

The three sets of double-buttons were analysed using an integrated analytical approach across different facilities. Morphological analysis (SEM) was carried out at the Biogéosciences facility (mixed research unit 6282, CNRS/uB-FC, France); mineral phase characterization (FTIR-ATR) was obtained at the Department of Chemistry of the University Burgundy Franche-Comté (France). Isotope analyses of the Havnø and Hornstaad double-buttons were carried out the Light Stable Isotope Laboratory (School of Archaeological Sciences, University of Bradford, UK) and for Peştera Ungurească sample at the GISMO facility at the department of Biogéosciences (mixed research unit 6282, CNRS/uB-FC, France). The AAR data were

obtained at Department of Chemistry (University of York, UK). Sample preparation for proteomic analysis was carried out at CNRS/uB 6282 Biogéosciences facility (uB-FC) and Department of Life Sciences and Systems Biology (University of Turin, Italy). LC-MS/MS analyses were performed at the MSAP CNRS laboratory, University of Lille (France) and the Mass Spectrometry Biomolecules core facility, Molecular Biotechnology and Health Sciences Department, University of Turin (Italy).

## 3.1. Scanning Electron Microscopy

### 3.1.1. Shell morphology

The study of the morphology of shell microstructures is one of the most common approaches for taxonomic classification and can be applied also to the identification of archaeological samples. Furthermore, the mineralogical investigation of the shell layers, which can occur as either aragonite or calcite (polymorphs of calcium carbonate), can also help to distinguish different shell taxa.

- Unionoids are made of two fully mineralized layers, both aragonitic. The outer mineralized layer is composed of prisms that develop perpendicularly to the outer shell surface. The inner layer is nacreous, comprising extremely thin flat tablets that are superimposed in the typical 'brickwall microstructure' arrangement, which is common in most nacreous bivalves.
- *Modiolus modiolus* shells have a finely prismatic calcitic outer layer and an aragonitic nacreous inner layer, the latter with alternating sheets of nacre and aragonitic myostracal prisms (*Taylor et al., 1969*).
- *Ostrea edulis* has a thin calcitic outer prismatic layer of regular and simple prisms and an inner layer of very fine folia (the 'foliated calcite') parallel to the plane of the shell, except for the myostraca deposits, which are aragonitic (*Carter, 1990*). Discontinuous chalky layers are also observed. According to Bøggild, no trace of aragonite can be found in either recent or fossil Ostreidae (*Boggild, 1930*), but further research showed that oysters possess a small fraction of aragonite in their myostracum and inner ligament (*Taylor et al., 1969*).

### 3.1.2. Analytical procedure

The microstructure of the archaeological double-buttons and the six reference shells was determined by means of SEM (Hitachi TM1000 Tabletop Microscope in low vacuum mode). Prior to the analysis, the samples were etched with 1% (w/v) EDTA solution in a sonication bath, for two-three minutes for mollusc shells and up to one minute for archaeological ornaments, in order to provide better resolution of the topography of the surface. No carbon coating was applied.

### 3.1.3. Results

All the double-buttons are clearly made of mollusc shell: nacre tablets and prisms are visible at both high and low magnification (*Appendix 1—figure 5 a-c*). All of them display a very similar nacro-prismatic microstructure, consisting of thick nacre platelets (around 1 μm) and long prisms (around 150 μm). The overall microstructure is similar that of unionoid shells (*Appendix 1—figure 5, d*) and differs slightly from nacreous layer of *Modiolus modiolus* shell in terms of prism elongation. It does not resemble at all that of the marine oyster *Ostrea edulis* (*Appendix 1—figure 5, e*). All of the double-buttons, and especially the Hornstaad set (*Appendix 1—figure 5, b*), display clear signs of diagenetic alterations, resulting in the loss of connection between nacre tablets and the adjacent prisms.

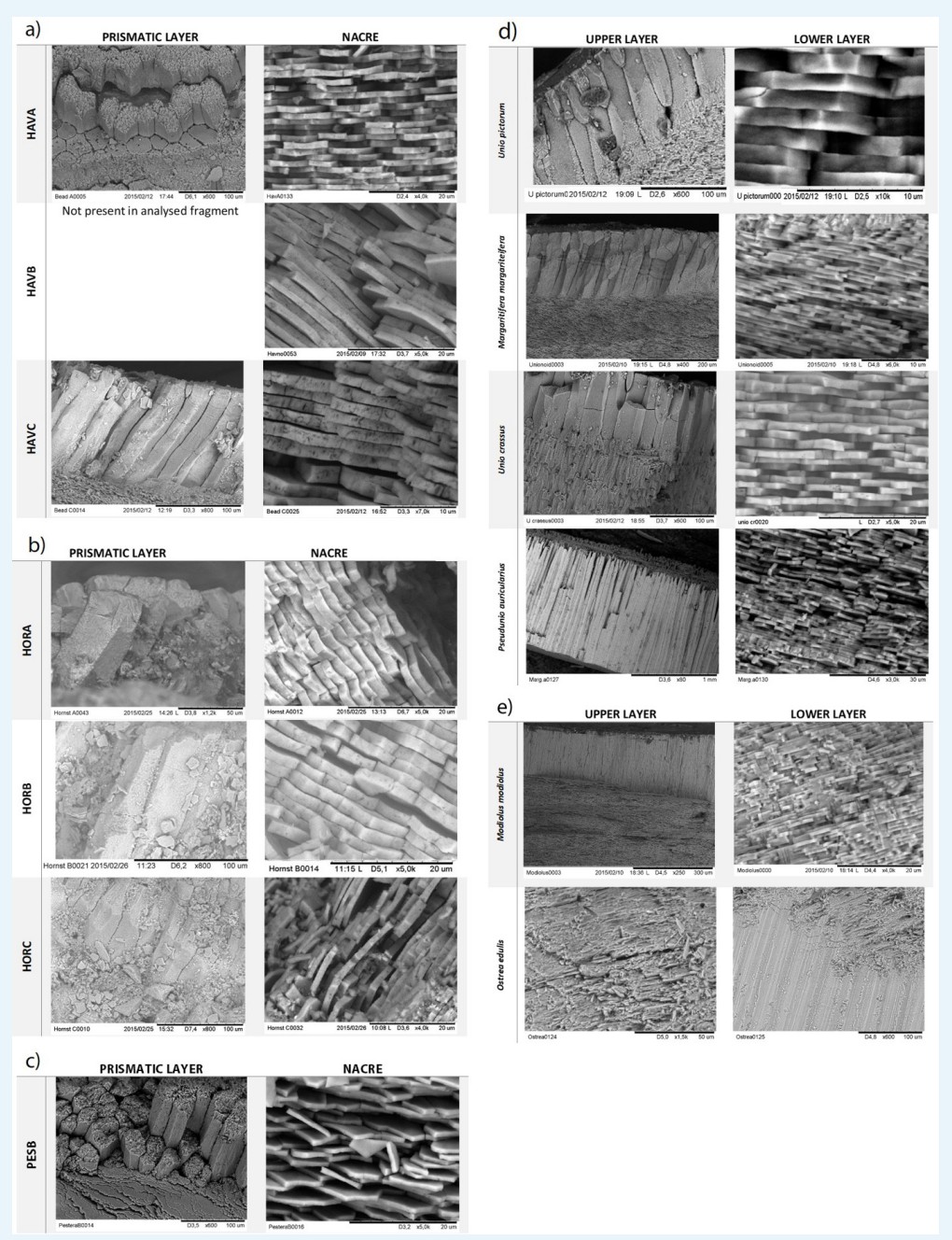

**Appendix 1—figure 5.** SEM microstructural analysis of archaeological double buttons (**a–c**) and mollusc shells (**d,e**). Double-buttons: a) Havnø (HavA, HavB, HavC), b) Hornstaad (HorA, HorB, HorC), c) Peştera Ungurească (PesB). Reference shells: d) freshwater unionoid shells (modern *Unio pictorum*, *Margaritifera margaritifera* and sub-fossil *Unio crassus*, *Pseudunio auricularius*), e) marine shells (*Modiolus modiolus* and *Ostrea edulis*).

DOI: https://doi.org/10.7554/eLife.45644.032

## 3.2. Infrared Spectroscopy

The identification of the mineral phases present in the archaeological double-buttons was obtained by Fourier Transform Infrared spectroscopy in attenuated total reflectance mode (FTIR-ATR). ATR spectra were recorded from small grains of material, delicately taken from the ornaments with a scalpel. The spectra were acquired with a Bruker Vector 22 instrument (BrukerOptics Sarl, France, Marne la Vallée) fitted with a GoldenGate attenuated

total reflectance (ATR) device (SpecacLtd, Orpington, UK) in the 4000–500 cm$^{-1}$ range (twelve scans at a spectral resolution of 4 cm$^{-1}$). Spectral analyses were performed with the OPUS software provided by the instrument manufacturer (BrukerOptics Sarl). The assignment of the different absorption bands was obtained by comparison with previous spectra descriptions available in the bibliography (*Henry et al., 2017*). Each double-button was sampled in 'bulk' but the nacreous and prismatic layers of samples HavC and PesB were also analysed separately, in order to determine the mineralogical composition of each microstructural layer.

### 3.2.1. Results

The analysis of the nacre and prismatic layers from samples HavC and PesB showed the presence of distinctive aragonite absorption bands (*Appendix 1—figure 6, a,b*): a doublet at around 712 and 700 cm$^{-1}$ (v4, CO in-plane bending); a mid-size peak positioned at 853 cm$^{-1}$ (v2, out-of-plane bending) and a weak peak at 1082 cm$^{-1}$ (v1, symmetrical stretching). Calcite has only a single peak of v4 at 711 cm$^{-1}$, its v2 peak is shifted to 870 cm$^{-1}$ and shows absence of the v1 stretching at 1082 cm$^{-1}$ (*Loftus et al., 2015*) - this is not observed in our samples, as can be seen comparing the double-button spectra to those of the purely-calcitic prisms of *Pinna nobilis* (*Appendix 1—figure 6, c*). The analysis confirms that the double-buttons are composed of aragonitic nacreous sheets with a thin aragonitic prismatic layer on top. The combination of these two aragonitic microstructures is common in unionoid bivalves.

Considering that aragonite is a less stable form than calcite and under natural diagenesis or strong heating biogenic aragonite re-crystallizes to calcite, infrared spectra can be used to assess the extent of degradation of the archaeological samples. It is particularly interesting for the Hornstaad double-buttons, as they have been exposed to heat (burning). FTIR-ATR spectra (*Appendix 1—figure 6, d*) from bulk samples obtained on each double-button show no presence of secondary calcite and thus no form of recrystallization - only the doublet (v4, CO in-plane bending) characteristic of aragonite is present.

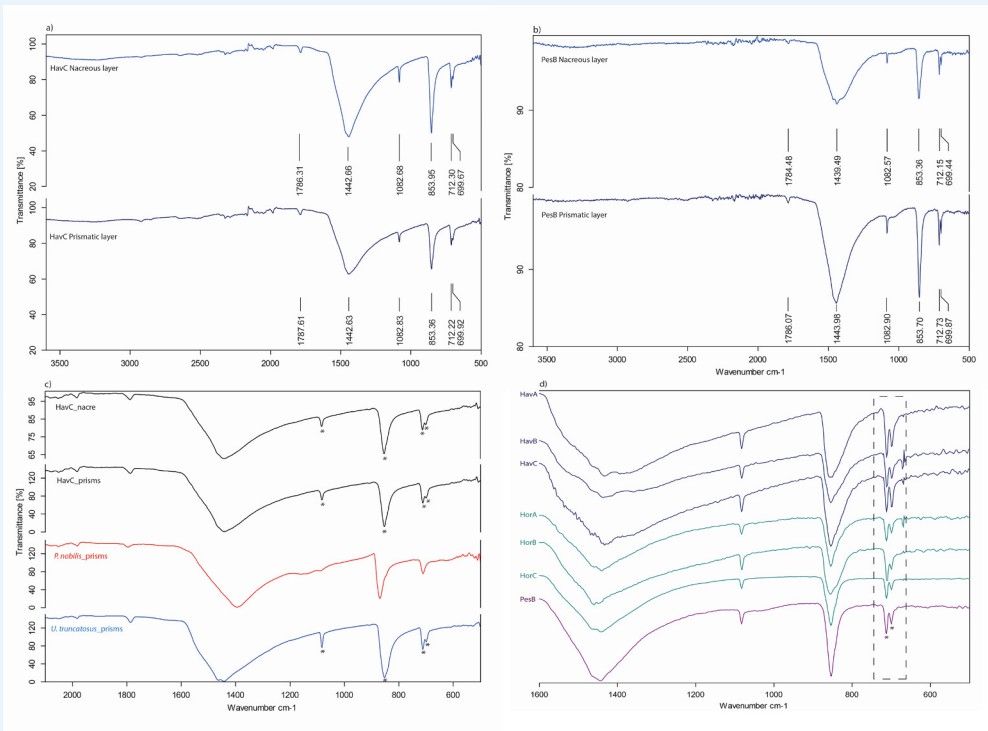

**Appendix 1—figure 6.** FTIR-ATR spectra of the double-buttons. Asterisks mark aragonite marker absorption bands. Nacreous and prismatic layers of (**a**) HavC, (**b**) PesB; (**c**) FTIR-ATR spectra comparison between HavC nacre and prismatic layers (black) with calcitic prismatic layer of *Pinna nobilis* (red) and aragonitic *Unio truncatosus* (blue), confirming the fully

aragonitic mineralogy of both layers; (**d**) FTIR-ATR spectra of all the double-buttons, sampled in 'bulk': the presence of doublets (CO in-plane bending mode) at $\sim$ 712 and$\sim$700 cm$^{-1}$ (dashed line) in all samples indicates absence of recrystallization of the biogenic carbonate.
DOI: https://doi.org/10.7554/eLife.45644.033

## 3.3. Stable isotope analysis

Tracing the geographical origin of the raw material used to make archaeological ornaments is challenging; however, isotope geochemistry can provide palaeoenvironmental information on malacological fauna as well as shell ornaments (*Bajnóczi et al., 2013*; Bar-Yosef Mayer et al., 2012; *Eerkens et al., 2005*; *Vanhaeren et al., 2004*). This is because the stable isotopic composition of shells records the environmental conditions at the time when the organisms were active and deposited the shell mineral. In particular, the variation of $^{18}O/^{16}O$ ($\delta^{18}O$), which are relatively $^{18}O$-enriched in oceans and $^{18}O$-depleted in continental waters (*Keith et al., 1964*), can help to discriminate between freshwater *vs* marine carbonates. Similarly, the $^{13}C/^{12}C$ ($\delta^{13}C$) reflects the origin of the dissolved organic content in water and can be used to validate the inferences drawn from the analysis of $\delta^{18}O$.

### 3.3.1. Analytical procedure

Carbon and oxygen isotope analyses of biogenic carbonate (i.e. $\delta^{13}C_{carb}$ and $\delta^{18}O_{carb}$) for the Havnø and Hornstaad samples were measured at the Light Stable Isotope Laboratory (School of Archaeological Sciences, University of Bradford, UK) and for the Peştera Ungurească sample at the GISMO platform (UBFC, France). All samples were bleached for 48 hours using NaOCl (12% w/v) prior the analysis. The prismatic and nacreous layers of PesB were sampled separately.

Analytical procedure at Light Stable Isotope Laboratory (School of Archaeological Sciences, University of Bradford, UK): between 100 and 300 μg of the bleached calcium carbonate powders (Hornstaad and Havnø double-buttons) were loaded into 12 ml Exetainer tubes, and carbon and oxygen isotope values were determined by online phosphoric acid digestion at 70° C using a Thermo GasBench two preparation system coupled to a Thermo Delta V Advantage Isotope-Ratio mass spectrometer. Standardization of $\delta^{18}O$ values against the V-PDB reference was undertaken using repeated measurements of international standards IAEA NBS-19, IAEA-CO-8 and IAEA-CO-1, as well as internal laboratory standards (Merck $CaCO_3$ and OES). The analytical precision of the instrument was better than $\pm$ 0.1 ‰.

Analytical procedure at GISMO platform (Biogéosciences, UBFC, France): small amounts of carbonate from each layer was removed from a bleached fragment of the Peştera Ungurească sample by scratching with a scalpel. The powder obtained (between 35 and 40 μg) were loaded into glass vials for isotopic analysis. Samples were reacted with 250 μl of 100% phosphoric acid at 70°C for 12 min. All isotopic values are reported in the standard $\delta$-notation in permil (‰) vs VPDB. The reproducibility (2σ) of the IAEA NBS19 used as an external standard is better than 0.04 ‰ for the $\delta^{13}C$ and 0.08 ‰ for the $\delta^{18}O$.

### 3.3.2. Results

Stable isotope analyses yielded average $\delta^{18}O$ and $\delta^{13}C$ values ranging between $-5.3 \pm 0.4$ and $-11.1 \pm 0.6$ ‰ for Havnø (n = 3),$-6.1 \pm 1.0$ and $-11.9 \pm 1.7$ ‰ for Peştera Ungurească (n = 2) and $-9.3 \pm 0.5$ and $-10.6 \pm 1.6$ ‰ for Hornstaad (n = 3), respectively (*Appendix 1—table 1*).

**Appendix 1—table 1.** Stable isotope composition of the biogenic carbonate of the double-buttons.

| Sample | $\delta^{13}C$ (‰) | $\delta^{18}O$ (‰) |
|---|---|---|
| HavA | −11.8 ± 0.19 | −5.0 ± 0.09 |
| HavB | −10.9 ± 0.09 | −5.7 ± 0.17 |
| HavC | −10.6 ± 0.07 | −5.3 ± 0.08 |

*Appendix 1—table 1 continued on next page*

*Appendix 1—table 1 continued*

| Sample | $\delta^{13}$C (‰) | $\delta^{18}$O (‰) |
|--------|-------------------|-------------------|
| HorA | −11.7 ± 0.07 | −9.8 ± 0.11 |
| HorB | −11.3 ± 0.07 | −9.2 ± 0.07 |
| HorC | −8.7 ± 0.07 | −8.9 ± 0.06 |
| PesB_n | −13.1 ± 0.01 | −6.8 ± 0.03 |
| PesB_p | −10.7 ± 0.01 | −5.5 ± 0.03 |

DOI: https://doi.org/10.7554/eLife.45644.034

The low average $\delta^{13}$C and $\delta^{18}$O values obtained for all of the samples suggest the inland provenance of the biogenic carbonates (*Keith et al., 1964*; *Leng and Lewis, 2016*). The average $\delta^{18}$O values for Havnø and Hornstaad show a difference of ~ 4‰. The $\delta^{18}$O of PesB is closer to that of Havnø, with an average of −6.1 ± 1.0‰ between the two layers of the shell.

A sensible difference between $\delta^{18}$O values probably implies different waters of provenance of the shells: $\delta^{18}$O values obtained from molluscan carbonate reflect the average stable oxygen isotopic composition of the environmental waters and temperature (*Verdegaal et al., 2005*) and thus different paleoenvironmental conditions in which the animals lived. *Keith et al. (1964)* note that at progressively higher altitudes and higher latitudes there is a depletion of $^{18}$O in continental waters. Indeed, Hornstaad is situated at the northern foot of the Alps and is influenced by waters running from the Swiss Alps, while Havnø lies at just 9 m above the sea level and the Peştera Ungurească site can be characterized by Carpathian landscape but lies in a valley surrounded by low altitude (up to 700 m) mountains.

The $\delta^{13}$C value of freshwater shells is mainly influenced by dissolved inorganic carbon, which typically display a wide range of values (e.g. −28‰ to −1‰) compared to seawater (0–1‰) (*Aucour et al., 2003*; *Gillikin et al., 2009*). However, the $\delta^{13}$C values of freshwater bivalve shells can also be affected by some contribution of metabolic carbon (*McConnaughey and Gillikin, 2008*). The $\delta^{13}$C in the three double-button sets show similar average values: −11.1 ± 0.6‰ for Havnø, −10.6 ± 1.6‰ for Hornstaad and −11.9 ± 1.7‰ for Peştera Ungurească, and fall in the range of values reported for freshwater shells in European continental waters, including unionoids (*Aucour et al., 2003*; *Versteegh et al., 2010*). The shell $\delta^{13}$C values likely reflect similar environmental conditions, all indicating freshwater habitats. Taken together, the $\delta^{18}$O and $\delta^{13}$C values suggest that the shells used to make the ornaments were probably sourced locally, from freshwater or freshwater-dominant environment (estuaries).

## 3.4. Chiral Amino Acid Analysis

The chiral amino acids isolated and extracted from the intracrystalline fraction of proteins contained in shells can be used to obtain geochronological information, based on the extent of diagenesis or racemization (AAR) (*Bosch et al., 2015*; *Demarchi et al., 2015*; *Demarchi et al., 2011*; *Ortiz et al., 2018*; *Penkman et al., 2011*; *Penkman et al., 2007*; *Pierini et al., 2016*). Additionally, the bulk amino acid composition can yield taxonomic information at a broad level (order), and this can be applied to shell ornaments (*Demarchi et al., 2014*). Here we used a well-established preparation procedure (see for example *Penkman et al., 2008*), which includes a 48-hours bleaching step (NaOCl, 12% w/v) followed by acid hydrolysis (7M HCl, 24 hours at 110˚C) in order to obtain the composition and extent of racemization of the total hydrolysable amino acids (THAA) of all shell ornaments. Hydrolyzates were evaporated to dryness and rehydrated with a solution containing an internal standard (the non-protein amino acid L-homo-arginine) for analysis in duplicate by reverse-phase high-pressure liquid chromatography (RP-HPLC). One subsample was analysed for each of the Hornstaad and Peştera Ungurească double-buttons, and two for

each of the ones from Havnø (due to the larger size of the latter). *Appendix 1—table 2* and *Appendix 1—table 3* report the THAA concentration and D/L values for each of the double-buttons.

The relative composition data (*Appendix 1—figure 7*) show that all double-buttons display similar bulk amino acid signatures, except for HorA, where the %Asx is lower than in the other examples. This suggests that the proteins in this sample have undergone rapid degradation, and this is likely due to their exposure to high temperatures. PesB also displays low %Asx values, but this might be attributed to a species effect. The Principal Component Analysis (PCA) scores plot in *Appendix 1—figure 8* shows that the bulk amino acid composition of the Havnø double-buttons is very similar to that of freshwater molluscs *Unio pictorum* and *Margaritifera margaritifera*. HorB and HorC are also falling in a similar area of the PCA plot (close to the tiny aragonitic gastropod *Rissoa*), while HorA and PesB both display different compositions. Diagenesis might be invoked to explain the results for HorA, but PesB cannot be easily classified using bulk amino acid composition data. The D/L values (*Appendix 1—figure 9*) are low for all amino acids. PesB is the sample with the lowest values, consistent with the slightly younger age of the Peştera Ungurească site compared to the other two. Notable exception are all the D/L values measured in HorA, as well as Asx D/Ls for HorB and HorC, which are higher than in all the other beads. This may indicate mild exposure to heat, as observed in other studies (*Crisp, 2013*; *Demarchi et al., 2011*). The fact that the double-buttons from Hornstaad display signs of possible burning is hardly surprising, given that the site was affected by a fire!, but this type of analysis is fundamental in order to assess whether the D/L values yield genuine information regarding the age of the specimen, and for any taxonomic inference.

**Appendix 1—table 2.** Total hydrolysable amino acid (THAA) concentrations measured in archaeological double-button samples (pmol/mg). Average and standard deviation were calculated on two analytical replicates. Values for Havnø include the average and standard deviation for the two subsamples taken from each double-button.

| | [Asx] | | [Glx] | | [Ser] | | [Gly] | | [Ala] | | [Val] | | [Phe] | | [Ile] | |
| --- | --- | --- | --- | --- | --- | --- | --- | --- | --- | --- | --- | --- | --- | --- | --- | --- |
| | AV | SD | AV | SD | AV | SD | AV | SD | AV | SD | AV | SD | AV | SD | AV | SD |
| HOR-A | 150 | 1 | 497 | 6 | 72 | 0 | 965 | 36 | 604 | 0 | 300 | 1 | 227 | 2 | 223 | 3 |
| HOR-B | 340 | 128 | 322 | 126 | 142 | 47 | 516 | 301 | 372 | 167 | 250 | 40 | 175 | 39 | 181 | 32 |
| HOR-C | 429 | 61 | 340 | 11 | 177 | 25 | 574 | 68 | 434 | 46 | 194 | 43 | 163 | 11 | 160 | 19 |
| HAV-A | 723 | 48 | 470 | 17 | 334 | 12 | 1274 | 82 | 644 | 57 | 262 | 25 | 212 | 20 | 163 | 11 |
| HAV-B | 711 | 87 | 470 | 45 | 346 | 25 | 1214 | 176 | 656 | 66 | 264 | 24 | 213 | 18 | 173 | 18 |
| HAV-C | 703 | 14 | 452 | 13 | 321 | 21 | 1132 | 228 | 602 | 19 | 241 | 5 | 204 | 6 | 138 | 8 |
| PES-B | 452 | 8 | 344 | 13 | 494 | 1 | 1605 | 66 | 699 | 1 | 368 | 1 | 282 | 1 | 264 | 13 |

DOI: https://doi.org/10.7554/eLife.45644.035

**Appendix 1—table 3.** Total hydrolyzable amino acid (THAA) D/L values measured in archaeological double-buttons. Average and standard deviation were calculated on two analytical replicates. Values for Havnø include the average and standard deviation for the two subsamples taken from each double-button.

| | Asx D/L | | Glx D/L | | Ser D/L | | Ala D/L | |
| --- | --- | --- | --- | --- | --- | --- | --- | --- |
| | AV | SD | AV | SD | AV | SD | AV | SD |
| HOR-A | 0.430 | 0.000 | 0.740 | 0.003 | 0.000 | 0.000 | 0.880 | 0.001 |
| HOR-B | 0.530 | 0.073 | 0.150 | 0.421 | 0.720 | 0.515 | 0.200 | 0.477 |

*Appendix 1—table 3 continued on next page*

*Appendix 1—table 3 continued*

| | Asx D/L | | Glx D/L | | Ser D/L | | Ala D/L | |
|---|---|---|---|---|---|---|---|---|
| | AV | SD | AV | SD | AV | SD | AV | SD |
| HOR-C | 0.580 | 0.033 | 0.170 | 0.016 | 0.760 | 0.026 | 0.210 | 0.010 |
| HAV-A | 0.313 | 0.015 | 0.128 | 0.032 | 0.455 | 0.030 | 0.165 | 0.013 |
| HAV-B | 0.318 | 0.010 | 0.135 | 0.010 | 0.480 | 0.024 | 0.168 | 0.010 |
| HAV-C | 0.308 | 0.010 | 0.120 | 0.008 | 0.475 | 0.026 | 0.175 | 0.006 |
| PES-B | 0.310 | 0.000 | 0.080 | 0.000 | 0.280 | 0.000 | 0.100 | 0.000 |
| | Val D/L | | Phe D/L | | Ile D/L | | | |
| | AV | SD | AV | SD | AV | SD | | |
| HOR-A | 0.740 | 0.001 | 0.730 | 0.004 | 0.850 | 0.029 | | |
| HOR-B | 0.110 | 0.449 | 0.290 | 0.308 | 0.140 | 0.488 | | |
| HOR-C | 0.080 | 0.019 | 0.320 | 0.026 | 0.100 | 0.039 | | |
| HAV-A | 0.073 | 0.015 | 0.180 | 0.008 | 0.045 | 0.052 | | |
| HAV-B | 0.075 | 0.013 | 0.193 | 0.013 | 0.065 | 0.044 | | |
| HAV-C | 0.078 | 0.010 | 0.170 | 0.008 | 0.023 | 0.045 | | |
| PES-B | 0.000 | 0.000 | 0.140 | 0.000 | 0.000 | 0.000 | | |

DOI: https://doi.org/10.7554/eLife.45644.036

Overall, chiral amino acid data support the hypothesis that the shells used by the makers of the double-buttons were either fresh or recently dead and had not been collected from fossil deposits. This is important information in order to address the debate over the added value of using fossil shells for making jewellery (e.g. *Taborin, 1974*).

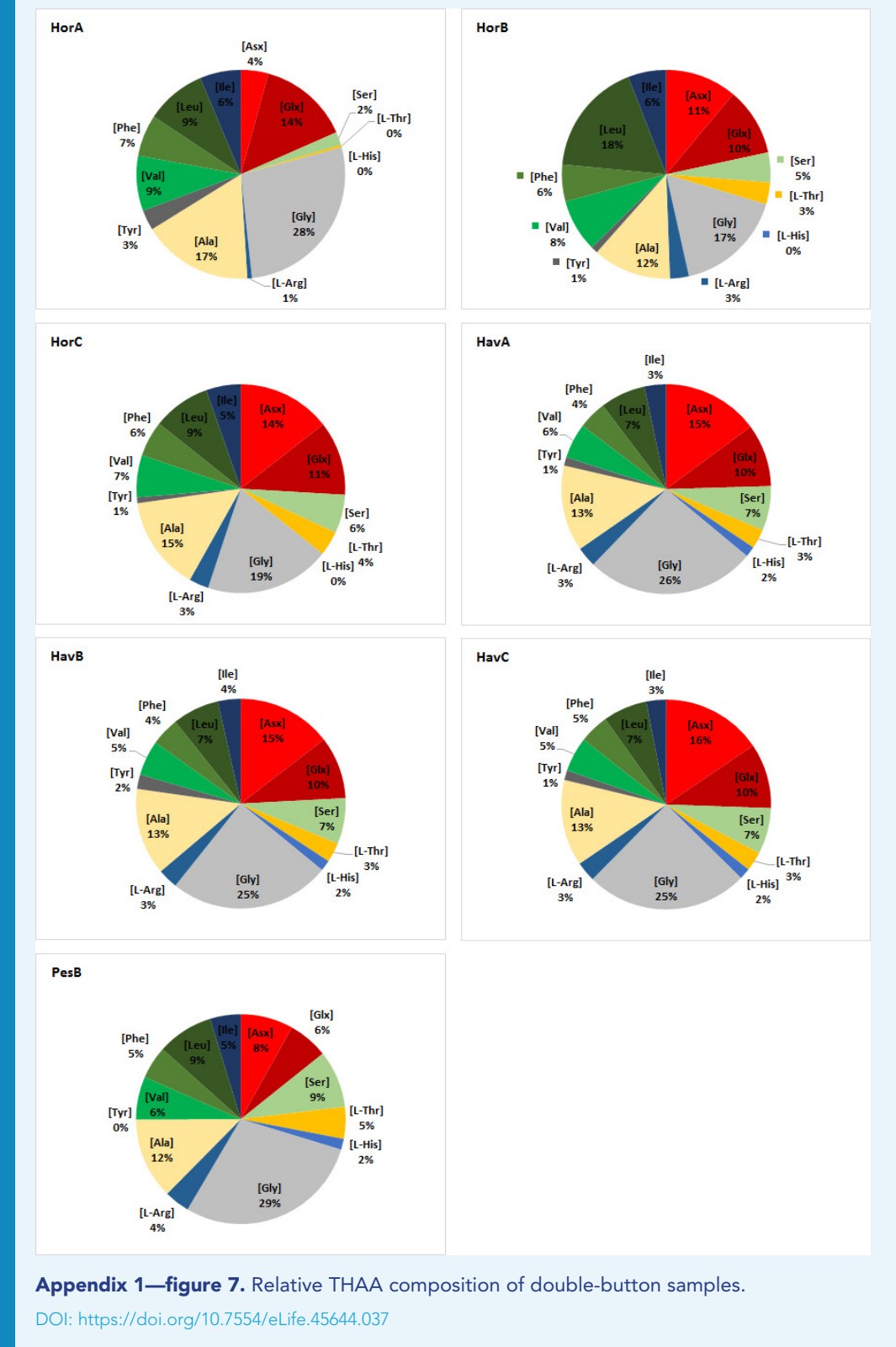

**Appendix 1—figure 7.** Relative THAA composition of double-button samples.

DOI: https://doi.org/10.7554/eLife.45644.037

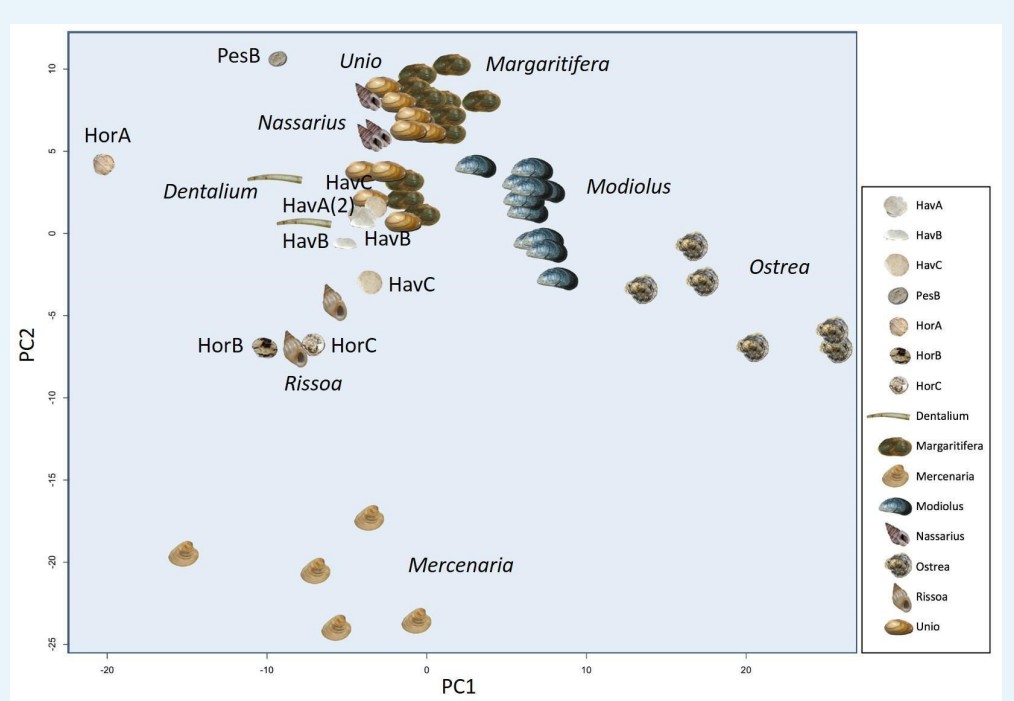

**Appendix 1—figure 8.** Principal Component Analysis (PCA) plot showing the similarity or differences between the amino acid composition of double-buttons and a range of shell taxa (reference taxa from *Demarchi et al., 2014*).

DOI: https://doi.org/10.7554/eLife.45644.038

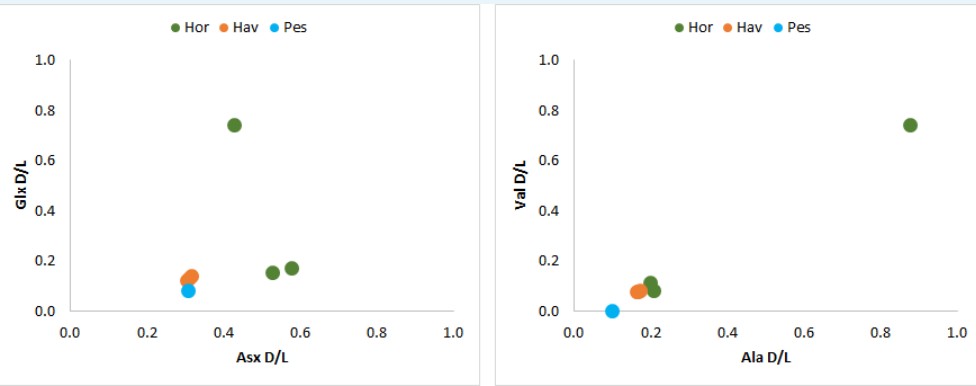

**Appendix 1—figure 9.** Total hydrolysable amino acid D/L values for all double-buttons (Glx vs Asx, left; Val vs Ala, right).

DOI: https://doi.org/10.7554/eLife.45644.039

## 3.5. 'Shellomics'

### 3.5.1. Insights into the 'Biomineralization toolkit'

'Shellomics' - a proteomics-based approach to study the processes of biomineralization, fills an important gap of knowledge in the evolution of shell biomineralization (*Marin et al., 2016*), and can also help to answer diverse environmental and archaeological questions. Despite the great attention that this field has gained in the past years and the numerous scientific articles published, considering there are at least 85000 mollusc species documented, our knowledge of the proteins trapped inside the biomineral skeleton and their functions still remains in its infancy.

Just after the first studies of shell proteomes, which were driven by the idea of uncovering the ancestral shell 'biomineralization toolkit' (**Marin et al., 2012**), there has been a steady increase in the number of mollusc shell proteins identified, owing to the development of 'shellomic' approaches, mainly by LC-MS/MS analysis (**Marin et al., 2013**). Shell matrix proteins, usually subdivided into acid-soluble (ASM) and acid-insoluble (AIM), have been extensively investigated for their role in biomineralization processes (**Albeck et al., 1993**; **Arivalagan et al., 2017**; **Marin and Luquet, 2004**) and with regard to their presence within the mineral skeleton (intracrystalline vs intercrystalline; related to specific structural layers) (**Marin et al., 2007**). Most of the research concerning biomineralization has been carried out on nacreous shells, especially marine pearl oysters (**Marie et al., 2017**; **Marin et al., 2013**; **Marin et al., 2012**), because of the importance of mother-of-pearl both commercially and as a source for biomimetic and bioinspired materials (**Westbroek and Marin, 1998**).

### 3.5.2. Protein extraction

Modern shell samples *Ostrea edulis, Modiolus modiolus, Unio pictorum, Margaritifera margaritifera* and sub-fossil shells *Unio crassus* and *Pseudunio auricularius* were scrupulously cleaned and abraded mechanically with a Dremel drill in order to remove the periostracal layer and surface contaminants. Shells were crushed and powdered to < 200 µm grain size. Shell powders (*Appendix 1—table 4*) were bleached for 48 hours in NaOCl (2.8% active chlorine) under constant agitation, then, bleach was removed by rinsing the powder in ultrapure water (5x) and ethanol (1x). Bleached dried shell powders were decalcified at 4 °C using 10% acetic acid (ratio: 50 mL of acid for 1 g of shell powder) under constant stirring overnight. The solution was centrifuged for 30 minutes at 4500 RPM at 5 °C in order to separate the acid-soluble matrix (ASM) from the acid-insoluble matrix (AIM). For modern shells, the supernatant (containing ASM proteins) was passed through a 5 µm filter, then ultrafiltered through a 10 kDa cut-off membrane (Amicon ultrafiltration cell) to reduce the volume to ~15 mL and finally dialysed (1 kDa cut-off) for 2 days against 1 L water (five water changes). The sub-fossil shell samples were ultrafiltered (10 kDa cut-off, final volume 0.75 mL) and washed with water (10 times). The AIM matrix was rinsed several times via cycles of resuspension in ultrapure water, centrifugation and removal of the supernatant. All matrices were lyophilized.

**Appendix 1—table 4.** Mass of biogenic carbonates analysed for proteomics

|  | Sample | Powder mass (mg) |
|---|---|---|
| Archaeological samples | HavA | 92.9 |
|  | HavB | 69.6 |
|  | HavC | 172.5 |
|  | HorA | 49.23 |
|  | HorB | 47.34 |
|  | Horc | 47.23 |
|  | PesB | 32.48 |
|  | Sample | Mass (g) |
| Freshwater unionoid shells | *U. pictorum* | 10 |
|  | *U. crassus* | 3 |
|  | *M. margaritifera* | 10 |
|  | *P. auricularius* | 3 |
| Marine shells | *O. edulis* | 10 |
|  | *M. modiolus* | 10 |

DOI: https://doi.org/10.7554/eLife.45644.040

The three archeological sample sets (HavA-B-C, HorA-B-C, PesB) were prepared as follows: fragments were crushed with a micro pestle directly in a clean eppendorf tube and

submerged in 2.8% NaOCl for 3 hours, then rinsed (five times with water and once in ethanol). Bleached powders were decalcified in 10% cold acetic acid under constant mixing (50 μL of acid per mg of shell powder) and upon the completion of demineralization, the solution was centrifuged at 13000 rpm for 30 minutes to separate AIM and ASM. AIM fractions were rinsed several times in water and lyophilized. The ASM matrices of HavA-B-C and HorA-B-C samples were passed through a 3 kDa cut-off VIVASPIN filter and rinsed with ultrapure water (using a total of 6 mL H$_2$O), to obtain a final volume of about 750 μL of ASM concentrate, which was then lyophilized. The ASM of PesB, due to the smaller amount of sample, was directly placed in a 3.5 kDa cut-off micro dialysis cassette (Slide-A-Lyzer G2 Dialysis Cassettes, 3 mL capacity, ThermoFisher) and dialysed in 1L of ultrapure water with gentle stirring (changing the water five times). The extract was then collected and lyophilized.

The purified protein extracts (of both reference shells and archeological double-buttons) were resuspended in a buffer (50 mM ammonium bicarbonate, pH 7.5–8). The reduction and alkylation of disulphide bonds was achieved using 1 M DL-Dithiothreitol (Sigma, Canada) for 1 hour at 65 °C and 0.5 M iodoacetamide (Sigma, USA) for 45 min at room temperature in the dark. Each sample was split into two aliquots for the enzymatic digestion with trypsin ('T') and elastase ('E'). Digestion was carried out overnight at 37 °C by adding: 4 μL trypsin (0.5 μg/μL; Promega, 2800 Woods Hollow Road Madison, WI 53,711 USA) for 'T' subsamples or 4 μL elastase (1 μg/μL; Worthington, Lakewood, NJ, USA) for 'E' subsamples. Digestion was stopped with 10% TFA (to a final concentration of 0.1% TFA), samples purified using C18 solid-phase extraction tips (Pierce zip-tip; Thermo-Fisher) and evaporated to dryness. Trypsin and elastase digests for each sample were resuspended and combined prior to LC-MS/MS analysis.

### 3.5.3. LC-MS/MS analysis

Reference shells *Ostrea edulis, Modiolus modiolus, Unio pictorum, Margaritifera margaritifera* and two sets of beads - HavA-B-C and HorA-B-C were analysed at MSAP CNRS laboratory, University of Lille; sample PesB and two sub-fossil reference shells, *Unio crassus* and *Pseudunio auricularius,* were analysed at the Mass Spectrometry Biomolecules core facility, Molecular Biotechnology and Health Sciences Department, University of Turin.

### Proteomic analysis (Lille)

Analysis of peptides were performed using a nanoflow HPLC instrument (U3000 RSLC Thermo Fisher Scientific) coupled to a Q Exactive Plus mass spectrometer (Thermo Fisher Scientific) equipped with a nanoelectrospray ion source. A volume of 1 μL of peptide mixture was loaded onto the preconcentration trap (Thermo Scientific, Acclaim PepMap100 C18, 5 μm, 300 μm i.d x 5 mm) using a partial loop injection and a flow rate of 10 μL.min$^{-1}$ (duration 5 min) with buffer A (5% acetonitrile and 0.1% formic acid). The peptide mixture was then separated using a nanocolumn (Acclaim PepMap100 C18, 3 μm, 75 mm i.d. × 500 mm) and a linear gradient of 5–40% buffer B (75% acetonitrile and 0.1% formic acid) at a flow rate of 250 nL.min$^{-1}$ and a temperature of 45 °C. The total duration of an LC MS/MS run was 120 min. MS data were acquired using a data-dependent top 20 method that selects the 20 most abundant precursor ions from the survey scan (400–1600 m/z range) for HCD fragmentation. The dynamic exclusion duration was 60 s. The isolation of precursors was performed with a 1.6 *m/z* window and MS/MS scans were acquired with a starting mass of 80 *m/z*. Survey scans were acquired at a resolution of 70,000 at *m/z* 400 (AGC set to 106 ions with a maximum fill time of 180 ms). Resolution for HCD spectra was set to 35,500 at *m/z* 200 (AGC set to 105 ions with a maximum fill time of 120 ms). Normalized collision energy was 28 eV. The underfill ratio, which specifies the minimum percentage of the target value likely to be reached at maximum fill time, was defined as 0.3%. The instrument was run with the peptide recognition mode (i.e. from 2 to 8 charges) and exclusions of singly charged ions and unassigned precursor ions. Fifteen blanks of 1 hr and one reference sample (80 fmol of Cytochrome C digest; Thermo Fisher Scientific) were injected and controlled (e.g. retention time, sensitivity, carry over, contaminations) between each sample injection.

## Proteomic analysis (Turin)

An Ultimate 3000 Dionex nanoHPLC instrument coupled with an Orbitrap Fusion (Thermo Scientific, Milan, Italy) mass analyzer were used. The separation was achieved using a PepMap RSLC C18, 2 µm, 100 Å, 75 µm × 50 cm column (Thermo Scientific) and a PepMap C18, 5 µm × 5 mm, 100 Å preconcentration column (Thermo Scientific). The eluent used for preconcentration step was 0.05% trifluoroacetic acid in water/acetonitrile 98/2 and the flowrate was 5 µL/min. The eluents used for chromatographic separation were 0.1% formic acid in water (solvent A) and 0.1% formic acid in acetonitrile/water 8/2 (solvent B) in a program which was initially isocratic at 5:95 (A:B %) for 5 min, increased to 75:25 in 55 min, run up to 60:40 in 6 min, and to 10:90 in 5 min. Recondition time was 20 min. The injection volume was 1 µL and the flow rate 300 nL min$^{-1}$. The nanocolumn was provided with the ESI source. The mass spectrometry parameters were: positive spray voltage 2300 (V), sweep gas 1 (Arb) and ion transfer tube temperature 275°C. Full scan spectra were acquired in the range of $m/z$ 375–1500 (resolution 120000 @ $m/z$ 200). MS$^n$ spectra in data dependent analysis were acquired in the range between the ion trap cut-off and precursor ion $m/z$ values. HCD collision energy was fixed at 28%, orbitrap resolution 50000 and the isolation window was 1.6 $m/z$ units.

### 3.5.4. Reference datasets used for proteomics data analysis

Publicly available reference sequences for molluscs are far from abundant, given the size of the phylum. Most of the proteomic studies on bivalve nacre proteins have been carried out on sub-class Pteriomorphia and very limited informations is known of other nacre bearing bivalves such as freshwater pearl mussels (**Marie et al., 2017**).

We performed two separate searches for each sample, using:

- A 'Protein' database, downloaded from NCBI (https://www.ncbi.nlm.nih.gov/) on 15/02/2018 and including 633061 sequences (restricting the taxonomy to 'Mollusca')
- An 'EST' database downloaded from NCBI (https://www.ncbi.nlm.nih.gov/) on 15/02/2018 and including 1149723 sequences (restricting the taxonomy to 'Mollusca')

The Protein database nominally includes a large number of species (e.g. more than 3000 bivalves), but only a few organisms are represented by a significant number of sequences (e.g. the marine bivalves *Crassostrea gigas*, *Mizuhopecten yessoensis*, *Crassostrea virginica*, *Mytilus galloprovincialis*, *Pinctada fucata*). The top-thirty freshwater bivalves represented in this database are (the number of sequences in brackets): *Cumberlandia monodonta* (561), *Popenaias popeii* (496), *Potomida littoralis* (357), *Elliptio hopetonensis* (269), *Dreissena presbensis* (200), *Cyprogenia aberti* (198), *Elliptio dariensis* (197), *Elliptio icterina* (182), *Obovaria jacksoniana* (158), *Unio tumidus* (129), *Anodonta anatina* (105), *Corbicula fluminea* (101), *Unio delphinus* (99), *Margaritifera falcata* (98), *Toxolasma parvus* (97), *Pyganodon grandis* (97), *Hyriopsis cumingii* (95), *Unio mancus* (89), *Utterbackia imbecillis* (89), *Utterbackia peninsularis* (86), *Lampsilis cardium* (84), *Strophitus radiatus* (84), *Unio elongatulus* (82), *Quadrula pustulosa* (76), *Venustaconcha ellipsiformis* (73), *Sinanodonta woodiana* (67), *Anodonta cygnea* (62), *Unio crassus* (61), *Elliptio complanata* (60). However, the sequences are usually not relevant to the shell or the shell mantle: for example, all 61 *Unio crassus* sequences refer to two proteins: cytochrome oxidase subunit I and NADH dehydrogenase subunit.

The EST sequence database contains 75 molluscan species (25 gastropods, 38 bivalves, eight cephalopods, 2 polyplacophorans and two shell-less molluscs). For bivalves, the EST sequences include well-studied pearl-producing species, most of which belonging to the Pterioida order (*Pinctada* sp.), but also the edible oyster and the mussel (*Ostreoida* and *Mytilidae* sp.). Only 2 EST sets correspond to freshwater bivalves: *Lamellidens marginalis* (native to southeast Asia) and *Hyriopsis cumingii* (commonly known as Triangle Sail Mussel, native to China and Vietnam), both belonging to order Unionoida.

### 3.5.5. Peptide and protein identification

Product ion spectra of reference shells and archaeological double-buttons were analysed using PEAKS Studio (v. 8.5, Bioinformatics Solutions Inc (BSI); (**Ma et al., 2003**) and searched

separately against the Mollusca Protein and Mollusca EST databases, including the search of common laboratory contaminants (cRAP; common Repository of Adventitious Proteins: http://www.thegpm.org/crap/). Search parameters were defined assuming no enzyme digestion, fragment ion mass tolerance of 0.05 Da and a parent ion tolerance of 10 ppm. Results obtained by SPIDER searches (i.e. including all possible modifications) were used for peptide identification and protein characterization, choosing the following threshold values for acceptance of high-quality peptides: false discovery rate (FDR) threshold 0.5%, protein scores $-10\lg P \geq 40$, unique peptides $\geq 2$, de novo sequences scores (ALC %) $\geq 50$. In some instances (indicated in the main text), less stringent parameters of $-10\lg P \geq 20$, unique peptides $\geq 1$, de novo sequences scores (ALC %) $\geq 50$ were also used to identify the presence or absence of specific proteins. The peptide sequences supporting shell protein identifications were individually checked using the Blastp tool (https://blast.ncbi.nlm.nih.gov/), and any sequences that were homologous to common laboratory contaminants were excluded from any further analysis.

### 3.5.6. Results

*Table 2—source data 1* summarizes the main proteins identified in the reference shell samples and the double-buttons, grouped by taxon. Here we include a brief overview of several freshwater shell 'specific' matrix proteins detected in our samples:

- Hic74 (GenBank: ARG42316.1) is an acidic (IP 4.68), alanine and glycine rich matrix protein, supposedly involved in nacreous layer formation (*Liu et al., 2017a*). Its structure consists of poly-alanine blocks (at least 17) with short acidic motifs, a 'GS loop' type coil structure of lustrin A (*Wustman et al., 2002*) at the C-terminus, the C-terminal 30 residues of which con- sist of short acidic-basic motifs. It is probably a structural, silk-like protein, providing mechanical function (a system that dissipates the cracks) with short acidic sites to bind the mineral.
- Hic52 (GenBank: ARH52598.1) is a very basic (IP 10.82), glycine (almost 29%) and alanine rich protein. It has poly-glutamine and poly-glycine blocks with several degenerate repeats of different lengths along the sequence. It was reported to have collagen-like structure (*Liu et al., 2017b*) and probably plays a structural role in nacre formation.
- Silkmapin (GenBank: AIZ03589.1) and its isoforms (nasilin 1 and 2 (GenBank: ASQ40996.1 and ASQ40997.1)) are glycine-rich (>34%), non-acidic proteins with a structural function and probably play a role in the deposition of both nacre and prismatic layers (*Liu et al., 2015*).
- Chitin deacetylase enzyme (present in isoforms A and B (GenBank: AFO53262.1 and AFO53263.1)) is a slightly basic (IP 8.34) protein, with a chitin deacetylation function and possibly has a role in the 3D structuring of the mineral framework.
- Carbonic anhydrase 3 (*Hyriopsis cumingii*, GenBank: ARG42317.1) is an asparagine (10.2%) and threonine (9.3%) rich, acidic (IP 5.9) protein. The active enzymatic domain comprises most of the sequence, except for the C-terminal 40 residues, which have completely biassed amino acid composition, enriched in asparagine and acidic residues. This protein is typical of a CA associated to calcium carbonate biominerals, as it possesses a supernumer- ary low complexity domain (*Le Roy et al., 2014*), which would promote the binding of the enzyme onto the mineral surface.

### 3.5.7. Protein identification in reference unionoid shells

All of the reference unionoid shells gave hits to proteins of the freshwater bivalve *Hyriopsis cumingii*. Protein Hic74 was the top hit protein identified in all of the unionoids, with coverages varying from 34% in *Margaritifera margaritifera* to 54% in *Unio pictorum* (*Appendix 1—figure 10 Appendix 1—figure 11*, *Appendix 1—figure 12*, *Appendix 1— figure 13*, *Appendix 1—figure 14*, *Appendix 1—figure 15*) - including the sub-fossil shells of *Pseudunio auricularius* (49%) and *Unio crassus* (50%). Another significant *H. cummingii* protein in unionoids was Hic52, which was found in all the analysed freshwater shells with protein coverage up to 17%. Silkmapin (and its isoforms nasilin 1, nasilin 2) were found in both *Unio* species (with coverage up to 17%), while in *M. margaritifera* and *P. auricularius* they were only identified using less stringent protein identification parameters (see section 3.5.5). We found carbonic anhydrase of *H. cumingii* in *Unio* species and in *M. margaritifera*,

while chitin deacetylase (isoform B) was only present in *U. pictorum*. Upsalin, a protein originally identified and described from the *U. pictorum* shell matrix (*Ramos-Silva et al., 2012*) was detected only in *Unio* species. In some of the unionoid samples we also detected some minor *Hyriopsis* proteins such as Krichin, Hic31, GTRPB5 and GTRPB7, which have not been described in the literature before and for which the function is currently unknown.

Proteins hits from shells other than *H. cummingii* were also found in the unionoid shell samples, such as MSI60-like and insoluble matrix proteins (from *Pinctada* species, the marine pearl oysters), poly-Ala protein Shelk2 and spidroin-1-like (from *Crassostrea* species, the marine oysters) and collagen-like protein precollagen D (*Mytilus* species). MSI60 is a structural protein of nacre and, together with Shelk2-like, spidroin-1-like and precollagen D, could be classified as a silk-like, repetitive low-complexity domain type (RLCD) proteins exhibiting poly-Ala motifs and repetitive Gly domains alternating with Ala, Ser and Leu. RLCD type proteins are structure-specific, with a molecular function that is probably not exclusively restricticted to nacre deposition (*Marin and Luquet, 2004*; *Sudo et al., 1997*) but to the overall biomineralization of nacro-prismatic structures (*Gao et al., 2015*; *Marie et al., 2017*). Supporting this, proteins such as MSI60-like was identified in this study in another nacreous shell, the marine mussel *Modiolus modiolus,* but were not present in the foliated calcitic shell of *Ostrea edulis*. This supported the lack of taxonomic specificity of RLC domains. Unfortunately, it still remains difficult to verify whether the RLCP-type shell matrix proteins are all true homologues, considering the limitations of the blastp tools for analysing sequences with low complexity domains.

The analysis of the Hic74 sequence from *U. pictorum* (*Appendix 1—figure 10*) and all protein-peptide data from other shells in *Table 2—source data 1*) shows several interesting points:

- cleavage sites for trypsin are relatively few, and therefore the use of elastase improved the coverage;
- 'natural' cleavage sites (e.g. N in position 756) can be observed;
- among the most frequent modifications are the loss of ammonia, oxidation, dihydroxylation, deamidation;
- the presence of RLCs (e.g. Ala-rich domains) means that spectra could cover any of the multiple domains, potentially biasing % coverage calculations.

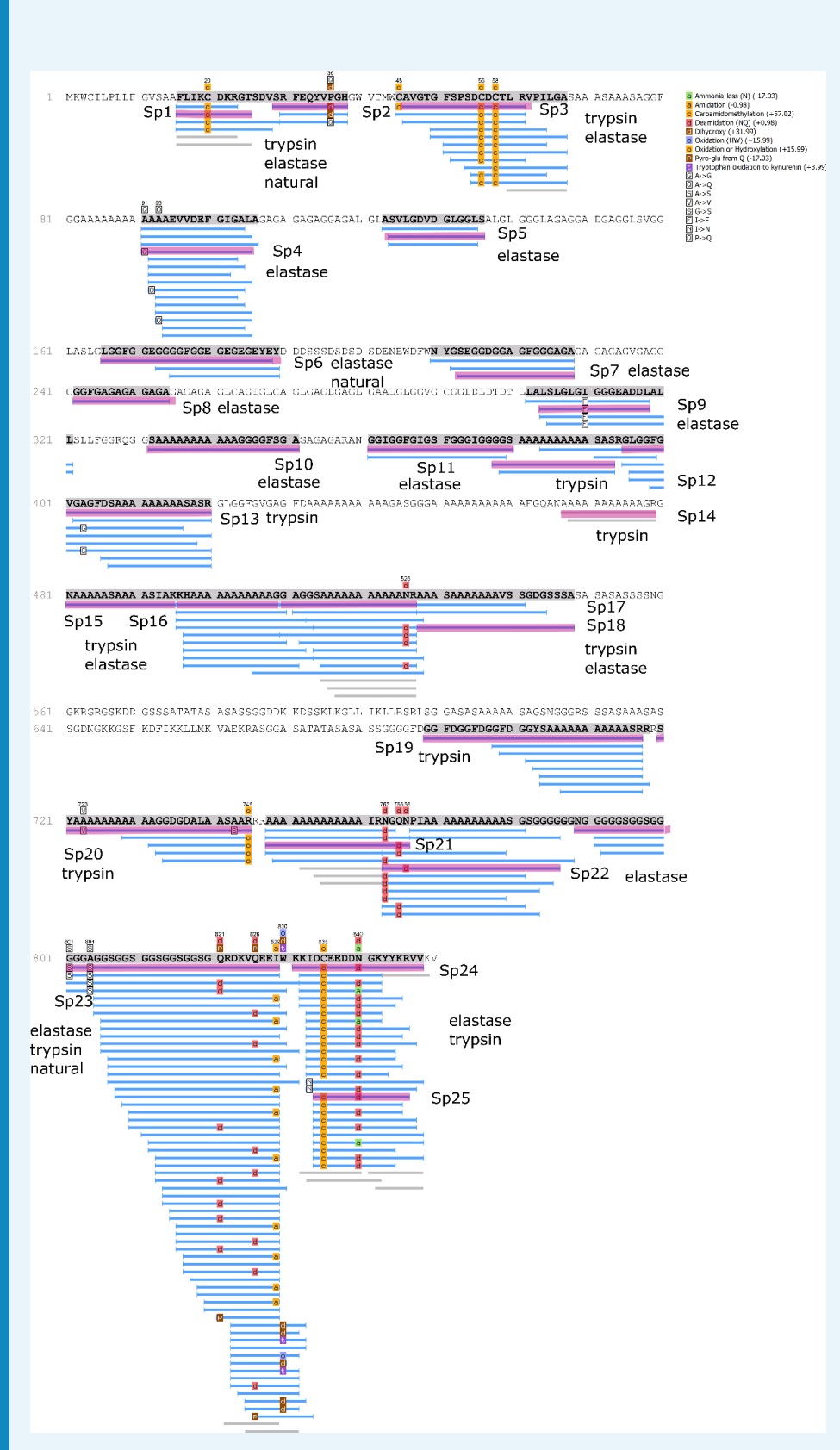

**Appendix 1—figure 10.** Protein Hic74 identified in *Unio pictorum*: sequence coverage, highlighting in pink the product ion spectra ('Sp') shown below. Sequences reconstructed by

assisted de novo on the basis of mono-charged ions mainly (spectra were acquired on the 400-1600 *m/z* range and multiply-charged ions were detected).

DOI: https://doi.org/10.7554/eLife.45644.041

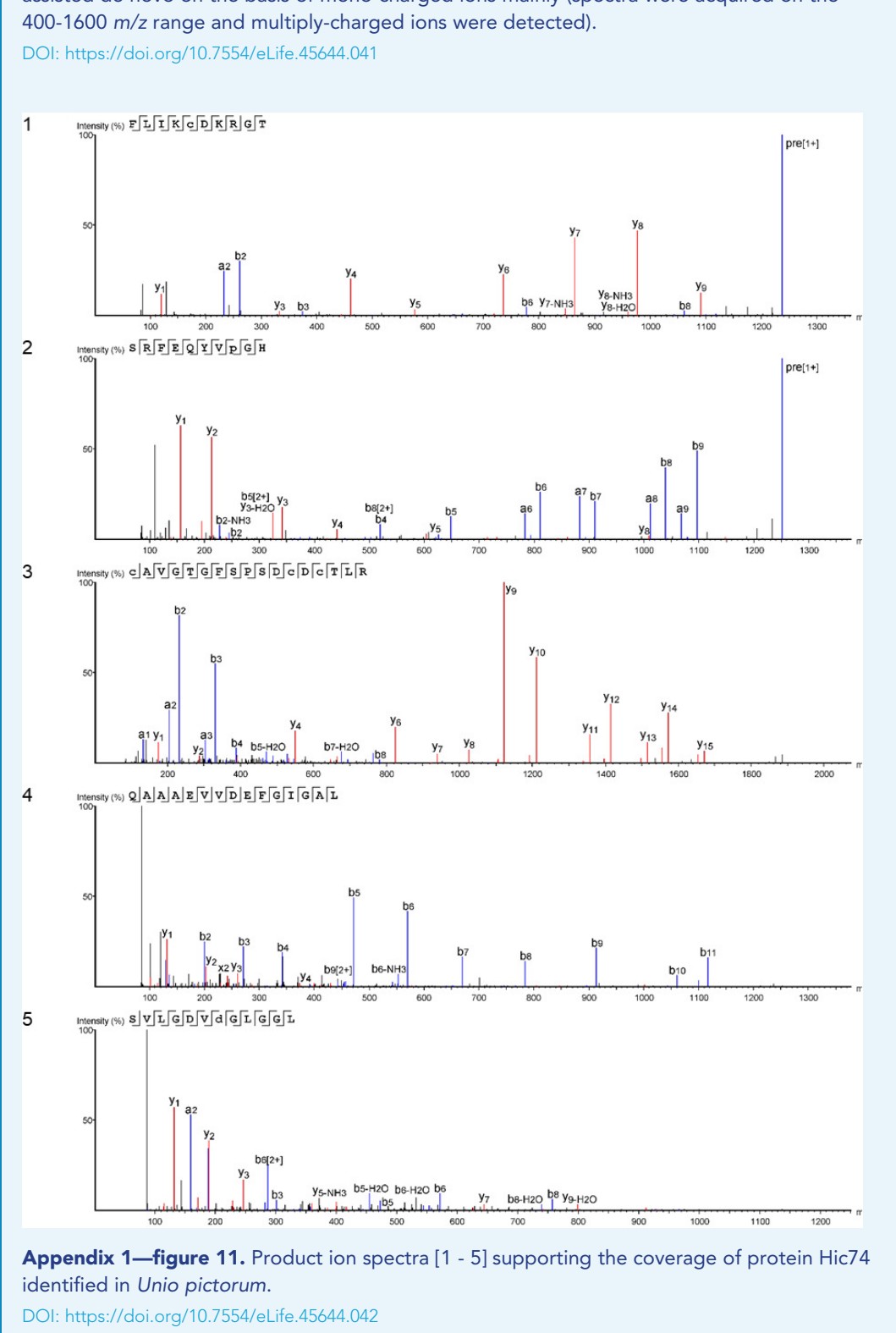

**Appendix 1—figure 11.** Product ion spectra [1 - 5] supporting the coverage of protein Hic74 identified in *Unio pictorum*.

DOI: https://doi.org/10.7554/eLife.45644.042

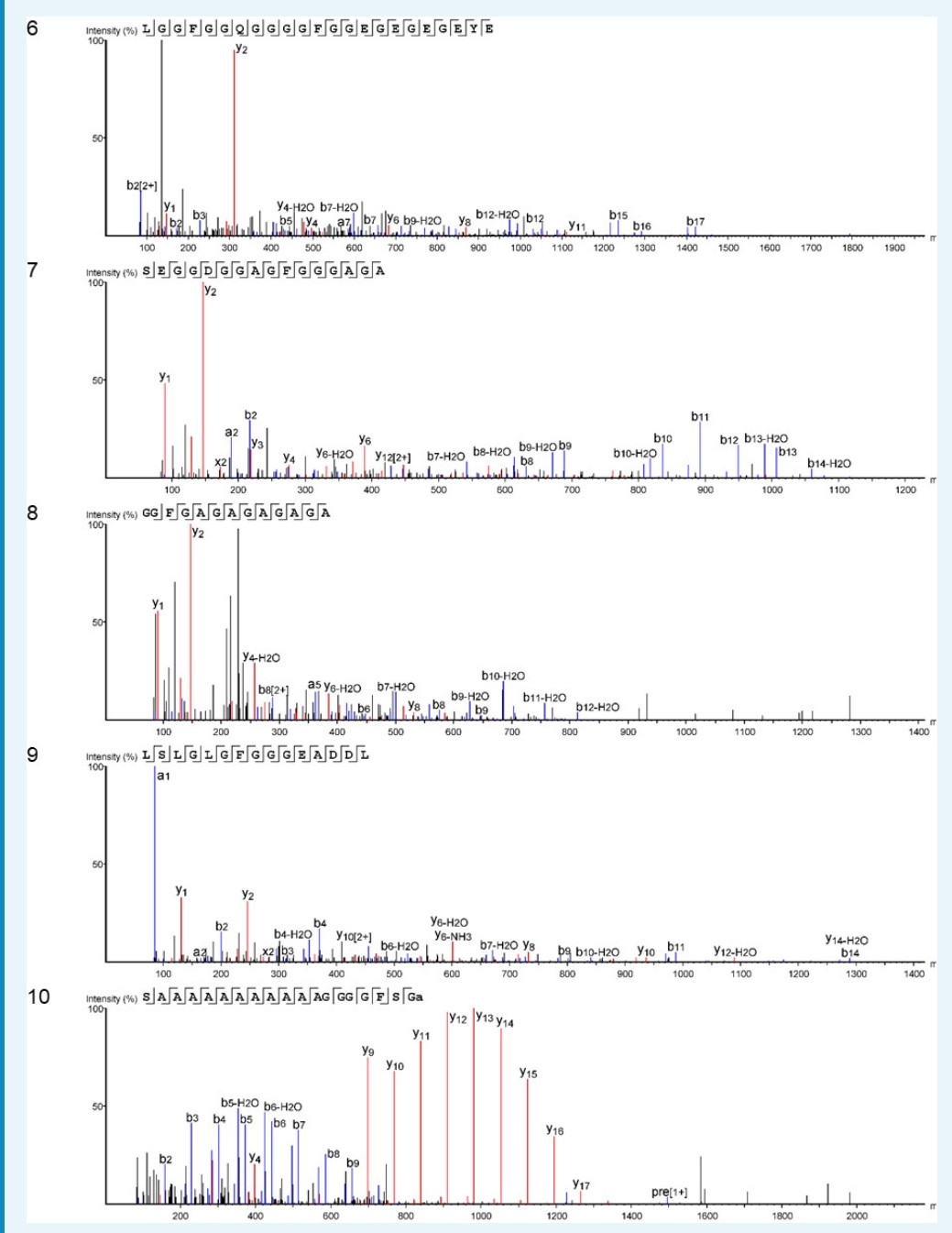

**Appendix 1—figure 12.** Product ion spectra [6 - 10] supporting the coverage of protein Hic74 identified in *Unio pictorum*.

DOI: https://doi.org/10.7554/eLife.45644.043

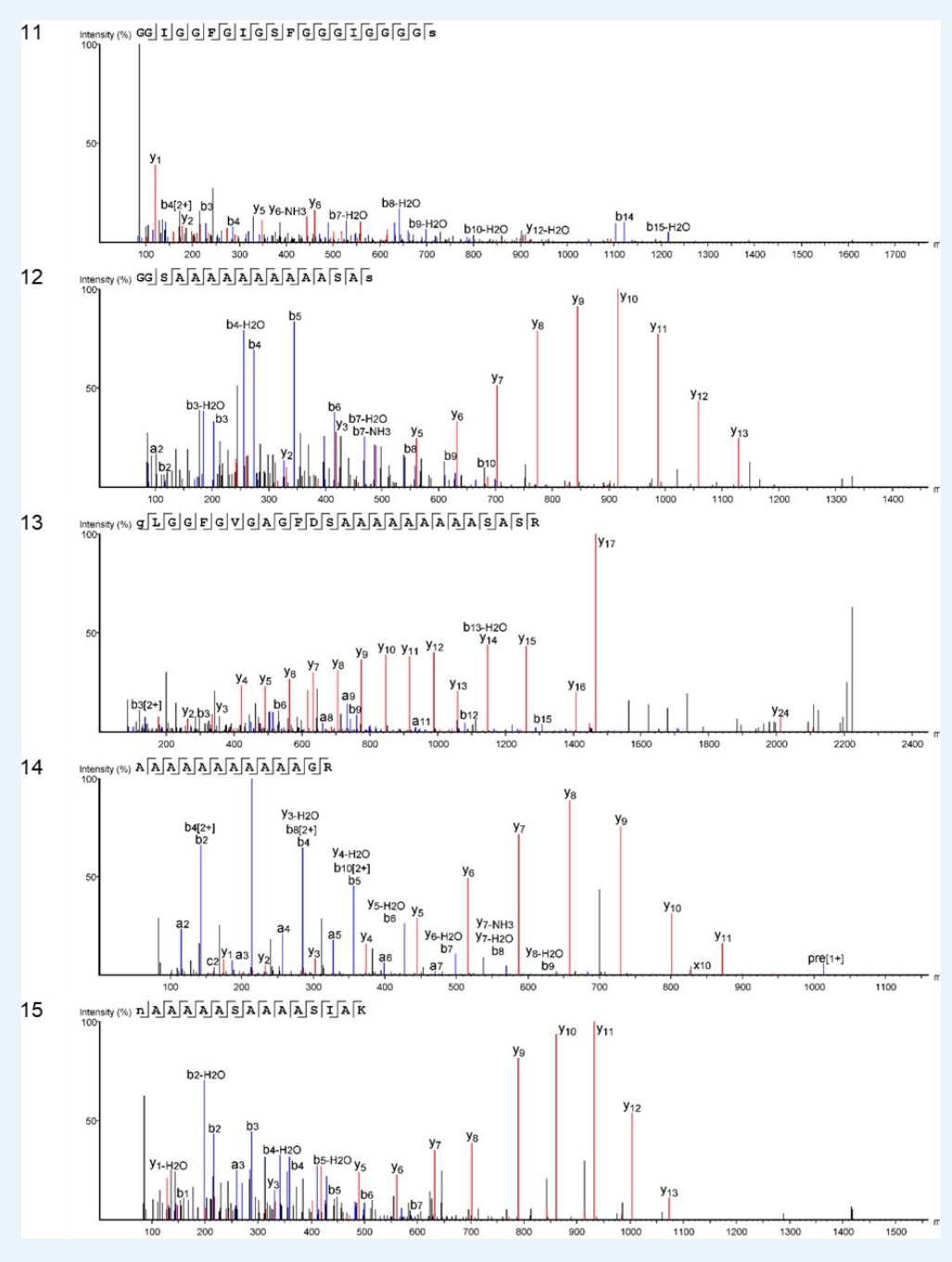

**Appendix 1—figure 13.** Product ion spectra [11 - 15] supporting the coverage of protein Hic74 identified in *Unio pictorum*.

DOI: https://doi.org/10.7554/eLife.45644.044

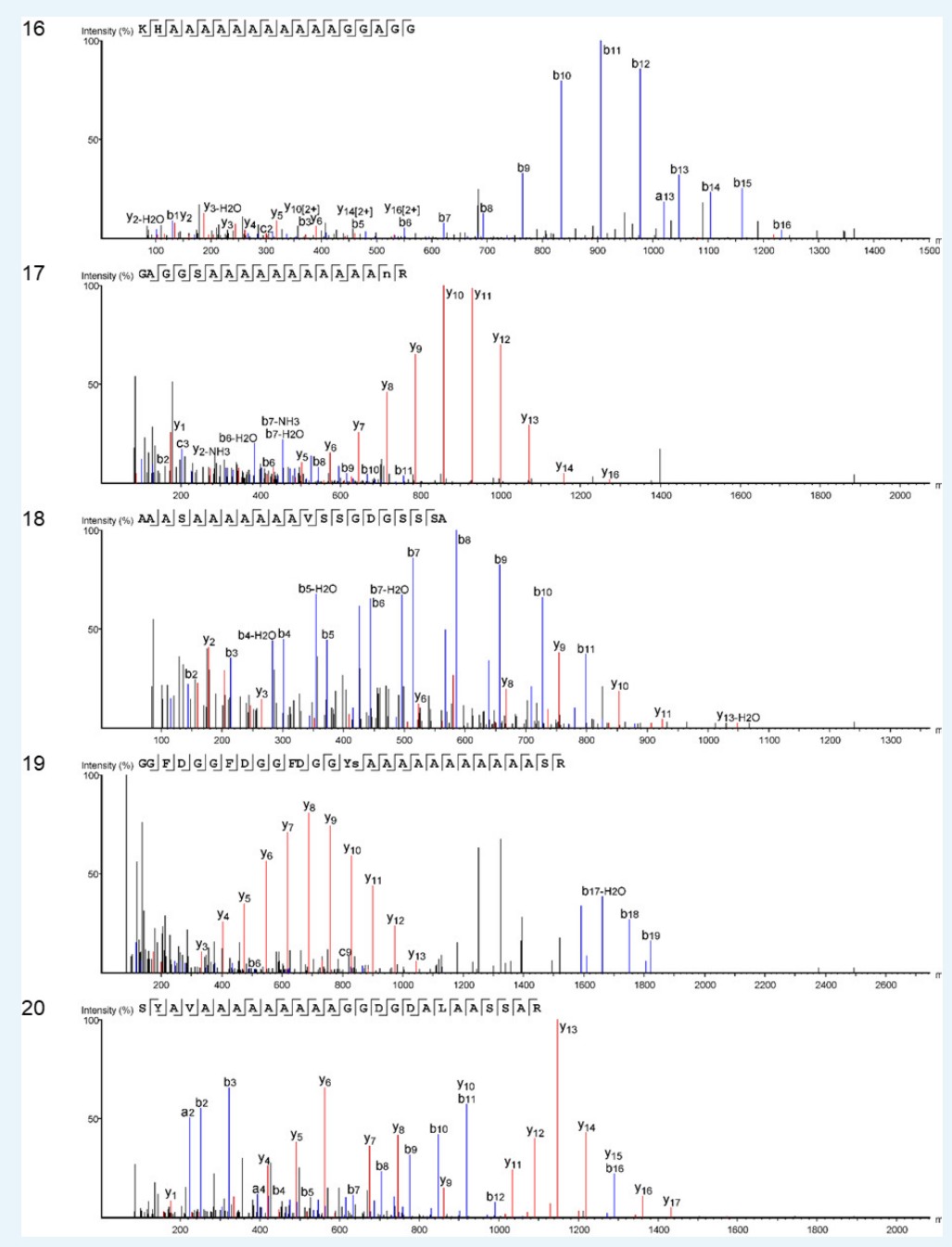

**Appendix 1—figure 14.** Product ion spectra [16 - 20] supporting the coverage of protein Hic74 identified in *Unio pictorum*.

DOI: https://doi.org/10.7554/eLife.45644.045

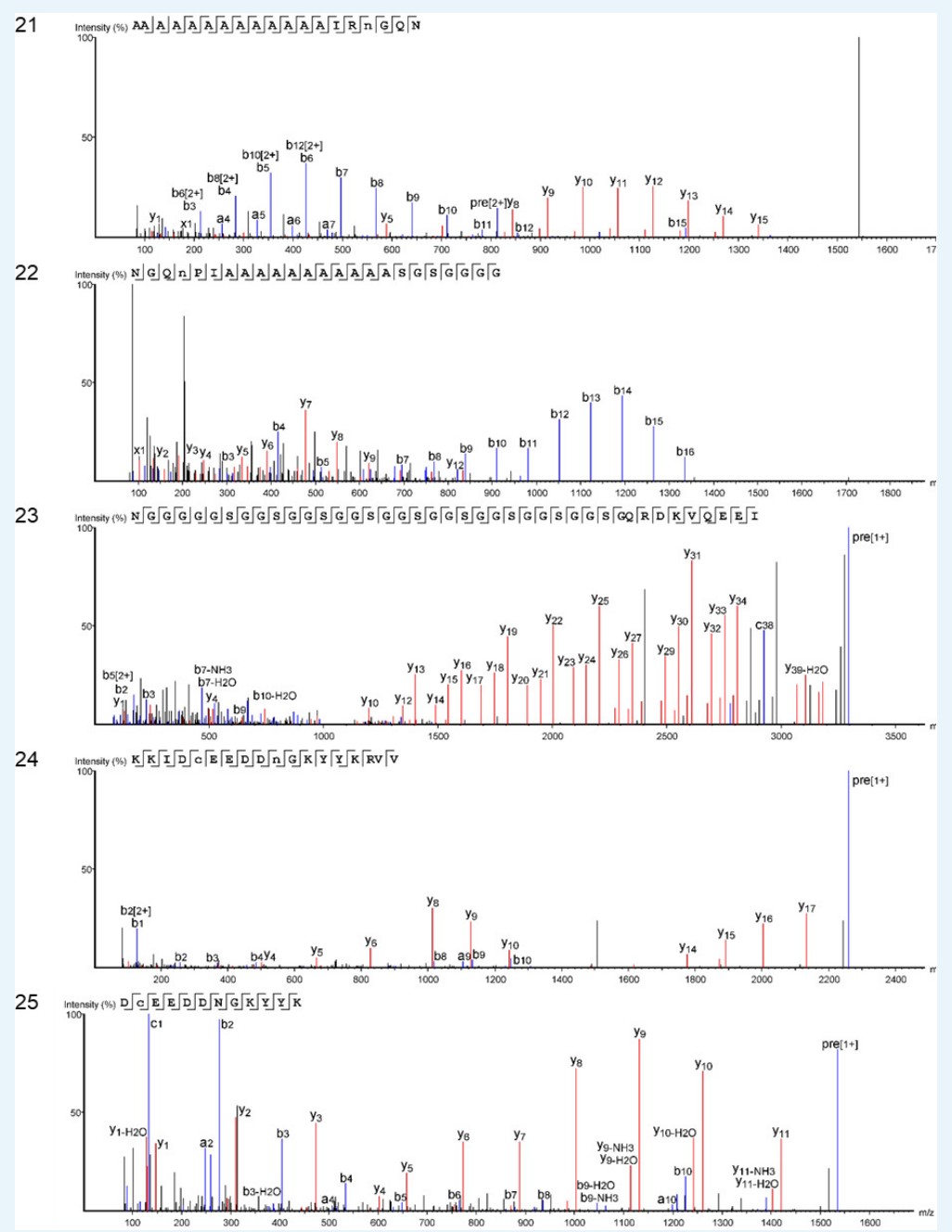

**Appendix 1—figure 15.** Product ion spectra [21 - 25] supporting the coverage of protein Hic74 identified in *Unio pictorum*.

DOI: https://doi.org/10.7554/eLife.45644.046

### 3.5.8. Protein identification in archaeological double-button samples

Protein Hic74 (*H. cummingii*) was the top hit for all the archaeological samples. *Appendix 1—figure 16*, *Appendix 1—figure 17*, *Appendix 1—figure 18*, *Appendix 1—figure 19*, *Appendix 1—figure 20*, *Appendix 1—figure 21* show the Hic74 sequence from sample HavC (for all other peptide-protein data see *Table 2—source data 1*) and it highlights that:

- some regions of Hic74 that were not covered in the sequence retrieved from *Unio pictorum* were present in the archaeological sample, for example residues 107→135;
- diagenetically-induced modifications such as deamidation and oxidation were frequent.

In particular, the extent of deamidation displayed by Gln (Q) and Asn (N) residues in Hic74 is broadly coherent with the relative rate of deamidation of N and Q, and with the age and the extent of degradation of the samples (*Appendix 1—table 5*):

- Hornstaad double-buttons were too degraded to yield any surviving N or Q;
- All remaining archaeological samples (Havnø, PesB, *U. crassus* and *P. auricularius*) display similar extent of N deamidation (~50–70%), which is slightly higher than for the modern shells *M. margaritifera* and *U. pictorum;*
- The extent of Q deamidation is low in HavA, PesB as well as in the modern shells *M. margaritifera* and *U. pictorum* (~10%), and higher in all other samples (~30–40%).

**Appendix 1—table 5.** Extent of Asn (N) and Gln (Q) deamidation (N→D; Q→E) in the peptides identified in the Hic74 sequence.

| | Hav A | Hav B | Hav C | Hor A | Hor B | Hor C | Pes B | *U. pict orum* | *U. cras sus* | *M. marga ritifera* | *P. auricu larius* |
|---|---|---|---|---|---|---|---|---|---|---|---|
| # Q | 28 | 105 | 196 | - | - | - | 12 | 136 | 237 | 13 | 37 |
| # N | 38 | 31 | 104 | | | | 4 | 71 | 96 | 8 | 88 |
| # Q→E | 3 | 33 | 71 | | | | 1 | 18 | 67 | 1 | 12 |
| # N→D | 19 | 17 | 53 | | | | 2 | 28 | 66 | 3 | 57 |
| % Q→E | 11 | 36 | 36 | | | | 8 | 13 | 28 | 8 | 32 |
| % N→D | 50 | 51 | 51 | | | | 50 | 39 | 69 | 37 | 65 |

DOI: https://doi.org/10.7554/eLife.45644.047

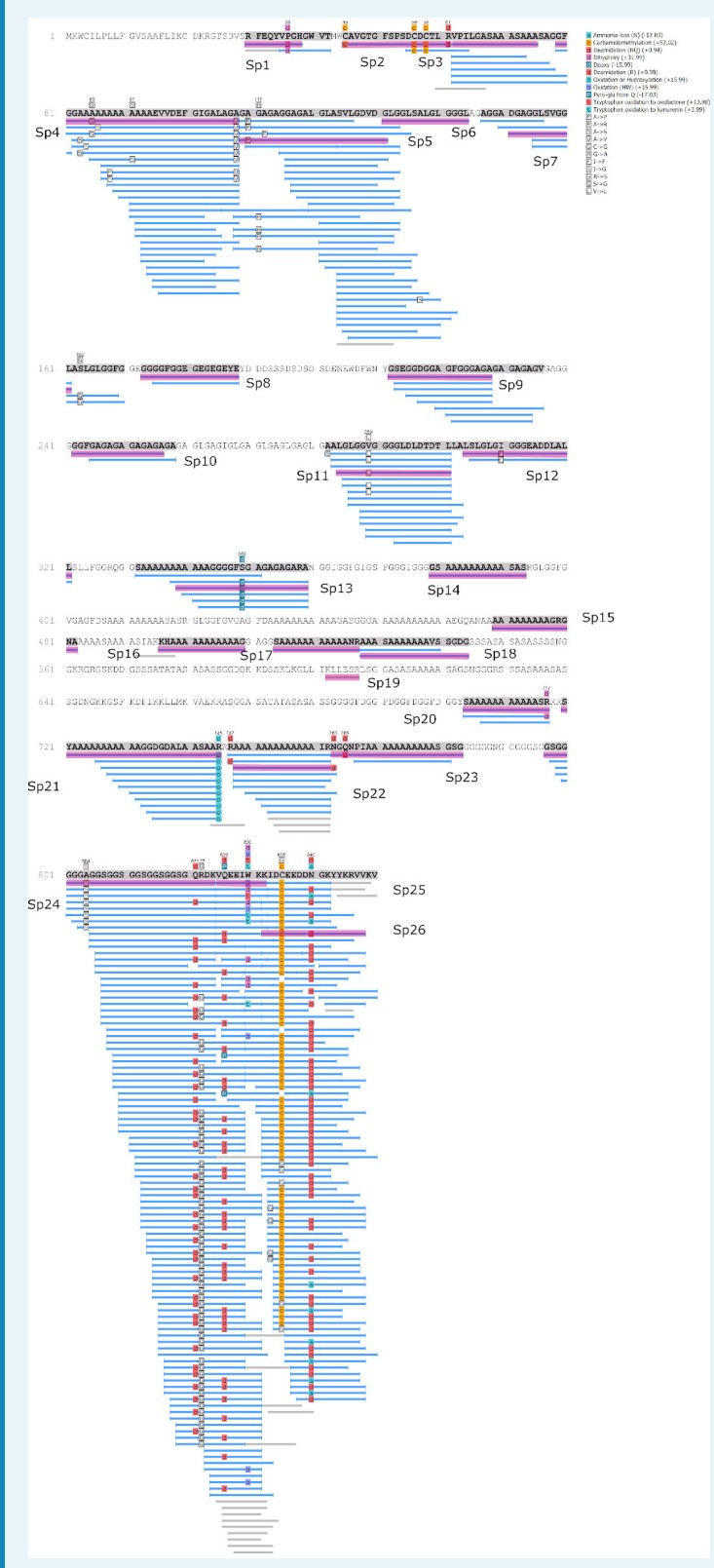

**Appendix 1—figure 16.** Protein Hic74 identified in the double-button HavC: sequence coverage, highlighting in pink the product ion spectra ('Sp') shown below. Sequences reconstructed by assisted de novo on the basis of mono-charged ions mainly (spectra were acquired on the 400-1600 *m/z* range and multiply-charged ions were detected).

DOI: https://doi.org/10.7554/eLife.45644.048

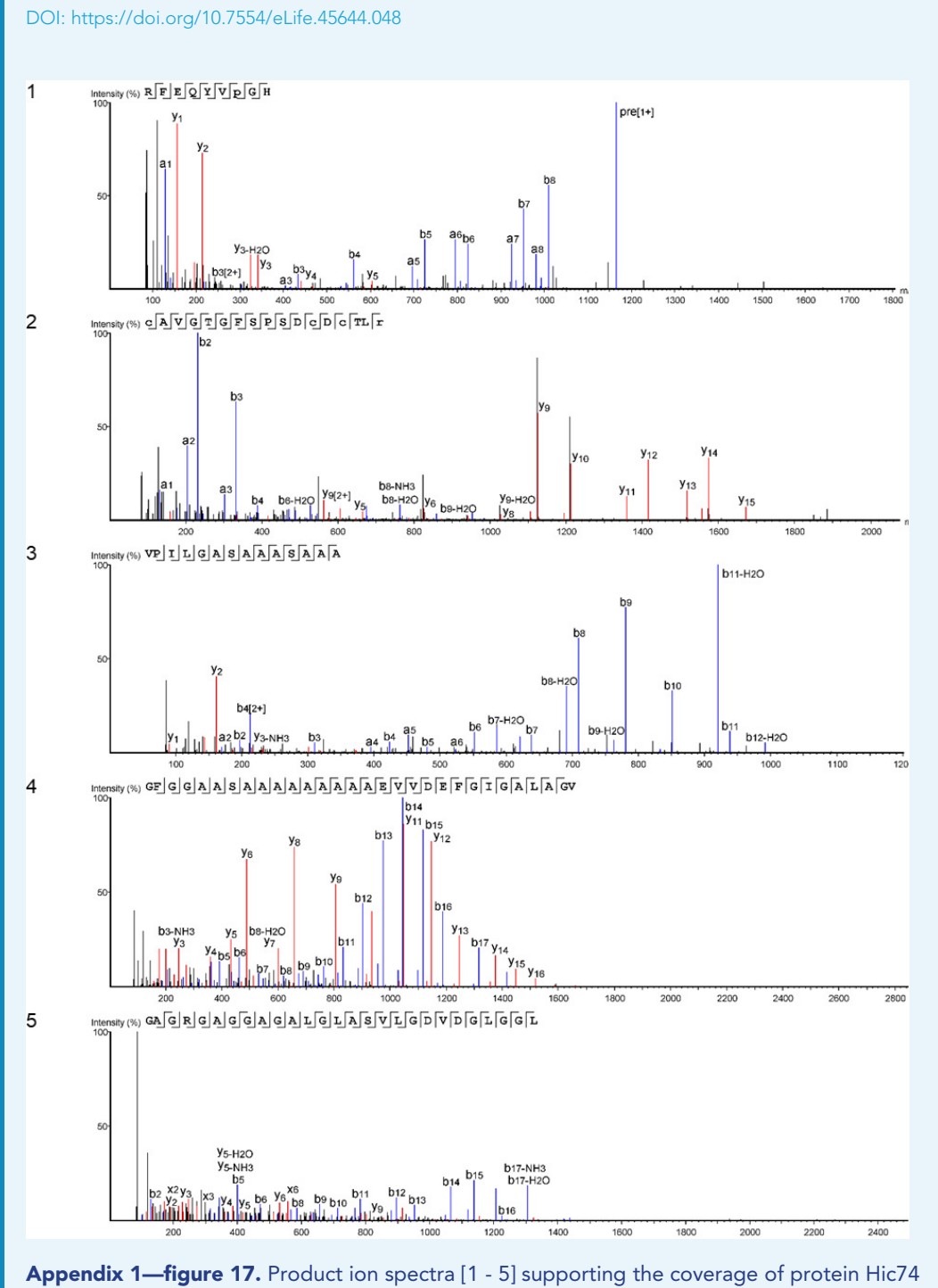

**Appendix 1—figure 17.** Product ion spectra [1 - 5] supporting the coverage of protein Hic74 identified in the double-button HavC.

DOI: https://doi.org/10.7554/eLife.45644.049

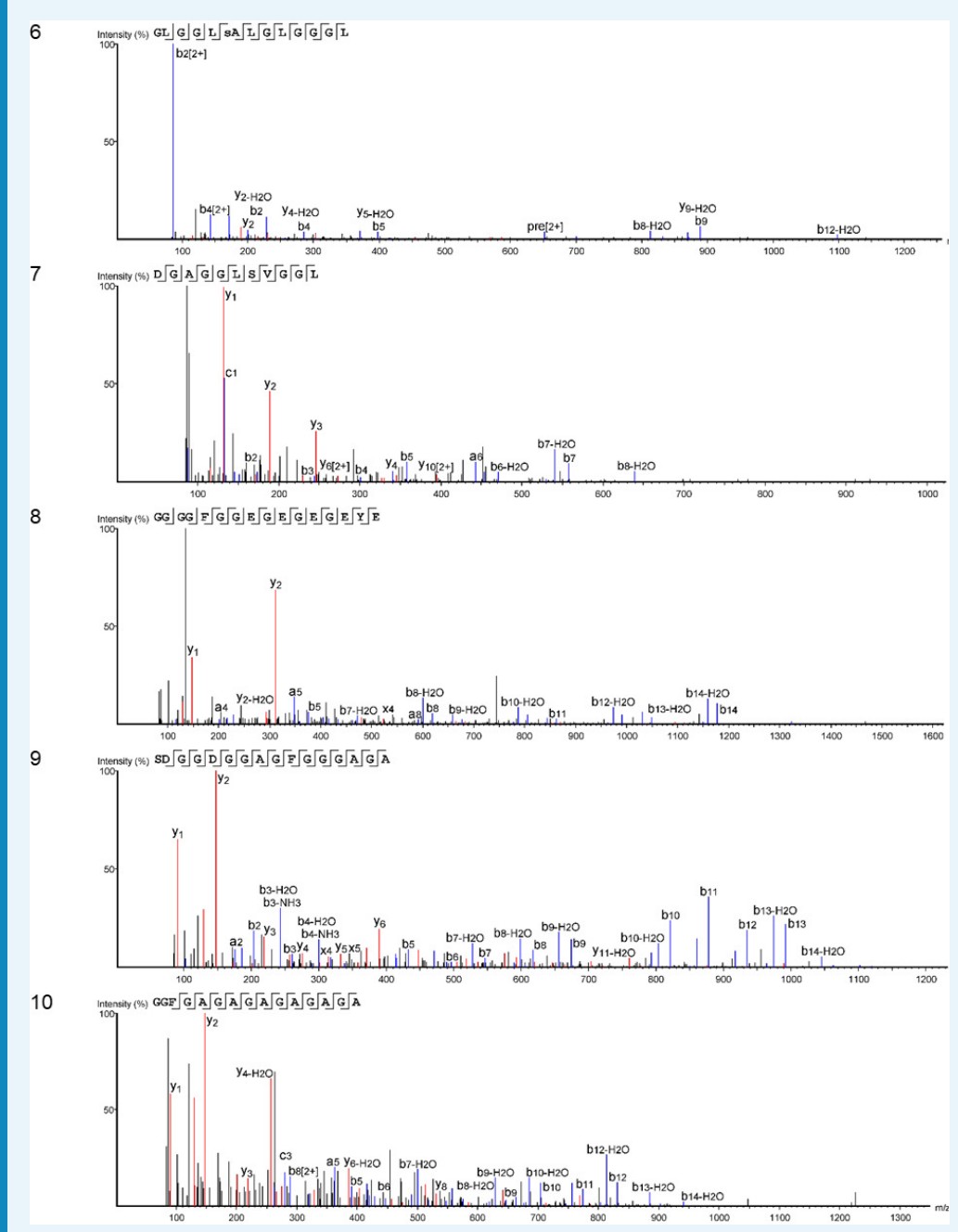

**Appendix 1—figure 18.** Product on spectra [6 -10] supporting the coverage of protein Hic74 identified in the double-button HavC.

DOI: https://doi.org/10.7554/eLife.45644.050

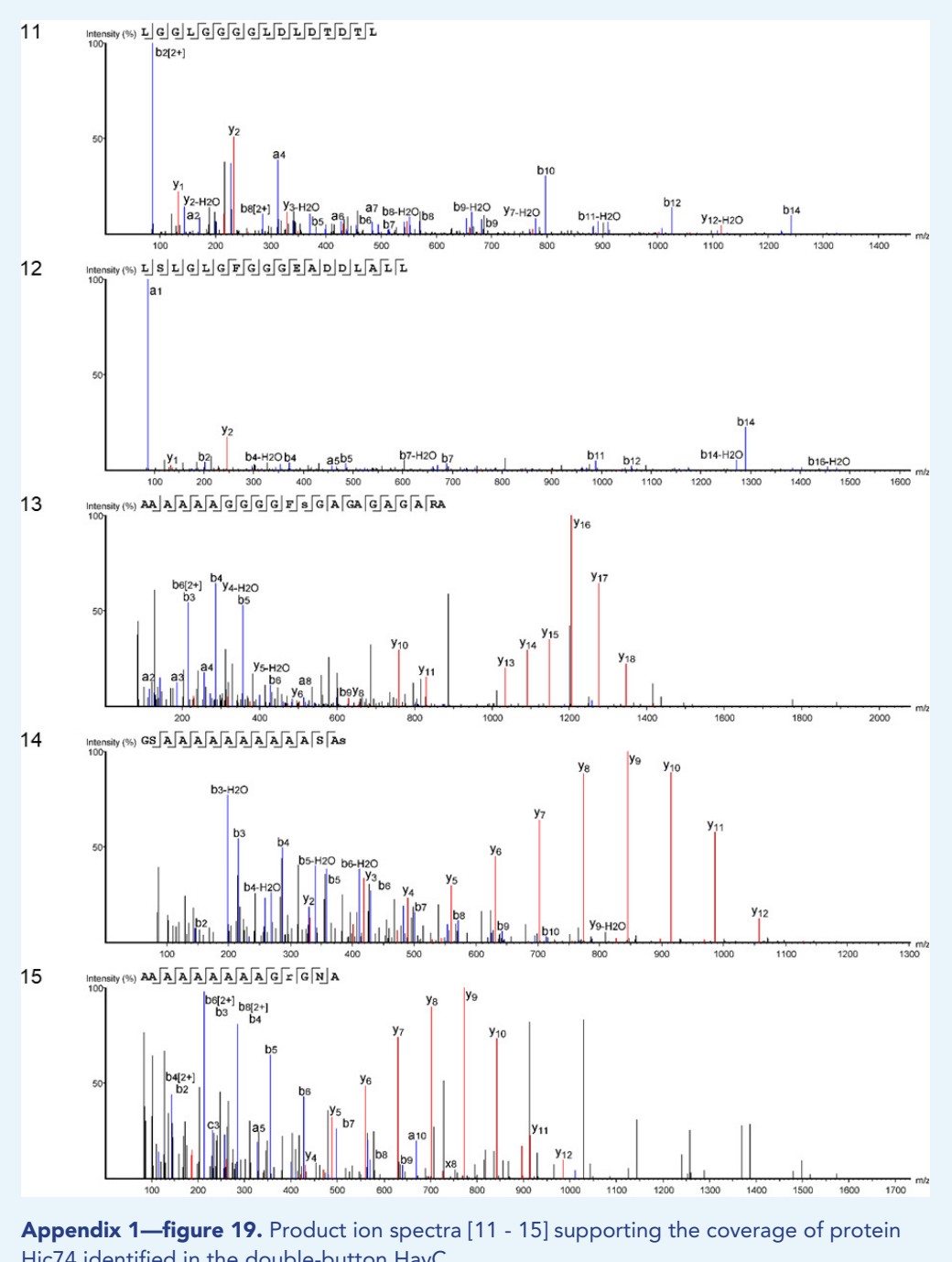

**Appendix 1—figure 19.** Product ion spectra [11 - 15] supporting the coverage of protein Hic74 identified in the double-button HavC.

DOI: https://doi.org/10.7554/eLife.45644.051

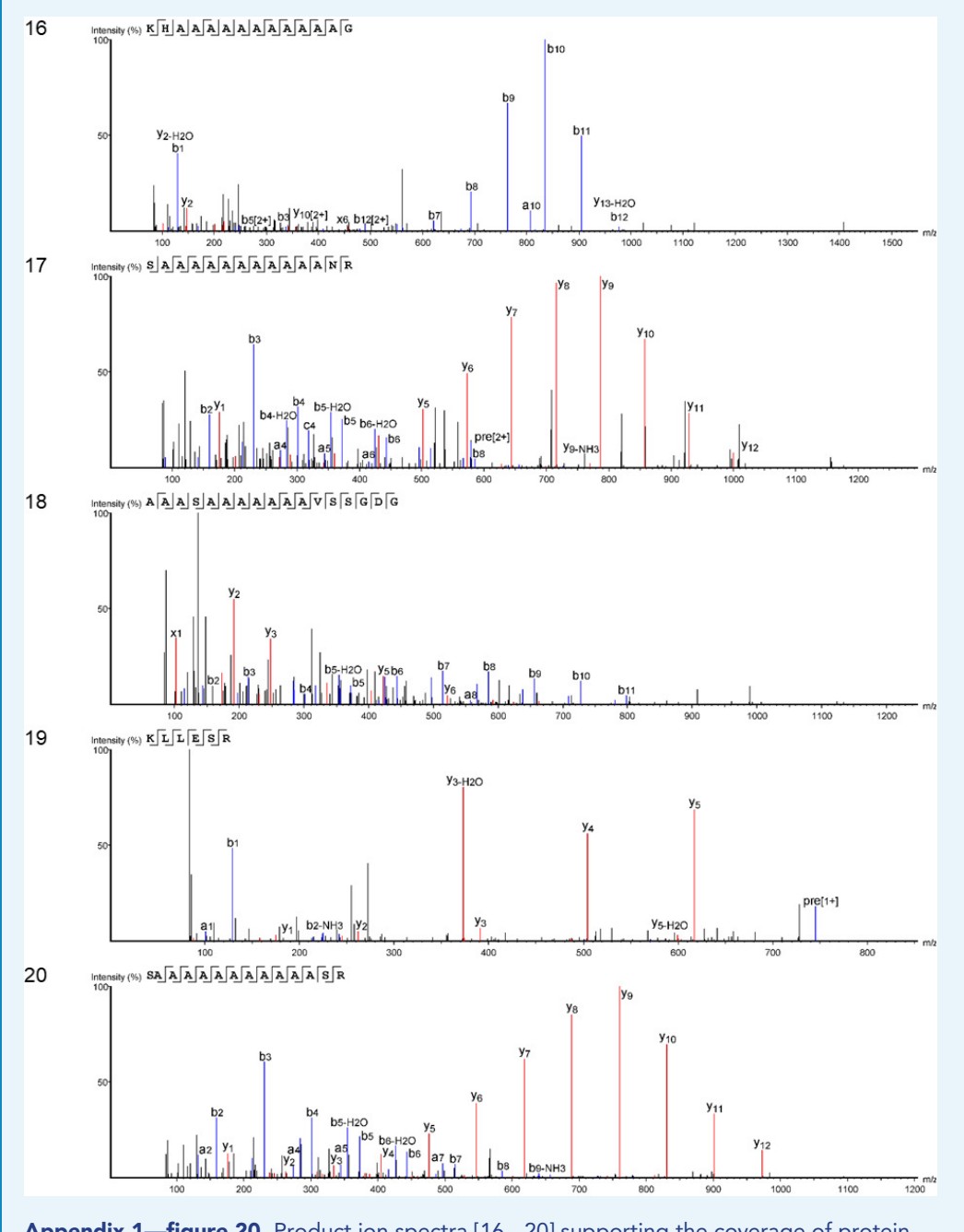

**Appendix 1—figure 20.** Product ion spectra [16 - 20] supporting the coverage of protein Hic74 identified in the double-button HavC.

DOI: https://doi.org/10.7554/eLife.45644.052

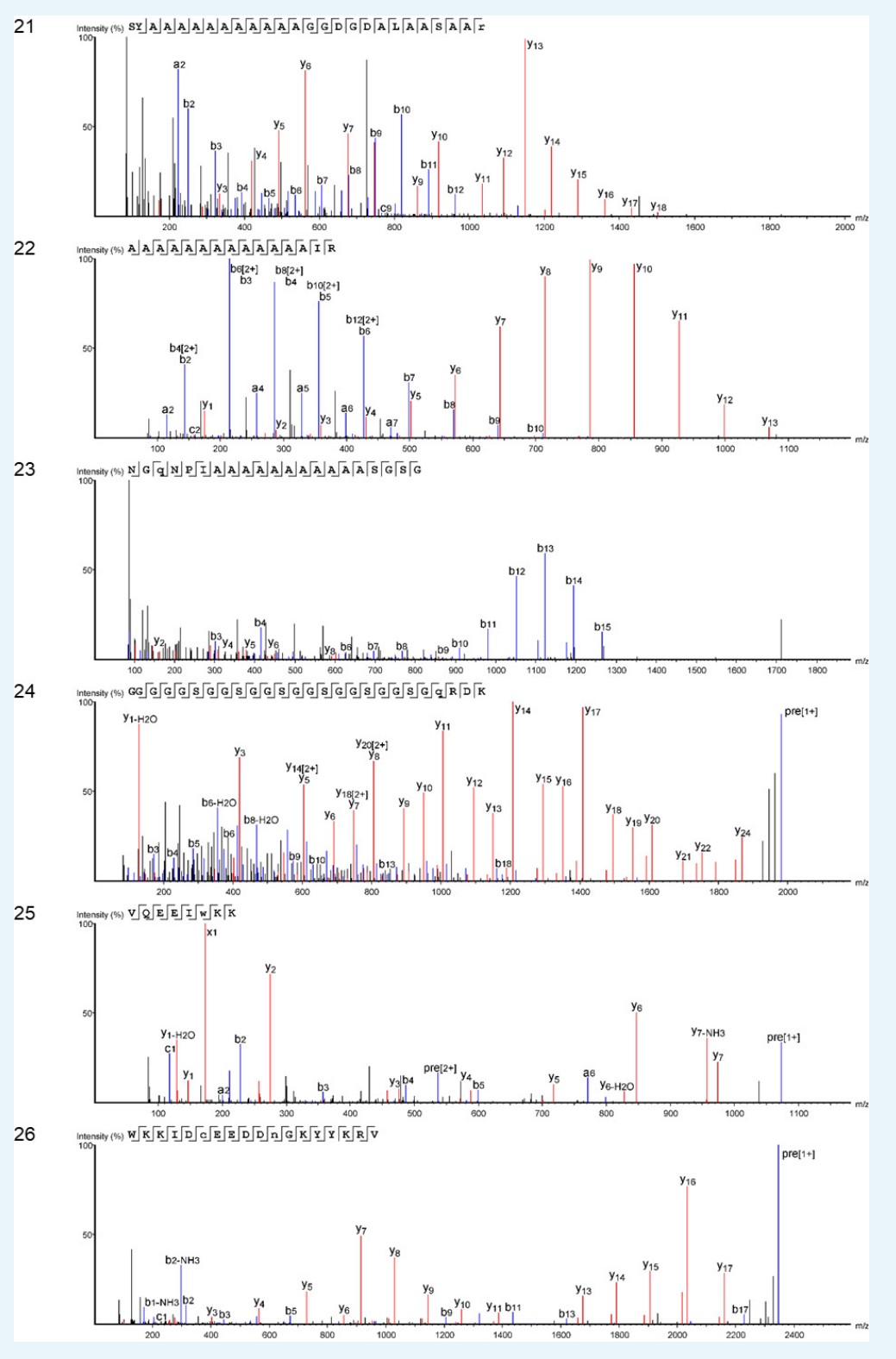

**Appendix 1—figure 21.** Product ion spectra [21 - 26] supporting the coverage of protein Hic74 identified in the double-button HavC.

DOI: https://doi.org/10.7554/eLife.45644.053

Hic74 coverages were 35%, 39% and 55% respectively, for HavA, HavB and HavC, where they were supported by a high number of peptides (132, 158, 260). In comparison, the

coverage of this protein obtained from the shell matrix of reference unionoid shells (where the extraction was performed on samples with sizes at least 100 times higher) was almost the same: 34% and 54% in modern *Margaritifera margaritifera* and *Unio pictorum* and 49% to 50% in sub-fossil unionoids *Pseudunio aricularius* and *Unio crassus*. Such results indicate that this protein is stable during early diagenesis. This might be partly due to the chemical stability of the Ala-Ala bonds that dominate the low complexity poly-Ala domains, as well as the mineral stabilization of the acidic motifs at the C-terminus of the protein, which showed a very high number of identified peptides, most of which were deamidated (*Appendix 1— figure 16*). This result also suggests that over geological times (but probably during early diagenesis), slow denaturation (uncoiling) of protein structures or splitting of cross-linkage sites with other organic macromolecules (i.e glycosides) may aid protein release, extraction and subsequent identification. PesB showed relatively low % coverage (19%) but this may be due to a species effect, as discussed elsewhere. Among the Hornstaad samples, the overall extent of preservation was low (for example Hic74 protein has a coverage of only 7% in HorB and 12% in HorC, supported by 6 and 11 peptides respectively), probably due to the effects of temperature. The exposure to heat of the archaeological biogenic carbonate did not induce recrystallization processes (as we did not observe any calcite in the ATR analyses), or extremely high D/L values in HorB and HorC, however the temperature was high enough to induce protein modifications and degradation.

Some proteins from other species, such as MSI60-related protein (*Pinctada*), glycine rich structural-like protein (*Crassostrea*) and precollagen D (*Mytilus*) were identified in PesB and all of the Havnø samples (*Appendix 1—figure 22*). However, the identification of these silk-like and collagen-like proteins in the archaeological samples were supported only by low complexity, Ala and Gly rich domains, exactly as for the freshwater unionoid shells, and could thus be attributed generally to the nacro-prismatic structure of the shell material.

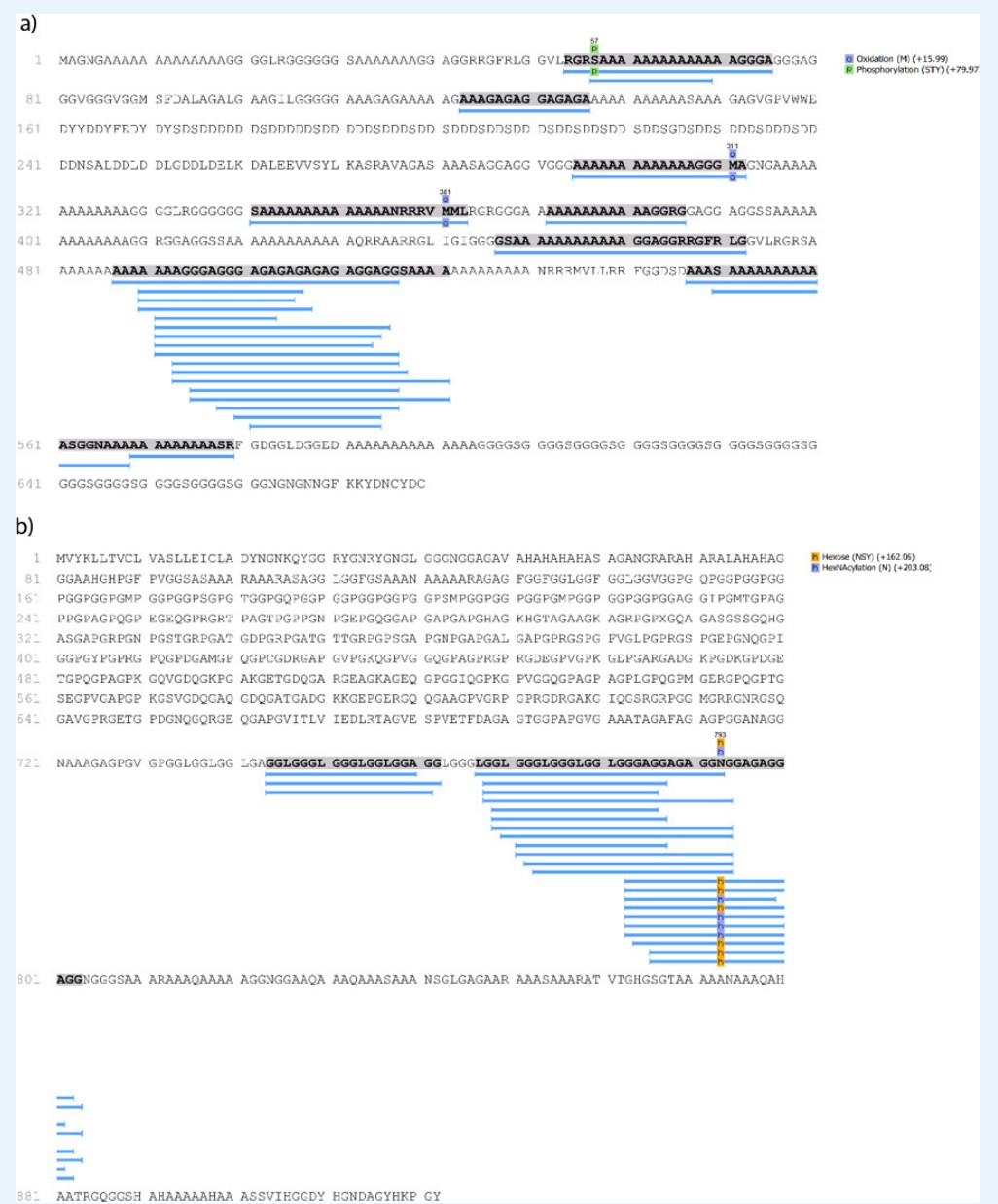

**Appendix 1—figure 22.** Marine shell proteins identified in double-button HavB: (**a**) MSI60-related protein (*Pinctada fucata*); (**b**) Precollagen D (*Mytilus edulis*). Note that both are supported only by repetitive low complexity (RLC) domains. Sequences reconstructed by assisted de novo on the basis of mono-charged ions mainly (spectra were acquired on the 400–1600 *m/z* range and multiply-charged ions were detected).

DOI: https://doi.org/10.7554/eLife.45644.054

### 3.5.9. Proteome similarities

Searching the de novo-reconstructed peptides against the expressed sequence tag (EST) database yielded additional peptide-protein matches, particularly in freshwater unionoid shells and archaeological samples. These included many hits from *Hyriopsis cumnigii* (e.g more than 20 EST hits in *U. pictorium*), mostly unannotated sequences (a search of *Hyriopsis cumingii* on NCBI will retrieve 246 protein sequences, but 10156 EST sequences).

The EST-derived dataset, consisting of lists of the unique identifiers of the ESTs retrieved from each sample, was used to explore the similarities between the proteomes of the reference shells and of the double-buttons. First, we attempted to understand whether it is

possible to discriminate the six reference shells taxa. The circular graph (*Appendix 1—figure 23*) represents the number of identified EST sequences in each of the shells and shows the proportion of sequences shared between species.

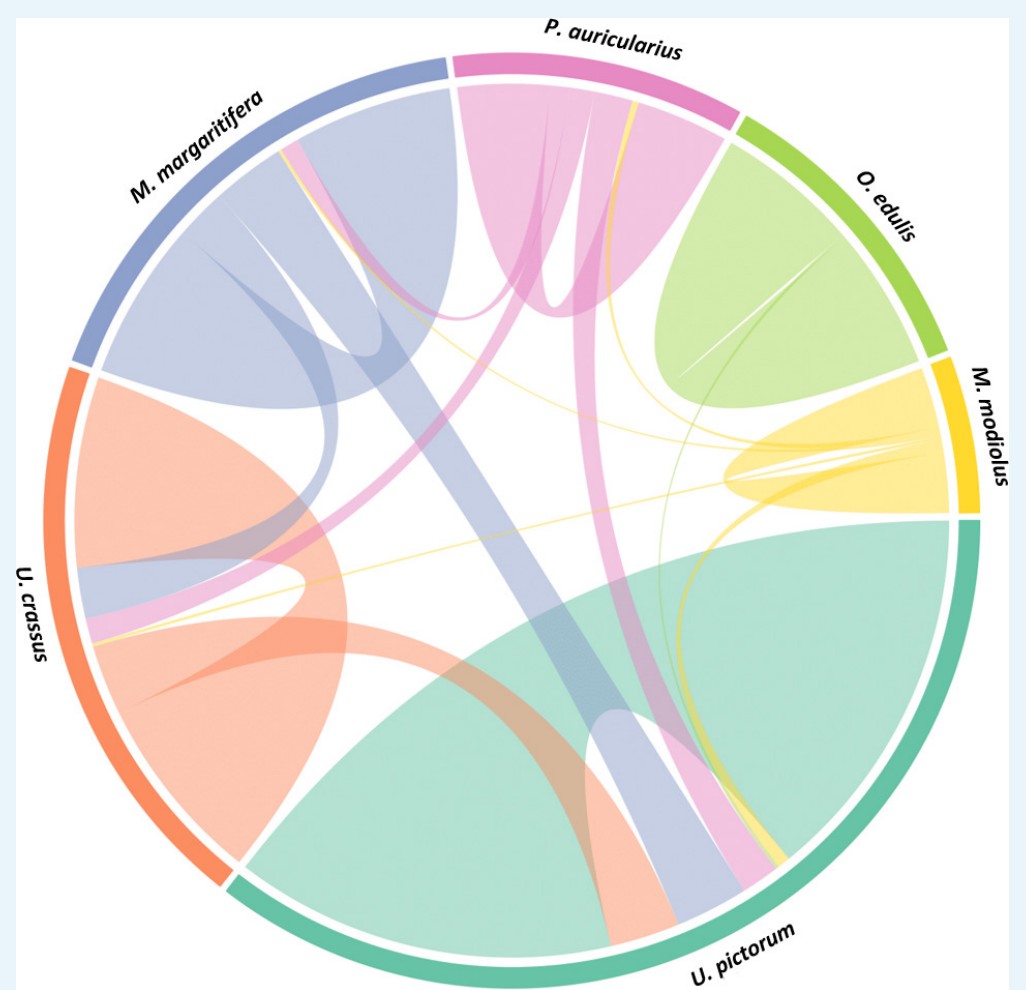

**Appendix 1—figure 23.** Circular diagram representing the extent of similarity between the proteomes of six reference mollusc shells based on the identified EST sequences.
DOI: https://doi.org/10.7554/eLife.45644.055

*Unio pictorum* samples yielded the highest number of EST hits (as seen from the proportion of the occupied circular area), while the marine shell *Modiolus modiolus* yielded the least. The four freshwater shells - *U. pictorum, U. crassus, M. margaritifera, P. auricularius* shared the highest proportion of sequences, while the two marine shells - *M. modiolus* and *O. edulis*, seemed to have very distinctive proteomes and only few sequences in common with the unionoid shells. These results indicate that there is a definite similarity between the proteomes of freshwater nacre shells here studied: *U. pictorum* is mostly similar to *U. crassus* and *M. margaritifera* and slightly less to *P. auricularius.* We cannot exclude, however, that this may be due to the fact that *U. pictorum* yielded the highest number of identified EST sequences out of all the unionoid shells. Few sequences were shared between freshwater *U. pictorum* and marine *M. modiolus*, owing to the nacro-prismatic shell structure, present in both. Overall, the data shows that marine and freshwater reference shells can be easily discriminated on the basis of their proteome profiles (even if they share the same nacro-prismatic structure). However, we stress that it is difficult to retrieve a simple phylogenetic signal within the group of unionoids.

The same analysis was repeated for the seven archaeological double-buttons and the results are presented in *Appendix 1—figure 24*. Searching against the EST database, the Havnø samples yielded the highest number of identified sequences (as can be seen in the figure, more than two thirds of the circular area is occupied by Havnø). While the highest degree of closeness was observed within the the same set of samples (Havnø and Hornstaad), a degree of similarity was also observed between the three different sets. This was particularly evident for HavA, which shared sequences not only with HavB and HavC, but also with all the three Hornstaad samples and with PesB. In the Hornstaad double-buttons we observed a lower amount of identified EST sequences and a more limited extent of similarity between sample sets. This is in accordance with our previous analysis, which showed that the preservation of the Hornastaad double-buttons was the least optimal. Nevertheless, the criss-crossing displayed by the circular diagram implies that the same raw material was likely used to manufacture the ornaments in all of the archaeological sites.

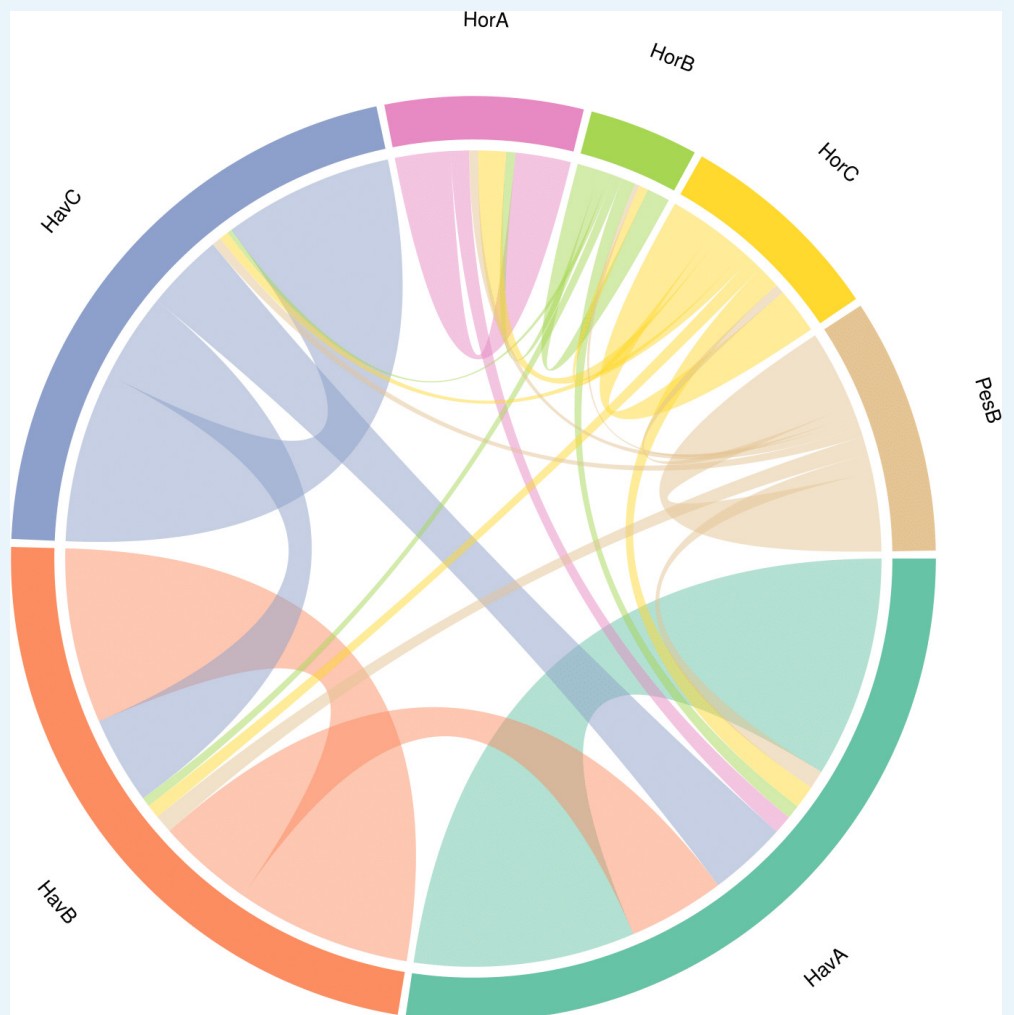

**Appendix 1—figure 24.** Circular diagram representing the similarity between the proteomes of the seven double-buttons based on the identified EST sequences.

DOI: https://doi.org/10.7554/eLife.45644.056

## 3.6. Database independent biomolecular comparisons

The classification of samples on a molecular level has been widely applied for phylogenetics to study the evolutionary relationships between species. However, database-dependant search algorithms of DNA and/or protein sequences cannot take into account peptides and

proteins that do not yet *exist* in databases. In recent years, as shotgun proteomics has become a routine tool for biochemistry, and huge amounts of data have been generated on novel systems and organisms (including extinct), there has been an increasing interest in developing methods enabling to compare proteome-wide measurements that would be flexible and allow database-free workflows (*Palmblad and Deelder, 2012*; *Rieder et al., 2017*; *Yılmaz et al., 2016*).

### 3.6.1. Peptide sequence similarity

The peptide sequences generated by de novo algorithm of the software PEAKS Studio 8.5 (Bioinformatics Solutions Inc) were used in a database-independent biomolecular comparison. Any peptides belonging to contaminants were removed prior to the analysis, by searching each sample against a common laboratory contaminant database (cRAP; https://www.thegpm.org/crap/). The lists generated by this analysis contained peptide sequences reconstructed by the PEAKS software on the basis of the raw product ion spectra. Only common post translational (phosphorylation) and diagenesis-induced (deamidation, oxidation and dioxidation, pyro-Glu formation) modifications were taken into account by the PEAKS algorithm. Each sequence is present only once (unique), thus the number of peptide sequences does not affect the overall analysis.

The algorithm, developed in-house in C language, provides a score for the sequence similarity between two lists of peptides. This is achieved by finding the best-matching peptide in the second file for each peptide in the first, taking into account the number of consecutive matches and mismatches within the overlapping region of the two peptides. The scores for the best matches (which may be zero) are then combined to give a score for the similarity between the two files. However, the similarity score from file 1 to file two is not necessarily the same as the score from file 2 to file one as the two files may contain different numbers of peptides. Therefore, a score is also calculated from the best match in the first file to each peptide in the second and the two scores averaged to provide the similarity metric. A similarity matrix is obtained from all pairwise comparisons and converted to a distance matrix.

Multidimensional scaling (MDS) is a technique that allows the similarity of individual observations (here lists of peptides) to be visualised (*Gower, 1966*). The method provides new coordinates in just two dimensions that represent the information in a distance matrix. *Figure 5* (main text) shows the map obtained from the distance matrix calculated by scoring peptide matches between the buttons and several mollusc species. The plot illustrates the great level of similarity of all the Havnø archaeological samples with the freshwater unionoid shells, especially *Unio pictorum*, *Unio crassus* and *Pseudunio auricularius*. The Hornstaad samples cluster slightly apart, likely because of alterated proteome profiles due to burning, and the PesB sample falls a bit further from the biggest unionoid cluster, but far from the marine shells. Therefore, the MDS plot obtained on the basis of the reconstructed de novo sequences was coherent with the pattern highlighted by the bulk amino acid compositions (*Appendix 1—figure 8*).

### 3.6.2. Proteome wide distance calculations based on product ion spectra

We attempted to perform comparative shell proteome analysis based solely on the proteome-wide distance calculation of different product ion spectra using a DISMS2 algorithm implemented in R language (*Rieder et al., 2017*). The algorithm is a four-step procedure, consisting of spectra filtering, checking constraints for matching, matching of product ion spectra and calculation of the distance matrix with pairwise distances between runs. Raw data of each sample were processed using PEAKS Studio 8.5 (Bioinformatics Solutions Inc) - searched against a common laboratory contaminant database cRAP (https://www.thegpm.org/crap/) to eliminate the possibility of matching contaminant spectra among the different samples. The exported de novo only peptide spectral data were converted to mzXML format using ProteoWizard and used directly as a primary input in R. Since the parameters are flexible, during the first preprocessing step, the topn function was disabled - considering that some sub-fossil samples and archaeological beads may be altered, all of the

peaks (and not only those with the highest intensities) were included in the analysis. Binning was done with a fixed bin size where bin = 0.01 (a trial run without binning did not show any changes for the distance matrix calculated, only the time of analysis was prolonged). Cosine distance (dcos) was used to calculate the distance between the mass spectra and the cutoff value (cdis) was set at 0.3. The obtained dissimilarity matrix was visualised by hierarchical clustering using the package *factoextra*, and clustering performed with the *ward.D2* method.

A dendrogram representation was used to explore the shell proteome differences between samples (*Appendix 1—figure 25*). The position of *M. modiolus*, which also has a nacreous inner structure with an upper prismatic layer, clustering apart from all of the freshwater unionoids, shows the the evolution and adaptation of different calcification processes (*Arivalagan et al., 2017*; *Marie et al., 2009*). In fact, *M. modiolus* clustering together with *O. edulis* supports our previous results, that is that marine shells have a distinct proteome profile, diverging from freshwater molluscs. The archaeological double-buttons from Hornstaad and Havnø cluster close to *Unio pictorum,* except for HavA, which falls together with the freshwater pearl mussel *Margaritifera margaritifera*. Sub-fossil samples of *Unio crassus* and *Pseudunio auricularius* cluster together independently and this could be explained by the vicinity of shell proteomes and possibly the alteration of the profiles due to diagenesis. The PesB sample falls in the middle, not showing a clear and evident proximity to one particular sample, suggesting phylogenetic distance or alteration of the sample, coherently with chiral amino acid data (*Appendix 1—figure 8*) and all other database-dependant analyses. Overall, the consistent pattern revealed by database-dependent and database-independent protein data analysis as well as chiral amino acid analysis (these were all performed in the same laboratory in York) strongly support a biological effect (phylogenetic or diagenetic) and not an analytical effect for the clustering.

## Cluster dendrogram of pairwise MS/MS spectra matching

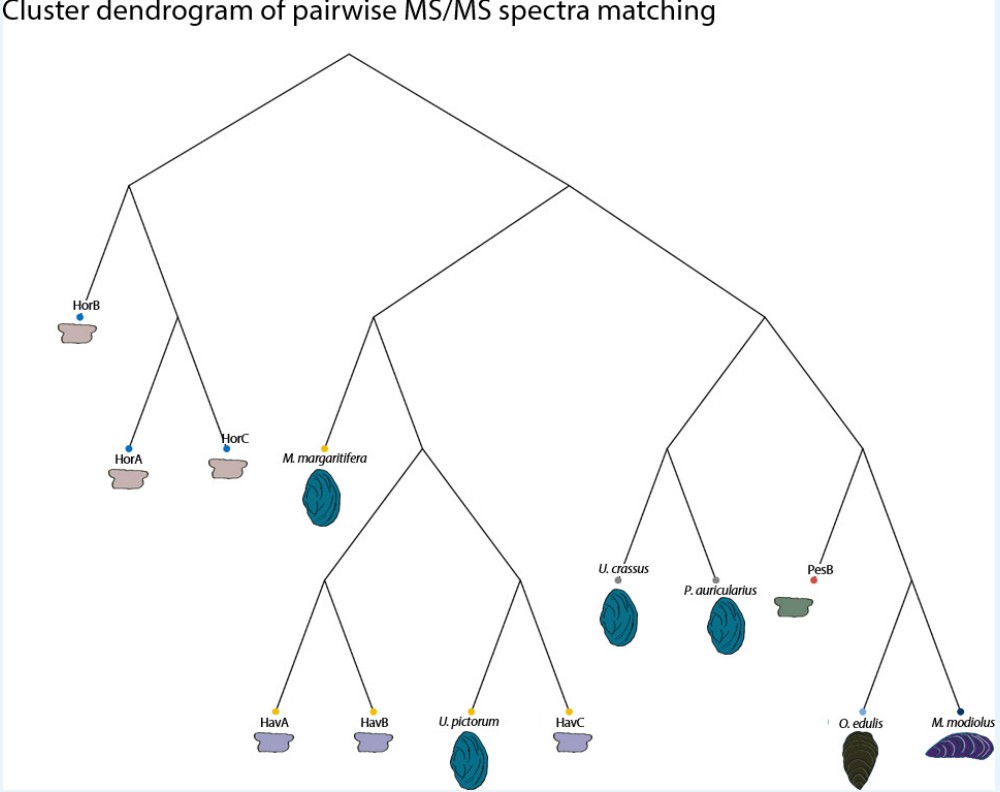

**Appendix 1—figure 25.** Pairwise MS/MS comparison of the seven archaeological double-buttons and six reference shells (freshwater and marine): the cluster dendrogram is obtained from a distance matrix from proteome-wide distance calculations of product ion spectra implemented in R using the DISMS2 code.

DOI: https://doi.org/10.7554/eLife.45644.057

