## [Decision Letter]

Thank you for submitting your article "Pearls of grace: "Palaeoshellomics" reveals the use of freshwater mother-of-pearl in prehistory" for consideration by *eLife*. Your article has been reviewed by three peer reviewers, and the evaluation has been overseen by a Reviewing Editor and Detlef Weigel as the Senior Editor. The following individual involved in review of your submission has agreed to reveal their identity: Peter Rowley-Conwy (Reviewer #2).

The reviewers have discussed the reviews with one another and the Reviewing Editor has drafted this decision to help you prepare a revised submission.

Summary:

This manuscript applies proteomics methods to the study of calcified mollusk tissue with the primary aim of obtaining taxonomic/phylogenetic information. The general application is to distinguish marine from freshwater shell in archaeological contexts. The manuscript is very well-written, lucid, and represents a significant advance in methodology. The double-buttons are widespread, and it is very interesting that they were produced using local freshwater mother-of-pearl. This is a result of major importance, indicating a common symbolic approach to this type of ornament: however the material and the objects were understood, it is clear that the understanding was the same all over the area where the buttons are found – a remarkably extensive area, as Figure 1 makes clear. The reviewers were uniformly positive, and feel that with minor revisions this manuscript should be published in *eLife*.

Application to shell is a sorely-needed methodology in archaeology for all of the reasons the authors clearly state, and this initial application to archaeological "double-buttons" is a good test case for initial work in this area. The ability to obtain taxonomic information from mollusk shells that have been fragmented and/or abraded to a point where they can no longer be identified is potentially transformative in many archaeological contexts. These include cases such as that provided here, where ornaments may contain valuable social information that has implications for the sourcing and modification of raw materials. They can also include cases where morphologically-similar species can have different habitats that have implications for seasonality, diet, and other aspects of subsistence behavior. Thus, the manuscript offers a novel and exciting use of proteomics to ancient remains. The results are somewhat discouraging as no reliable phylogenetic signal is present within the genus *Unio*, which dominates freshwater species and exhibits the nacreous layer that is desirable for ornament manufacture. However, this is likely attributable to the incomplete nature of existing proteomic databases, and this contribution provides valuable data that will help move this work forward.

Essential revisions:

1) Slight revision of the framing: There is somewhat of a disjoin between the goals of this manuscript and the (need for) the data that are presented. Both the goals and the methods are independently significant, and certainly the data do support the conclusions. However, it is also clear whilst reading the manuscript that the main hypotheses set out at the start were largely answered already without the use of palaeoshellomics – which is the main methodological premise of the paper. The authors first use oxygen and carbon and find evidence for a local source. Of course, the difference between marine and freshwater shell should also be apparent in Sr isotope ratios, which they (mysteriously) do not attempt. Therefore, if the marine-freshwater dichotomy was the critical problem being tested, there was already a way (actually multiple ways) to do this. In the Discussion, all other lines of evidence prior to the use of proteomics appear to point to a freshwater origin (subsection “Raw material identification”, first paragraph), so one has to ask what information is added by the highly-involved proteomic analysis. Additional independent lines of evidence are always welcome, but then the question becomes one of cost and feasibility. In short, what does the palaeoproteomics give us that we need in order to do this work? If the answer is in this case not much, then perhaps the framing of the problem itself should be subtly revised. It might be that the authors initially hoped to have finer phylogenetic resolution, and that when this was not the case they moved to a simple marine/freshwater dichotomy. If so, then this should be stated.

2) Clarification of use of archaeological shells: There needs to be a clear explanation for why the archaeological shells *U. crassus* and *P. auricularius* are used as reference shells (main text, last paragraph), rather than modern shells. Archaeological specimens are more ambiguous to use as (morphological) references because the taxon is not known with the same certainty as a modern specimen. This is likely less of a problem for invertebrates than it is for vertebrates, but this should still be addressed: how confident are the identifications, and on what basis? What was the context of their recovery? Why were reference shells selected from these specific sites? Is it because they are suspected to be the raw material for the double-buttons, or simply because this is what was available? Even if the morphological identifications are very strong, what is the advantage of using archaeological references rather than modern references? Is it because they are likely to have had similar depositional histories to the double-buttons, and this could affect the proteomic analysis? Subsection “Raw material identification”, point #2) – are any of the published "shellomes" from those taxa analyzed here as references, or from their close relatives? Perhaps part of the value of the paper is in its description of new protein spectra (shellomes) for these taxa, but if so that should be made explicit. It is somewhat hidden away in the conclusions, but might actually make a better foundation for the paper since the aim of distinguishing marine versus freshwater appears to be largely achieved even before the proteomic work.

3) Discussion: As a submission to *eLife*, which primarily deals with the life sciences, I would recommend shortening the more theoretical and speculative aspects of the Discussion about river symbology at the end of the manuscript. The manuscript would benefit more from using this space to describe the shells a bit more. Much of this is in the supplementary data, but could also be summarized here in the main body. Are any of these taxa (modern and archaeological) edible, and may they have been collected for this purpose as well? How old are the double-buttons from each of the sites, and what was the context for each? One can follow the citations and of course look at the supplementary data, but it would be good to have a table that summarizes the context, dates, and arguments about the double-buttons from each site, as well as what archaeological mollusk species have been found at each site that might serve as raw material, food, or simply naturally occur there. The connections between these specific sites and the questions about them as more than simply applying a single broad brush about "river symbolism" are also unclear.

4) Source code: Subsection “Database-independent comparison”: An unpublished tool was used to generate scores for the distance matrix underlying Figure 4. The statistics underlying the tool are not specified. Is the source code for this tool available? It does not seem to be included in the submitted article files, and should be submitted.

---

## [Author Response]

Summary:[…] Application to shell is a sorely-needed methodology in archaeology for all of the reasons the authors clearly state, and this initial application to archaeological "double-buttons" is a good test case for initial work in this area. The ability to obtain taxonomic information from mollusk shells that have been fragmented and/or abraded to a point where they can no longer be identified is potentially transformative in many archaeological contexts. These include cases such as that provided here, where ornaments may contain valuable social information that has implications for the sourcing and modification of raw materials. They can also include cases where morphologically-similar species can have different habitats that have implications for seasonality, diet, and other aspects of subsistence behavior. Thus, the manuscript offers a novel and exciting use of proteomics to ancient remains. The results are somewhat discouraging as no reliable phylogenetic signal is present within the genus Unio, which dominates freshwater species and exhibits the nacreous layer that is desirable for ornament manufacture. However, this is likely attributable to the incomplete nature of existing proteomic databases, and this contribution provides valuable data that will help move this work forward.

In this revised version, we have especially focussed on clarifying that, while we could not obtain identification to the level of species, “palaeoshellomics” allowed us to go beyond the distinction of freshwater vs marine. In fact, we could confidently identify the order of the shells (Unionoida) by searching the mass spectrometry data against a public database of molluscan protein sequences (marine and freshwater). We now show that the analysis of the sequence of the top-scoring protein (Hic74), recovered from the ornaments and the reference unionoids, supports *Unio* or *Margaritifera* and excludes Pseudunio as a potential raw material for the Havnø double-buttons.

We have added the following to the Results and Discussion section, as well as a new Figure 6, Table 3, and supplementary figures:

“Analysis of the Hic74 sequence. We examined the sequence of the top-scoring protein, Hic74, recovered from the reference shells and ornaments, with the aim of assessing the presence and frequency of any amino acid substitutions, which could potentially yield taxonomic resolution within Unionoida. […] Furthermore, Table 3 shows that this taxon was unlikely to be *P. auricularius*, and more likely to be *Unio* or *Margaritifera*.”

Essential revisions:1) Slight revision of the framing: There is somewhat of a disjoin between the goals of this manuscript and the (need for) the data that are presented. Both the goals and the methods are independently significant, and certainly the data do support the conclusions. However, it is also clear whilst reading the manuscript that the main hypotheses set out at the start were largely answered already without the use of palaeoshellomics – which is the main methodological premise of the paper. The authors first use oxygen and carbon and find evidence for a local source. Of course, the difference between marine and freshwater shell should also be apparent in Sr isotope ratios, which they (mysteriously) do not attempt. Therefore, if the marine-freshwater dichotomy was the critical problem being tested, there was already a way (actually multiple ways) to do this. In the Discussion, all other lines of evidence prior to the use of proteomics appear to point to a freshwater origin (subsection “Raw material identification”, first paragraph), so one has to ask what information is added by the highly-involved proteomic analysis. Additional independent lines of evidence are always welcome, but then the question becomes one of cost and feasibility. In short, what does the palaeoproteomics give us that we need in order to do this work? If the answer is in this case not much, then perhaps the framing of the problem itself should be subtly revised. It might be that the authors initially hoped to have finer phylogenetic resolution, and that when this was not the case they moved to a simple marine/freshwater dichotomy. If so, then this should be stated.

We agree with the reviewers that the manuscript appeared to have the main goal as to determine marine vs. freshwater origin of the ornaments, and that this would have been easily accomplished using stable isotope analysis, without the need for palaeoshellomics. However, our primary focus was indeed to obtain molecular taxonomic identification, and we used oxygen and carbon isotopes to obtain an indication of the local vs exotic origin of the shells. We did not deem Sr isotope analyses necessary, as the freshwater origin was already supported by multiple lines of evidence.

Palaeoshellomics is indeed a costly and intensive enterprise, and the interpretation of the data is not straightforward. However, it allowed the confident identification of Unionoida (among all possible freshwater and marine orders) as the raw material for all double-buttons, and suggested the genera *Unio/Margaritifera* for the Havnø set. Furthermore, as more molluscan genomes become available and more palaeoshellomics datasets are obtained and published, identification will become more straightforward. We have clarified these issues throughout the manuscript and added new data (summarised in Figure 6 and Table 3), which support our statements.

We have also revised the framing as suggested, by stating more clearly in the Introduction that we had two main aims, one methodological (developing “palaeoshellomics”) and one essentially cultural (settling the debate over the shell taxon used to make the ornaments). In order to do so, we have added the following:

“Our work had two main aims:

1) to develop “palaeoshellomics”, a new molecular approach to characterize ancient proteomes preserved in mollusc shells. […] This is archaeologically significant, as similar ornaments were recovered from three geographically distant sites belonging to three different and broadly contemporary cultural groups (Table 1): Late Mesolithic (Ertebølle), early Late Neolithic (Hornstaad Group), and Copper Age (Toarte Pastilate/Coţofeni).“

2) Clarification of use of archaeological shells: There needs to be a clear explanation for why the archaeological shells U. crassus and P. auricularius are used as reference shells (main text, last paragraph), rather than modern shells. Archaeological specimens are more ambiguous to use as (morphological) references because the taxon is not known with the same certainty as a modern specimen. This is likely less of a problem for invertebrates than it is for vertebrates, but this should still be addressed: how confident are the identifications, and on what basis? What was the context of their recovery? Why were reference shells selected from these specific sites? Is it because they are suspected to be the raw material for the double-buttons, or simply because this is what was available? Even if the morphological identifications are very strong, what is the advantage of using archaeological references rather than modern references? Is it because they are likely to have had similar depositional histories to the double-buttons, and this could affect the proteomic analysis? Subsection “Raw material identification”, point #2) – are any of the published "shellomes" from those taxa analyzed here as references, or from their close relatives? Perhaps part of the value of the paper is in its description of new protein spectra (shellomes) for these taxa, but if so that should be made explicit. It is somewhat hidden away in the conclusions, but might actually make a better foundation for the paper since the aim of distinguishing marine versus freshwater appears to be largely achieved even before the proteomic work.

We have added a new section (“sample selection”), which also includes a new table (Table 1), summarising all information available on both the reference shells and the ornaments. We detail the rationale behind the choice of each reference shell (modern and archaeological), and the method used for their taxonomic determination, with references to the relevant publications.

“Sample selection

Two modern marine mollusc shells, *O. edulis* and *M. modiolus*, were collected in northern Jutland (Denmark) by Søren H. Andersen and were selected for the following reasons: *O. edulis* shells had been suggested as the potential raw material for the Hornstaad-Hörnle IA assemblage (Heumüller, 2010) and are very abundant at the shell midden site of Havnø; *M. modiolus* is a thick-shelled mussel with a nacreous layer, therefore a suitable raw material for the Havnø ornaments (Appendix 1, section 2). […] None of the Unionoida species is well-represented in public sequence databases, especially with regard to proteins related to shell biomineralization (see Materials and methods section).”

We also added important information on the presence or absence of suitable reference shellomes in public databases (Materials and methods section):

“The Molluscan Protein Database used in this study comprised 633,061 protein sequences, i.e. all sequences available on the National Center for Biotechnology Information (NCBI) repository restricting the taxonomy to Mollusca (fasta database downloaded on 15/02/2018), excluding all common contaminants (cRAP; common Repository of Adventitious Proteins: http://www.thegpm.org/crap/). […] The same search for Margaritiferidae yields 207 entries, none of which related to shell matrix proteins.”

3) Discussion: As a submission to eLife, which primarily deals with the life sciences, I would recommend shortening the more theoretical and speculative aspects of the Discussion about river symbology at the end of the manuscript. The manuscript would benefit more from using this space to describe the shells a bit more. Much of this is in the supplementary data, but could also be summarized here in the main body. Are any of these taxa (modern and archaeological) edible, and may they have been collected for this purpose as well? How old are the double-buttons from each of the sites, and what was the context for each? One can follow the citations and of course look at the supplementary data, but it would be good to have a table that summarizes the context, dates, and arguments about the double-buttons from each site, as well as what archaeological mollusk species have been found at each site that might serve as raw material, food, or simply naturally occur there. The connections between these specific sites and the questions about them as more than simply applying a single broad brush about "river symbolism" are also unclear.

We agree with this suggestion and the new table (Table 1, in section “sample selection”, see above) provides information on the archaeological context, the chronology of the ornaments and the site, a summary of the potential identification of the raw material prior to our study and the presence of other shell taxa (marine, freshwater, edible, exotic, local). In the Discussion section, we softened the statements about river symbolism and shortened the paragraph regarding the role of freshwater environments as follows:

“The choice of the raw material could also be a reflection of the role of freshwater environments: the Neolithic is the period in which water, together with plants and animals, is “domesticated” (Garfinkel et al., 2006; Mithen, 2010). […] Despite their “fluidity”, rivers and lakes were meaningful and persistent places in the prehistoric landscape.”

We added a “conclusions” section, where we stated the methodological and archaeological significance of our findings, as follows:

“Conclusions

The first application of “palaeoshellomics” has demonstrated that it is possible to recover and identify ancient proteins sequences from mollusc shell, despite significant analytical challenges due to the combined effects of several factors, including low protein concentrations, small samples sizes, diagenesis and database insufficiency (Table 2). […] Our in-depth study therefore puts into question the most commonly accepted interpretations, which privilege the preponderant use of exotic marine shells as prestigious raw materials for the manufacture of prehistoric shell ornaments.”

4) Source code: Subsection “Database-independent comparison”: An unpublished tool was used to generate scores for the distance matrix underlying Figure 4. The statistics underlying the tool are not specified. Is the source code for this tool available? It does not seem to be included in the submitted article files, and should be submitted.

We have provided the code with the revised manuscript: Figure 5—source code 1 (pepmatch).